# BO-Muse: A human expert and AI teaming framework for accelerated experimental design

## Abstract

In this paper we introduce BO-Muse, a new approach to human-AI teaming for the optimisation of expensive blackbox functions. Inspired by the intrinsic difficulty of extracting expert knowledge and distilling it back into AI models and by observations of human behaviour in real-world experimental design, our algorithm lets the human expert take the lead in the experimental process. The human expert can use their domain expertise to its full potential, while the AI plays the role of a muse, injecting novelty and searching for areas of weakness to break the human out of over-exploitation induced by cognitive entrenchment. With mild assumptions, we show that our algorithm converges sub-linearly, at a rate faster than the AI or human alone. We validate our algorithm using synthetic data and with human experts performing real-world experiments.

## 1 Introduction

Bayesian Optimisation (BO) (Shahriari et al., 2015) is a popular sample-efficient optimisation technique to solve problems where the objective is expensive. It has been applied successfully in diverse areas (Greenhill et al., 2020) including material discovery (Li et al., 2017), alloy design (Barnett et al., 2020) and molecular design (Gómez-Bombarelli et al., 2018). However, standard BO typically operates tabula rasa, building its model of the objective from minimal priors that do not include domain-specific detail. While there has been some progress made incorporating domain-specific knowledge to accelerate BO (Li et al., 2018; Hvarfner et al., 2022) or transfer learnings from previous experiments (Shilton et al., 2017), it remains the case that there is a significant corpus of knowledge and expertise that could potentially accelerate BO even further but which remain largely untapped due to the inherent complexities involved in knowledge extraction and exploitation. In particular, this often arises from the fact that experts tend to organise their knowledge in complex schema containing concepts, attributes and relationships (Rousseau, 2001), making the elicitation of relevant expert knowledge, both quantitative and qualitative, a difficult task.

Experimental design underpins the discovery of new materials, processes and products. However, experiments are costly, the target function is unknown and the search space unclear. To be sample-efficient, the least number of experiments must be performed. Traditionally experimental design is guided by (human) experts who uses their domain expertise and intuition to formulate an experimental design, test it, and iterate based on observations. Living beings from fungi (Watkinson et al., 2005) to ants

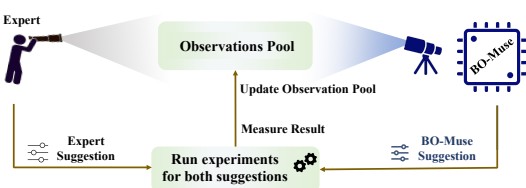

Figure 1: Bo-Muse Workflow.

(Pratt & Sumpter, 2006) and humans (Daw et al., 2006; Cohen et al., 2007) face a dilemma when they make these decisions: exploit the information they have, or explore to gather new information. How humans balance this dilemma has been studied in Daw et al. (2006)– examining human choices in a n-arm bandit problem, they showed that humans were highly skewed towards exploitation. Moreover, when the task requires specialised experts, cognitive entrenchment is heightened and the balance between expertise and flexibility swings further towards remaining in known paradigms.

To break out of this, dynamic environments of engagement are needed to force experts to incorporate new points of view (Dane, 2010). For such lateral thinking to catalyse creativity, Beaney (2005) has further confirmed that random stimuli are crucial. For example, using random stimuli to boost creativity has been attempted in the context of games (Yannakakis et al., 2014). In Sentient Sketchbook, a machine creates sketches that the human can refine, and sketches are readily created through machine learning models trained on ample data. Other approaches use machine representations to learn models of human knowledge, narrowing down options for the human to consider. Recently, Vasylenko et al. (2021) constructs a variational auto-encoder from underlying patterns of chemistry based on structure/composition and then a human generated hypothesis guides possible solutions. An entirely different approach refines a target function by allowing machine learning to discover relations between mathematical objects, and guides humans to make new conjectures (Davies et al., 2021). Note, however, that there is still a requirement for large datasets to formulate representations of mathematical objects, which is antithetical to sample-efficiency as typically in experimental design we have a budget on the number of experiments, data from past designs is lean, and formulation of hypothesis is difficult in this lean data space.

The use of BO for experimental design overcomes the problems of over-exploitation and cognitive entrenchment and provides mathematically rigorous guarantees of convergence to the optimal design. However, as noted previously, this often means that domain-specific knowledge and expertise is lost. In this paper, motivated by our observations, rather than attempting to enrich AI models using expert knowledge to accelerate BO, we propose the BO-Muse algorithm that lets the human expert take the lead in experimental design with the aid of an AI "muse" whose job is to augment the expert's intuition through AI suggestions. Thus the AI's role is to provide dynamism to break an expert's cognitive entrenchment and go beyond the state-of-the-art in new problems, while the expert's role is to harness their vast knowledge and extensive experience to produce state-of-the-art designs. Combining these roles in a formal framework is the main contribution of this paper.

BO-Muse is a formal framework that inserts BO into the expert's workflow (see Figure 1), allowing adjustment of the AI exploit/explore strategy in response to the human expert suggestions. This process results in a batch of suggestions from the human expert and the AI at each iteration. This batch of designs is experimentally evaluated and shared with the human and the AI. The AI model is updated and the process iterated until the target is reached. We analyse the sample-efficiency of BO-Muse and provide a sub-linear regret bound. We validate BO-Muse using optimisation benchmarks and teaming with experts to perform complex real-world tasks. Our contributions are:

- Design of a framework (BO-Muse) for a human expert and an AI to work in concert to accelerate experimental design, taking advantage of the human's deeper insight into the problem and the AI's advantage in using rigorous models to complement the expert to achieve sample-efficiency;

- Design of an algorithm that compensates for the human tendency to be overly exploitative by appropriately boosting the AI exploration;

- Provide a sub-linear regret bound for BO-Muse to demonstrate the accelerated convergence due to the human-AI teaming in the optimisation process; and

- Provide experimental validation both using optimisation benchmark functions and with human experts to perform complex real-world tasks.

## 2 BACKGROUND

### 2.1 HUMAN MACHINE PARTNERSHIPS

Mixed initiative creative interfaces propose a tight coupling of human and machine to foster creativity. Thus far, however, research has been largely restricted to game design (Deterding et al., 2017), where the authors identified open challenges including "what kinds of human-AI co-creativity can we envision across and beyond creative practice?". Our work is the first example of the use such a paradigm to accelerate experimental design. Also of importance, though beyond the scope of this study, is the design of interfaces for such systems (Rezwana & Maher, 2022) and how the differing ways human and machine express confidence affects performance (Steyvers et al., 2022).

## 2.2 BAYESIAN OPTIMISATION

Bayesian Optimisation (BO, (Brochu et al., 2010)) is an optimisation method for solving the problems of the form: $\mathbf{x}^\star = \operatorname{argmax}_{\mathbf{x} \in \mathbb{X}} f^\star(\mathbf{x})$ in the least possible number of iterations when $f^\star$ is an expensive blackbox function. Bayesian optimisation models $f^\star$ as a draw from a Gaussian Process $GP(0, K)$ with prior covariance (kernel) $K$ (Rasmussen, 2006) and, at each iteration $t$, recommends the next function evaluation point $\mathbf{x}_t$ by optimising a (cheap) acquisition function $a_t : \mathbb{X} \to \mathbb{R}$ based on the posterior mean and variance given a dataset of observations $\mathbb{D}_{t-1} = \{(\mathbf{x}_i, y_i) : i \in \mathbb{N}_{t-1}\}$:

$$\mu_{t-1}(\mathbf{x}) = \mathbf{y}_{t-1}^{\mathrm{T}} \left(\mathbf{K}_{t-1} + \sigma^2 \mathbf{I}\right)^{-1} \mathbf{k}_{t-1}(\mathbf{x})$$
$$\sigma_{t-1}^2(\mathbf{x}) = K_{t-1}(\mathbf{x}, \mathbf{x}) - \mathbf{k}_{t-1}^{\mathrm{T}}(\mathbf{x}) \left(\mathbf{K}_{t-1} + \sigma^2 \mathbf{I}\right)^{-1} \mathbf{k}_{t-1}(\mathbf{x})$$

where $\mathbf{y}_{t-1} = [y_i]_{i \in \mathbb{N}_{t-1}}$ is the set of observed outputs, $\mathbf{K}_{t-1} = [K(\mathbf{x}_i, \mathbf{x}_j)]_{i,j \in \mathbb{N}_{t-1}}$ and $\mathbf{k}_{t-1}(\mathbf{x}) = [K(\mathbf{x}_i, \mathbf{x})]_{i \in \mathbb{N}_{t-1}}$. Experiments evaluate $y_t = f^\star(\mathbf{x}_t) + \nu_t$, where $\nu_t$ is noise, the GP model is updated to include the new observation, and the process repeats either until a convergence criteria is met or a fixed budget of evaluations is exhausted. Common acquisition functions include Expected Improvement (EI, (Jones et al., 1998)) and GP-UCB (Srinivas et al., 2012). GP-UCB uses:

$$a_t(\mathbf{x}) = \mu_{t-1}(\mathbf{x}) + \beta_t^{1/2} \sigma_{t-1}(\mathbf{x}) \tag{1}$$

Here $\beta_t$ is a variable controlling the trade-off between exploitation of known minima (if $\beta_t$ is small) and exploration (if $\beta_t$ is large). The optimisation performance depends on $\beta_t$. To achieve sub-linear convergence Chowdhury & Gopalan (2017) recommend $\beta_t = \chi_t$, where:

$$\chi_t = \left(\tfrac{\Sigma}{\sqrt{\sigma}} \sqrt{2 \ln\left(\tfrac{1}{\delta}\right) + 1 + \gamma_t} + \|f^\star\|_{\mathcal{H}_K}\right)^2 \tag{2}$$

and $\gamma_t$ is the maximum information gain (maximum mutual information between $f^\star$ and any $t$ observations (Srinivas et al., 2012)). Typically it is assumed that $f^\star \sim GP(0, K)$ or $f^\star \in \mathcal{H}_K$, where $\mathcal{H}_K$ is the reproducing kernel Hilbert space of the kernel $K$ (Srinivas et al., 2012). Otherwise the problem is *mis-specified*. One approach to mis-specified BO is enlarged confidence GP-UCB (EC-GP-UCB, (Bogunovic & Krause, 2021)). In EC-GP-UCB, the function closest to the objective $f^\star$ is denoted $f^\# = \operatorname{argmin}_{f \in \mathcal{H}_K} \|f^\star - f\|_\infty$, and the acquisition function takes the form:

$$a_t(\mathbf{x}) = \mu_{t-1}(\mathbf{x}) + \left(\beta_t^{1/2} + \tfrac{\epsilon \sqrt{t}}{\sigma}\right) \sigma_{t-1}(\mathbf{x}) \tag{3}$$

where $\epsilon = \|f^\# - f^\star\|_\infty$ is the mis-specification gap. The additional term in the acquisition function is required to ensure sub-linear convergence in this case.

## 3 FRAMEWORK

We consider the following optimisation problem $\mathbf{x}^\star = \operatorname{argmax}_{\mathbf{x} \in \mathbb{X}} f^\star(\mathbf{x})$, where $f^\star : \mathbb{X} \to \mathbb{R}$ is expensive and evaluation is noisy, with results $y = f^\star(\mathbf{x}) + \nu$, where $\nu$ is $\Sigma$-sub-Gaussian noise. We optimise $f^\star$ as a series of experimental batches, indexed by $s = 1, 2, \ldots$. For each batch a human expert suggests $\hat{\mathbf{x}}_s$ for evaluation and an AI suggests $\breve{\mathbf{x}}_s$ (**hatted** variables relate to the human expert, **breved** variables to the AI). Both experiments are carried out, the human expert and the AI update their models based on the results, and the process is repeated for a total of $S$ batches.

We do not know precisely how the human expert will model $f^\star$, but the AI maintains a GP model with kernel (prior covariance) $\breve{K}_s$ and we assume the human expert does the same with some unknown kernel $\hat{K}_s$ (we discuss this in detail in section 3.1), where $\breve{K}_s$ and $\hat{K}_s$ may be updated after each batch. The posterior means and variances given dataset $\mathbb{D}_s = \{(\hat{\mathbf{x}}_i, \hat{y}_i), (\breve{\mathbf{x}}_i, \breve{y}_i) : i \leq s\}$ are:

$$\hat{\mu}_s(\mathbf{x}) = \mathbf{y}_s^{\mathrm{T}} (\hat{\mathbf{K}}_s + \sigma^2 \mathbf{I})^{-1} \hat{\mathbf{k}}_s(\mathbf{x}), \quad \hat{\sigma}_s^2(\mathbf{x}) = \hat{K}_s(\mathbf{x}, \mathbf{x}) - \hat{\mathbf{k}}_s^{\mathrm{T}}(\mathbf{x}) (\hat{\mathbf{K}}_s + \sigma^2 \mathbf{I})^{-1} \hat{\mathbf{k}}_s(\mathbf{x})$$
$$\breve{\mu}_s(\mathbf{x}) = \mathbf{y}_s^{\mathrm{T}} (\breve{\mathbf{K}}_s + \sigma^2 \mathbf{I})^{-1} \breve{\mathbf{k}}_s(\mathbf{x}), \quad \breve{\sigma}_s^2(\mathbf{x}) = \breve{K}_s(\mathbf{x}, \mathbf{x}) - \breve{\mathbf{k}}_s^{\mathrm{T}}(\mathbf{x}) (\breve{\mathbf{K}}_s + \sigma^2 \mathbf{I})^{-1} \breve{\mathbf{k}}_s(\mathbf{x})$$

for the human and AI, respectively. We assume $f^\star \in \breve{\mathcal{H}}_s = \mathcal{H}_{\breve{K}_s}$ lies in the RKHS of the AI's kernel $\breve{K}_s$. We do not make this assumption for the human expert, making the problem mis-specified from their perspective. Hence borrowing from (Bogunovic & Krause, 2021), we assume that for batch $s$ the human by default attempts to maximise the closest function to $f^\star$ in $\hat{\mathcal{H}}_s$:

$$\hat{f}_s^\# = \operatorname{argmin}_{f \in \hat{\mathcal{H}}_s : \|f\|_{\hat{\mathcal{H}}_s} \leq \hat{B}} \|f - f^\star\|_\infty \quad (\text{here } \hat{\mathcal{H}}_s = \mathcal{H}_{\hat{K}_s})$$

We further assume that the gap between $f^\star$ and $\hat{f}_s^\#$ is bounded as $\|\hat{f}_{i,s}^\# - f^\star\|_\infty \leq \hat{\epsilon}_s$, and that the human is able to learn (in effect, update their kernel) to "close the gap", so $\lim_{s\to\infty} \hat{\epsilon}_s \to 0$.

The AI generates recommendations using GP-UCB. Motivated by (Borji & Itti, 2013), we assume the human in effect generates recommendations using EC-GP-UCB (Bogunovic & Krause, 2021):

$$\begin{aligned}
\hat{\mathbf{x}}_s &= \mathrm{argmin}_{\mathbf{x}\in\mathbb{X}}\hat{a}_s\left(\mathbf{x}\right) = \hat{\mu}_{s-1}\left(\mathbf{x}\right) + (\hat{\beta}_s^{1/2} + \tfrac{\hat{\epsilon}_s}{\sigma}\sqrt{2s})\hat{\sigma}_{s-1}\left(\mathbf{x}\right) \\
\breve{\mathbf{x}}_s &= \mathrm{argmin}_{\mathbf{x}\in\mathbb{X}}\breve{a}_s\left(\mathbf{x}\right) = \breve{\mu}_{s-1}\left(\mathbf{x}\right) + \breve{\beta}_s^{1/2}\breve{\sigma}_{s-1}\left(\mathbf{x}\right)
\end{aligned} \quad (4)$$

The form of $\hat{a}_s$ captures the broad behaviour of human recommendation selection, balancing exploitation of known good regions, which corresponds to $\hat{\beta}_s \to 0$ (e.g. "based on my model, this experiment should yield good results"), and exploration of areas of uncertainty, which corresponds to $\hat{\beta}_s \to \infty$ (e.g. "explore here, I'm curious"). We do not presume to know the precise trade-off $\hat{\beta}_s$ used by the human but, based on (Borji & Itti, 2013; Dane, 2010), it appears probable that the human will pursue a conservative policy, so $\hat{\beta}_s$ may be small. We therefore use the AI trade-off $\breve{\beta}_s$, which we control, to compensate for the potential conservative tendencies of the expert.

It is convenient to specify the exploration/exploitation trade-off parameters $\hat{\beta}_s$ and $\breve{\beta}_s$ relative to (2) (Chowdhury & Gopalan, 2017; Bogunovic & Krause, 2021). Without loss of generality we require that $\hat{\beta}_s \in (\hat{\zeta}_{s\downarrow}\hat{\chi}_s, \hat{\zeta}_{s\uparrow}\hat{\chi}_s)$ and $\breve{\beta}_s = \breve{\zeta}_s\breve{\chi}_s$, where $0 \leq \hat{\zeta}_{s\downarrow} \leq 1 \leq \hat{\zeta}_{s\uparrow} \leq \infty$, $\breve{\zeta}_s \geq 1$ and (2):

$$\hat{\chi}_s = \left(\tfrac{\Sigma}{\sqrt{\sigma}}\sqrt{2\ln\left(1/\delta\right) + 1 + \hat{\gamma}_s} + \|\hat{f}_s^\#\|_{\hat{\mathcal{H}}_s}\right)^2, \; \breve{\chi}_s = \left(\tfrac{\Sigma}{\sqrt{\sigma}}\sqrt{2\ln\left(1/\delta\right) + 1 + \breve{\gamma}_s} + \|f^\star\|_{\breve{\mathcal{H}}_s}\right)^2 \;\; (5)$$

where $\hat{\gamma}_s$ and $\breve{\gamma}_s$ are, respectively, the max information-gain for human expert and the AI. We show in section 3.3 that the this suffices to ensure sublinear convergence for any human expert selections, and moreover that the convergence rate can be improved beyond what can be achieved by standard GP-UCB if the human operates as described here.

## 3.1 THE HUMAN MODEL

We posit that the human expert will maintain, explicitly or implicitly, an *evolving* weight-space model (**we do not presume to know the details of this, only that it exists, explicitly or otherwise**):

$$\hat{f}_s\left(\mathbf{x}\right) = \hat{g}_s\left(\hat{\mathbf{p}}_s\left(\mathbf{x}\right)\right)$$

of $f^\star$, where $\hat{g}_s$ is in some sense "simple" and $\hat{\mathbf{p}}_s : \mathbb{R}^n \to \mathbb{R}^{\hat{m}_s}$ represents the human's understanding of important features. This fits into the above scheme if we let the form of $\hat{g}_s$ dictate the kernel $\hat{K}_s$. For example if we know that $\hat{g}_s$ is linear (so $\hat{f}_s(\mathbf{x}) = \hat{\mathbf{w}}_s^\mathrm{T}\hat{\mathbf{p}}_s(\mathbf{x})$ in weight-space) then we can say that the human is, in effect, using a GP model with a linear-derived model:

$$\hat{K}_s\left(\mathbf{x}, \mathbf{x}'\right) = \hat{\mathbf{p}}_s^\mathrm{T}\left(\mathbf{x}\right)\hat{\mathbf{p}}_s\left(\mathbf{x}'\right)$$

Similarly if the human is using a model $\hat{g}_{i,s}$ captured by a $d^\mathrm{th}$-order approximation model $\hat{f}_s(\mathbf{x}) = \hat{\mathbf{w}}_s^\mathrm{T}[\hat{\mathbf{p}}_s^{\otimes q}(\mathbf{x})]_{q=0,1,\dots,d}$ then we can assume a GP with a polynomial-derived kernel:

$$\hat{K}_s\left(\mathbf{x}, \mathbf{x}'\right) = \left(1 + \hat{\mathbf{p}}_s^\mathrm{T}\left(\mathbf{x}\right)\hat{\mathbf{p}}_s\left(\mathbf{x}'\right)\right)^d$$

As discussed previously we do not assume that the human expert's evolving model suffices to completely replicate $f^\star$, particularly in the earlier stages of the algorithm, but we do assume that the human expert is capable of learning from observations of $f^\star$ to refine their model, closing the gap $\hat{\epsilon}_{2s}$ between their model and $f^\star$ so that $\lim_{s\to\infty} \hat{\epsilon}_{2s} = 0$.

With regard to max information gain, we can reasonably assume that the human expert (the expertise is important) starts with a better understanding of $f^\star$ than the AI - the underlying physics of the system, the behaviour one might expect in similar experiments, etc. So the human expert may begin with an incomplete but informative set of features that relate to their knowledge of the system or similar systems similar, or an understanding of the covariance structure of design space, so:

- The prior variance of the human expert's GP will vary between a zero-knowledge base level in regions that are a mystery to the expert, and much lower in regions where the expert has a good understanding from past experience, understanding of underlying physics etc.

- The prior covariance of the expert's GP (the expert's kernel) will have a structure informed by the expert's understanding and knowledge, for example, of the "region A will behave like region B because they have feature/attribute C in common" type, so an experiment in region A will reduce the human expert's posterior variance both in regions.

By comparison, the AI will typically start with a generic kernel prior like an SE kernel, Matern kernel or similar. Such priors are "flat" over the design space, with no areas of lower prior variance and no "region A will behave like B" behaviour, so an experiment will only reduce the AI's local variance.[1] Thus as the algorithm progresses the human expert's variance will both start from a lower prior and decrease more quickly than the AI's. We know that max information gain is bounded as the sum of the logs of the pre-experiment posterior variance (Srinivas et al., 2012, Lemma 5.3), so we may reasonably assume that the expert's max information gain will be lower than the AI's. Finally, we argue that, as the human expert's kernel is built on relatively few, highly informative features, the expert's RKHS norm $\|f_s^{\#}\|_{\hat{K}_s} = \|\hat{\mathbf{w}}\|_2$ will be less than the AI's norm $\|f^{\star}\|_{\check{K}_s} = \|\check{\mathbf{w}}\|_2$.

### 3.2 THE BO-MUSE ALGORITHM

The BO-Muse algorithm is shown in algorithm 1, where $f^{\star}$ is optimised using a sequence of batches $s = 1, 2, \ldots, S$, each containing one human and one AI recommendation, respectively $\hat{\mathbf{x}}_s$ and $\check{\mathbf{x}}_s$. The AI recommendation minimises the GP-UCB acquisition function on the AI's GP posterior, with the exploration/exploitation trade-off $\check{\beta}_s$ given (see section 3.4). The human recommendation is assumed to be *implicitly* selected to minimise the EC-GP-UCB acquisition function (3), where the exploitation/exploration trade-off $\hat{\beta}_s$ is unknown but assumed to lie in the range $\hat{\beta}_s \in (\hat{\zeta}_{s\downarrow}\hat{\chi}_s, \hat{\zeta}_{s\uparrow}\hat{\chi}_s)$, and the gap $\hat{\epsilon}_s$ is unknown but assume to converge to 0 at the rate to be specified in theorem 1).

We use two approximations in our definition of $\check{\beta}_s$. Following the standard practice, we approximate the max information gain $\check{\gamma}_s$ as $\sum_{(\mathbf{x},\ldots)\in\mathbb{D}_{s-1}} \ln(1 + \sigma^{-2}\check{\sigma}_i^2(\mathbf{x}))$; and we approximate the RKHS norm bound $\check{B} = \|f^{\star}\|_{\mathcal{H}_s}$ as $\|\check{\mu}_s\|_{\mathcal{H}_s}$, which is updates using max to ensure it is non-decreasing with $s$ (this will tend to over-estimate the norm, but this should not affect the algorithm's rate of convergence). For simplicity we assume that $\sigma = \Sigma$ (the model parameter matches the noise).

---

**Algorithm 1** BO-Muse

> **Input:** Initial observations $\mathbb{D}_0 = \{(\mathbf{x}_1, y_1), (\mathbf{x}_2, y_2), \ldots\}$, AI prior $\mathrm{GP}(0, K)$. Let $\check{B} = 1$.
> **for** $s \in 1, 2, \ldots, S$ **do**
> > Set $\check{\beta}_s = 7\left(\sqrt{\sigma}\sqrt{2\ln(1/\delta) + 1 + \sum_{(\mathbf{x},\ldots)\in\mathbb{D}_{s-1}}\ln\left(1 + \sigma^{-2}\check{\sigma}_i^2(\mathbf{x})\right)} + \check{B}\right)^2$ as per (5), (8).
> > AI recommends $\check{\mathbf{x}}_s$ as $\check{\mathbf{x}}_s = \mathrm{argmax}_{\mathbf{x}\in\mathbb{X}}\check{a}_s(\mathbf{x}) = \check{\mu}_{s-1}(\mathbf{x}) + \check{\beta}_s^{1/2}\check{\sigma}_{s-1}(\mathbf{x})$.
> > Human recommends $\hat{\mathbf{x}}_s$.
> > Run experiments to obtain $\hat{y}_s = f^{\star}(\hat{\mathbf{x}}_s) + \hat{\nu}_s$ and $\check{y}_s = f^{\star}(\check{\mathbf{x}}_s) + \check{\nu}_s$.
> > Set $\mathbb{D}_s = \mathbb{D}_{s-1} \cup \{(\hat{\mathbf{x}}_s, \hat{y}_s), (\check{\mathbf{x}}_s, \check{y}_s)\}$ and update the AI GP posterior.
> > Set $\check{B} = \max\{\check{B}, \mathbf{y}_s^{\mathrm{T}}\left(\mathbf{K}_s + \sigma^2\mathbf{I}\right)^{-1}\mathbf{y}_s\}$.
> **end for**

---

### 3.3 CONVERGENCE AND REGRET BOUNDS

We now discuss the convergence properties of BO-Muse. Our goal here is twofold: first we show that BO-Muse converges even in the worst-case where the human operates arbitrarily; and second, assuming the human behaves according to our assumptions, we analyse how convergence is accelerated. Our approach is based on regret analysis (Srinivas et al., 2012; Chowdhury & Gopalan, 2017; Bogunovic & Krause, 2021). As experiments arebatched we are concerned with instantaneous regret per batch, not per experiment. The instantaneous regret for batch $s$ is:

$$r_s = \min\{f^{\star}(\mathbf{x}^{\star}) - f^{\star}(\hat{\mathbf{x}}_s), f^{\star}(\mathbf{x}^{\star}) - f^{\star}(\check{\mathbf{x}}_s)\}$$

and the cumulative regret up to and including batch $S$ is $R_S = \sum_{s=1}^{S} r_s$. If $R_S$ grows sub-linearly then the minimum instantaneous regret will converge to 0 as $S \to \infty$.

---

[1]We do not consider the possibility of transfer learning of this structure, as this requires the expert to distill their knowledge in an amenable form which, as noted previously, is highly non-trivial.

Consider first the worst-case scenario, i.e. an arbitrary, non-expert human. In this case the BO-Muse algorithm 1 effectively involves the AI using a GP-UCB acquisition function (the constant scaling factor on $\breve{\beta}_s$ does not change the convergence properties of GP-UCB) to design a sequence of experiments $\breve{\mathbf{x}}_s$, where each experiment costs twice as much as usual to evaluate and yields two observations, $(\breve{\mathbf{x}}_s, \breve{y}_s = f^\star(\breve{\mathbf{x}}_s) + \breve{\eta}_s)$ and $(\hat{\mathbf{x}}_s, \hat{y}_s = f^\star(\hat{\mathbf{x}}_s) + \hat{\eta}_s)$, where $\hat{\mathbf{x}}_s$ is arbitrary. Additional observations can only improve the accuracy of the posterior, so using standard methods (e.g. Chowdhury & Gopalan (2017); Bogunovic & Krause (2021); Srinivas et al. (2012)), we see that $R_S = \mathcal{O}(B\sqrt{\breve{\gamma}_{2S}2S} + \sqrt{(\ln(1/\delta) + \breve{\gamma}_{2S})\breve{\gamma}_{2S}2S})$, which is sub-linear for well-behaved kernels.

Thus we see that the worst-case convergence of BO-Muse is the same, in the big-O sense, as that of GP-UCB. However this is pessimistic: in reality we assume a human is generating experiments $\hat{\mathbf{x}}_s$ using an EC-GP-UCB style trade-off between exploitation and exploration, and moreover that the human is an expert with an implicit or explicit evolving model of the system that is superior to the AI's generic prior. For this case we have the following result:

**Theorem 1.** *Fix $\delta > 0$, $\theta_\downarrow > 0$, $\theta_s \in [\theta_\downarrow, \infty]$, $S \in \mathbb{N}_+$. Assume $\hat{\zeta}_{s\downarrow} \in (0, 1]$, $\breve{\zeta}_s = \breve{\zeta} \geq 1$ so:*

$$M_{\theta_s}\big(\exp(\hat{\chi}_s^{1/2}(1 - \hat{\zeta}_{s\downarrow}^{1/2})_+ \sigma_{s-1\uparrow}), \exp(\breve{\chi}_s^{1/2}(1 - \breve{\zeta}^{1/2})_- \sigma_{s-1\downarrow})\big) \leq 1 \tag{6}$$

*for all $s \leq S$, where $\sigma_{s-1\uparrow} = \max(\hat{\sigma}_{s-1}(\mathbf{x}^\star), \breve{\sigma}_{s-1}(\mathbf{x}^\star))$, $\sigma_{s-1\downarrow} = \min(\hat{\sigma}_{s-1}(\mathbf{x}^\star), \breve{\sigma}_{s-1}(\mathbf{x}^\star))$, $M_\theta(a, b) = (\frac{1}{2}(a^\theta + b^\theta))^{1/\theta}$ is the generalised mean, and $\hat{\chi}_s, \breve{\chi}_s$ as per (5). If $\hat{\epsilon}_{2s} = \mathcal{O}(s^{-\frac{1}{2}})$ then:*

$$\frac{1}{S}R_S \leq \ln\Big(M_{-\theta_\downarrow}\Big(\exp\Big(\sqrt{\hat{C}\frac{\hat{\chi}_s\hat{\gamma}_s}{S}} + \mathcal{O}\big(\frac{1}{S}\big)\Big), \exp\Big(\sqrt{\breve{C}\big(\frac{1}{2}(1 + \breve{\zeta}^{1/2})\big)^2 \frac{\breve{\chi}_s\breve{\gamma}_s}{S}}\Big)\Big)\Big) \tag{7}$$

*with probability $\geq 1 - \delta$, where $\hat{C}, \breve{C} > 0$ are constants.*

We present the proof of this theorem in the appendix. As discussed previously $\hat{\gamma}_S$ and $\breve{\gamma}_S$ are the max information gains for the human and AI, respectively; $\hat{\zeta}_{s\downarrow} \in (0, 1]$ defines the degree to which the human is allowed to over-exploit; and $\breve{\zeta}$ defines the degree to which the AI must over-explore to compensate. The parameters $\theta_s$ constrain the range of $\breve{\zeta}$ for which the theorem is applicable through the condition (6), and the "mix" between human and AI in the regret bound (7). Assuming the max information gain for the human model has better convergence properties than the AI model, which we consider reasonable based on previous discussion, and noting that the power mean $M_\theta(a, b)$ will be dominated by $\max(a, b)$ as $\theta \to \infty$ and by $\min(a, b)$ as $\theta \to -\infty$, we see that:

- When $\theta_s$ is large the AI trade-off parameter $\breve{\zeta}$ must be large to satisfy (6), corresponding to a very explorative AI, and the regret bound (7) will be *asymptotically* superior, being dominated by the human's max information gain $\breve{\gamma}_S$ for large $S$ which, as discussed in section 3.1, may be reasonably expected to be smaller than the AI's max information gain due to the human expert's superior understanding of $f^\star$.

- When $\theta_s$ is small the AI is less constrained, so the theorem is applicable for more expoitative AIs, but the regret bound (7) will be more strongly influenced by the AI max information gain and hence asymptotically inferior.

Note that, while larger $\theta_s$ will lead to tighter regret bounds in the asymptotic regime, this bound will only apply for very explorative AIs, and moreover the factor $(\frac{1}{2}(1 + \breve{\zeta}^{1/2}))^2$ will also be large, so the regret bound may only be superior for very large $S$. The "ideal" trade-off is unclear, but the important observation from this theorem is that the human's expertise, and subsequent superior maximum information gain, will improve the regret bound and thus convergence.

## 3.4 TUNING PARAMETER SELECTION

We now consider the selection of the tuning parameter $\breve{\beta}_s$ to ensure faster convergence than standard BO alone. The conditions that $\breve{\beta}_s$ must meet are specified by (6). Equivalently, taking the maximally pessimistic view of human under-exploration (i.e. $\hat{\zeta}_{s\downarrow} = 0$), we require that $\exp(\theta_s\sqrt{\breve{\chi}_s}(1 - \breve{\zeta}_s^{1/2})\sigma_{s-1\downarrow}) \leq 2 - \exp(\theta_s\sqrt{\breve{\chi}_s}\sigma_{s-1\uparrow})$ or, equivalently, $\breve{\zeta}_s \geq \big(1 + (\frac{\hat{\chi}_s}{\breve{\chi}_s})^{1/2}\frac{\sigma_{s-1\uparrow}}{\sigma_{s-1\downarrow}}\frac{1}{\phi_s}\ln\big(\frac{1}{2-e^{\phi_s}}\big)\big)^2$, where[2] $\phi_s = \theta_s\sqrt{\breve{\chi}_s}\sigma_{s-1\uparrow} \in (0, \ln 2)$, and, noting that the right of the inequality is strictly increasing, $\phi_s = 0$ corresponds to $\breve{\zeta}_s = 1 + (\frac{\hat{\chi}_s}{\breve{\chi}_s})^{1/2}\frac{\sigma_{s-1\uparrow}}{\sigma_{s-1\downarrow}}$ and $\phi_s = \ln 2$ to $\breve{\zeta}_s = \infty$.

---

[2]We need $\phi_s > 0$ to ensure $\theta_s > 0$, and note that $h(\phi) = \frac{1}{\phi}\ln(\frac{1}{2-e^\phi})$ is increasing and $h(\ln 2) = \infty$.

Table 1: Synthetic benchmark functions. Analytical forms are provided in the second column and the last column depicts the high level features used by a simulated human expert.

| Functions | $f^\star(\mathbf{x})$ | High Level Features |
|---|---|---|
| Matyas-2D | $0.26 * (x_1^2 + x_2^2) - 0.48 * (x_1 * x_2)$ | $x_1' = x_1^2, x_2' = x_2^2$ 
 $x_3' = x_1 * x_2$ |
| Ackley-4D | $-ae^{-b\sqrt{\frac{1}{d}\|x\|_2^2}} - e^{\sqrt{\frac{1}{d}\sum_i \cos(cx_i)}} + a + e^1$ | $x_1' = \cos(x_1), x_2' = \cos(x_2)$ 
 $x_3' = \cos(x_3), x_4' = \cos(x_4), x_5' = \|\mathbf{x}\|_2$ |
| Levy-6D | $\sin^2(\pi w_1) + s + (w_d - 1)^2[1 + \sin^2(2\pi w_d)]$ 
 where $s = \sum_i^{d-1}(w_1 - 1)^2[1 + 10\sin^2(\pi w_i + 1)]$ 
 and $w_i = 1 + \frac{\mathbf{x}_i - 1}{4} \ \forall i \in \mathbb{N}_6$ | $x_1' = (\sin x_1)^2,$ 
 $x_{j+1}' = x_j^2(\sin x_j)^2$ 
 $\forall j \in \mathbb{N}_6$ |

As discussed in section 3.1, it is reasonable to assume that the max information gains satisfy $\breve{\gamma}_s \geq \hat{\gamma}_s$, and that $\|f_s^{\#}\|_{\hat{K}_s} \leq \|f^\star\|_{\breve{K}_s}$. With these assumptions $\sqrt{\hat{\chi}_s} \leq \sqrt{\breve{\chi}_s}$. Furthermore we prove in the Appendix that $\lim_{s \to \infty} \frac{\sigma_{s-1\uparrow}}{\sigma_{s-1\downarrow}} = 1$, so we approximate the bound on $\breve{\zeta}_s$ as $\breve{\zeta}_s \geq (1 + \frac{1}{\phi_s}\ln(\frac{1}{2 - e^{\phi_s}}))^2$. Recalling $\phi_s \in (\ln 1, \ln 2)$ we finally, somewhat arbitrarily select $\phi_s = \ln 3/2$ (the middle of the range in the log domain), which leads to the following heuristic used in the BO-Muse algorithm:

$$\breve{\zeta}_s \geq \left(1 + \frac{\ln 2}{\ln 3/2}\right)^2 \approx 7 \tag{8}$$

## 4 EXPERIMENTS

We validate the performance of our proposed BO-Muse algorithm in the optimisation of synthetic benchmark functions, and the real-world tasks involving human experts. In all our experiments, we have used Squared Exponential (SE) kernel at all levels with associated hyper-parameters tuned using maximum-likelihood estimation. We measure the sample-efficiency of BO-Muse framework and other standard baselines in terms of the *simple regret* ($r_t$): $r_t = f^\star(\mathbf{x}^\star) - \max_{\mathbf{x}_t \in \mathcal{D}_t} f^\star(\mathbf{x}_t)$, where $f^\star(\mathbf{x}^\star)$ is the true global optima and $f^\star(\mathbf{x}_t)$ is the best solution observed in $t$ iterations. Experiments were run on an Intel Xeon CPU@ 3.60GHz workstation with 16 GB RAM capacity.

### 4.1 EXPERIMENTS WITH OPTIMISATION BENCHMARK FUNCTIONS

We have evaluated BO-Muse on synthetic test functions covering a range of dimensions, as detailed in Table 1. We compare the sample-efficiency of BO-Muse with **(i) Generic BO:** A standard GP-UCB based BO algorithm with the exploration-exploitation trade-off factor ($\beta$) set as per Srinivas et al. (2012); **(ii) Simulated Human:** A simulated human with access to higher level properties (refer to Section 3.1 and high level features in Table 1) that may help to model the optimisation function more accurately; and **(iii) Simulated Human + PE**: A simulated human teamed with an AI agent using a pure exploration strategy (that is, an AI policy with $\breve{\zeta}_s = \infty$). To simulate a human expert (with high exploitation), we use a standard BO algorithm with small exploration factor maximising $\hat{a}_s(\mathbf{x}) = \hat{\mu}_s(\mathbf{x}) + 0.001\hat{\sigma}_s(\mathbf{x})$. Furthermore, we have ensured to allocate the same function evaluation budget for all the competing methods.

Figure 2a and 2b shows the simple regret computed for the Ackley-4D and Levy-6D functions averaged over 10 randomly initialised runs – BO-Muse consistently outperforms all the baselines for both functions. We believe that the poor performance of Simulated Human + PE is due to the pure exploration strategy used. Additionally, we have conducted an ablation study by varying the degrees of exploitation-exploration strategy ($\hat{\beta}_s$) to simulate covering over-exploitative to over-explorative experts. Figure 2c shows the result of the ablation study for Matyas-2D function and we refer to the appendix for the additional results. To further understand the behaviour of BO-Muse framework with other human acquisition functions, we have also considered an additional function optimisation experiment to include Expected Improvement acquisition function for human experts. The experimental details and results of the ablation study are provided in the Appendix.

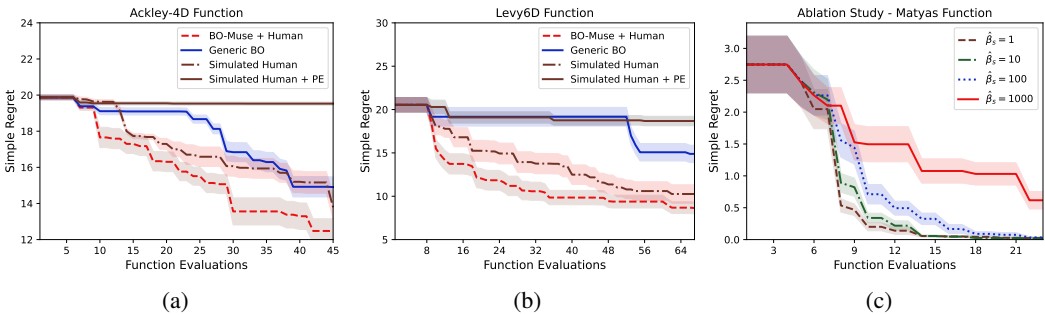

Figure 2: Simple regret versus iterations for (a) Ackley-4D (b) Levy-6D functions. Ablation study with varying degrees of exploitation-exploration parameter ($\hat{\beta}_s$) is shown in (c).

## 4.2 Real-world Experiments

We now evaluate the performance of our proposed framework in complex real-world tasks.

### 4.2.1 Classification Tasks – Support Vector Machines and Random Forests

**Experimental Set-up.** The expert task is to choose hyper-parameters for Support Vector Machine (SVM) and Random Forest (RF) classifiers operating on real-world *Biodeg* dataset from UCI repository (Dua & Graff, 2017). We divide the dataset into random 80/20 train/test splits. We set up two human expert teams. Each member of Team 1 works in partnership with BO-Muse, whilst members of Team 2 work individually without BO-Muse. We recruited 8 participants[3] consisting of 4 postdocs and 4 postgraduate students. 2 postdocs and 2 students are allocated to each team randomly so that each team has 4 participants, with roughly similar expertise. Each participant is given the same budget, 3 random initial designs + 30 further iterations. At the end of each iteration, the test classification error for the suggested hyper-parameter set is computed. We measure the overall performance using simple regret, which will be the minimum test classification error observed so far. The individual results from each team are averaged to compare (Team 1) BO-Muse + Human vs (Team 2) Human alone (baseline). Additionally, we also report the performance of Generic BO (AI alone) method to demonstrate the efficacy of our approach. Further, we have set the same seed initialisation and allocated the same evaluation budget for all the algorithms. As we provide each participant the same set of random initial observations, we do not compute the descriptive statistics for the Generic BO baseline. We note that BO-Muse is designed to work with experts at various levels as we assume an imperfect expert model via a mis-specified GP.

**Interfacing with Experts.** The human experts perform two hyper-parameter tuning experiments to minimise the test classification error of SVM and RF. The expert is provided with a simple graphical interface (Figure 3a) which shows accumulated observations of classifier performance as a function of hyper-parameters, with the best result so far shown in dark blue. Interfaces for both teams are similar excepting that teams working with BO-Muse also see the previous AI-generated observations indicated by ▼. Experts suggest the next hyper-parameter set by clicking at a point of their choice inside the plot. We refer to Appendix A.9 for more details.

**Experiment 1 – SVM Classification.** In this experiment we have considered C-SVM classifiers with Radial Basis Function (RBF) kernel. We have used LibSVM (Chang & Lin, 2011) implementation of SVM with hyper-parameters kernel scale $\gamma$ and the cost parameter $C$. The SVM hyper-parameters *i.e.,* $\gamma$ and $C$ are tuned in the exponent space of $[-3, 3]$.

**Experiment 2 – Random Forest Classification.** In the classification tasks with random forests we let the experts and BO-Muse tune the hyper-parameters *maximum depth of the decision tree* and the *number of samples per split* in the range $(0, 100]$, and $(1, 50]$, respectively.

---

[3]Necessary ethics approval obtained.

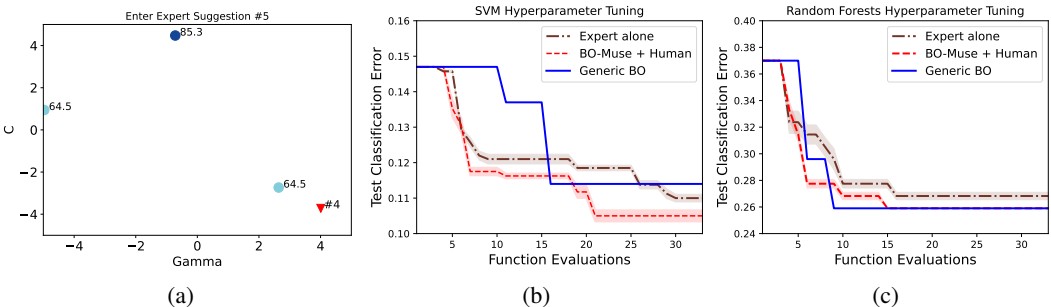

Figure 3: (a) Interface used by participants showing performance vs hyper-parameters: best performances indicated in blue (darker is higher). If working with BO-Muse, machine point from the previous iteration (#) are shown with ▼; Simple regret vs iterations for the hyper-parameter tuning of classifiers comparing Team 1 (BO-Muse + Human) (red) vs Team 2 (Human only) (black) vs AI alone (blue). We report the simple regret mean (along with its standard deviation) for (b) Support Vector Machines (SVM) and (c) Random Forests (RF) classifiers.

The results of our classification experiments are depicted in Figure 3b and Figure 3c. In both experiments, the BO-Muse + Human team outperforms the experts working on their own.

### 4.2.2 SUPPLEMENTARY SPACE SHIELD DESIGN EXPERIMENT

Our third experiment evaluates BO-Muse in a real applied engineering experiment. We consider the design of a shield for protecting spacecraft against the impact of a space debris particle by partnering with a world leading impact expert. For this problem, i.e., impact of a cubic steel debris particle, *there exists no state-of-the-art solution*. We include this experiment in Appendix A.10 because of the lack of familiarity of CS with this problem and the detail required to give sufficient background.

In the experiment (see Table 6 in Appendix A.10) we observe the human expert initially exploring solutions based on the state-of-the-art for more typical debris impact problems (which are normally simplified to spherical aluminium projectiles). The expert is observed to rapidly exploit their initial 3 designs to identify two feasible shielding solutions within the first four batch iterations (ID 4–11, marked in blue). The expert performs further exploitation of these successful designs (Result=0) over the next four batch iterations (ID 12-19, marked in brown) in an attempt to reduce the weight, but is unsuccessful. To this point the expert does not appear to have been influenced at all by the BO suggestions. However, in the next four batch iterations (ID 20-27, marked in green) we can observe the expert taking inspiration from the previous BO-Muse suggestions. One such exploitation results in a successful solution (ID 27), further exploitation of which provides the best solution identified by the experiment ID 29. This solution is highly irregular for spacecraft debris shields, utilising a polymer outer layer in contact with a metallic backing to disrupt the debris particle. Such a design is not reminiscent of any established flight hardware, see e.g., Christiansen et al. (2009). Thus, the BO-Muse is demonstrated to have performed its role as hypothesised, inspiring the human expert with novel designs that are subsequently subject to exploitation by the human expert. The experimental set-up and results are discussed in detail in Appendix A.10.

## 5 CONCLUSION

We have presented a new approach to human expert/AI teaming for experimental optimisation. Our algorithm lets the human expert take the lead in the experimental process thus allowing them to fully use their domain expertise, while the AI plays the role of a muse, injecting novelty and searching for regions the human may have overlooked to break the human out of over-exploitation induced by cognitive entrenchment. We show that our algorithm converges sub-linearly and faster than either the AI or human expert alone.

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

# A  APPENDIX

## A.1  PROOFS OF REGRET BOUNDS

In this appendix we consider a generalised version of the framework. For clarity (we favour brevity in the paper body, but clarity is essential here) we also require some additional notations. As in the main paper, our goal is to solve:

$$\mathbf{x}^\star = \underset{\mathbf{x} \in \mathbb{X}}{\operatorname{argmax}} f^\star (\mathbf{x})$$

where $f^\star$ is only measurable via an expensive and noisy process:

$$y = f^\star (\mathbf{x}) + \nu$$

where $\nu$ is $\Sigma$-sub-Gaussian noise.

Let's assume that we have a series of experimental batches $1, 2, \ldots$, where batch $s$ contains $\hat{p}$ human generated experiments and $\breve{p}$ AI generated experiments, giving a total of $p = \hat{p} + \breve{p}$ experiments. We use a hat with an index $i \in \hat{\mathbb{M}}$ to indicate a property relating to human $i$, and a breve with an index $j \in \breve{\mathbb{M}}$ to indicate a property relating to AI $j$, where:

$$\hat{\mathbb{M}} = \mathbb{N}_{\hat{p}} \quad \text{(set of humans)}$$
$$\breve{\mathbb{M}} = \mathbb{N}_{\breve{p}} \quad \text{(set of AIs)}$$

We assume the batches are run sequentially, and wlog that the experiments within each batch are nominally ordered, so the set of all experiments may be indexed with $t$. We use a bar to differentiate between a property indexed by experiment number $t$, which is unbarred (e.g. $c_t$), and a property indexed by batch number $s$, which is barred (e.g. $\bar{d}_s$). For simplicity we define:

$$
\begin{aligned}
s_t &= \left\lceil \tfrac{t}{p} \right\rceil && \text{(batch $s$ in which experiment $t$ occurs)} \\
\underline{t}_s &= p(s-1) + 1 && \text{(first experiment in batch $s$)} \\
\underline{t} &= p(s_t - 1) + 1 && \text{(first experiment in batch $s_t$)} \\
\underline{\hat{t}}_s &= p(s-1) + 1 && \text{(first human experiment in the batch $s$)} \\
\underline{\hat{t}} &= p(s_t - 1) + 1 && \text{(first human experiment in the batch $s_t$)} \\
\underline{\breve{t}}_s &= p(s-1) + 1 + \hat{p} && \text{(first AI experiment in the batch $s$)} \\
\underline{\breve{t}} &= p(s_t - 1) + 1 + \hat{p} && \text{(first AI experiment in the batch $s_t$)} \\
\mathbb{T}_s &= \mathbb{N}_p + p(s-1) + 1 && \text{(set of experiments in batch $s$)} \\
\mathbb{T}_{\leq s} &= \mathbb{N}_{sp} + 1 && \text{(set of experiments up to and including batch $s$)}
\end{aligned}
$$

We assume that humans and AIs maintain a GP model that is updated after each batch. So, after batch $s$, the posterior means and variances are, respectively:

$$\hat{\bar{\mu}}_{i,s}(\mathbf{x}) = \hat{\bar{\boldsymbol{\alpha}}}_{i,s}^{\mathrm{T}} \hat{\bar{\mathbf{k}}}_{i,s}(\mathbf{x}), \quad \hat{\bar{\sigma}}_{i,s}^2(\mathbf{x}) = \hat{\bar{K}}_{i,s}(\mathbf{x},\mathbf{x}) - \hat{\bar{\mathbf{k}}}_{i,s}^{\mathrm{T}}(\mathbf{x}) \left( \hat{\bar{\mathbf{K}}}_{i,s} + \hat{\sigma}^2 \mathbf{I} \right)^{-1} \hat{\bar{\mathbf{k}}}_{i,s}(\mathbf{x})$$

$$\breve{\bar{\mu}}_{j,s}(\mathbf{x}) = \breve{\bar{\boldsymbol{\alpha}}}_{j,s}^{\mathrm{T}} \breve{\bar{\mathbf{k}}}_{j,s}(\mathbf{x}), \quad \breve{\bar{\sigma}}_{j,s}^2(\mathbf{x}) = \breve{\bar{K}}_{j,s}(\mathbf{x},\mathbf{x}) - \breve{\bar{\mathbf{k}}}_{j,s}^{\mathrm{T}}(\mathbf{x}) \left( \breve{\bar{\mathbf{K}}}_{j,s} + \breve{\sigma}^2 \mathbf{I} \right)^{-1} \breve{\bar{\mathbf{k}}}_{j,s}(\mathbf{x})$$

where:

$$
\begin{aligned}
\hat{\bar{\boldsymbol{\alpha}}}_{i,s} &= \left( \hat{\bar{\mathbf{K}}}_{i,s} + \hat{\sigma}^2 \mathbf{I} \right)^{-1} \bar{\mathbf{y}}_s, & \breve{\bar{\boldsymbol{\alpha}}}_{j,s} &= \left( \breve{\bar{\mathbf{K}}}_{j,s} + \breve{\sigma}^2 \mathbf{I} \right)^{-1} \bar{\mathbf{y}}_s \\
\hat{\bar{\mathbf{K}}}_{i,s} &= \left[ \hat{\bar{K}}_{i,s}(\mathbf{x}_t, \mathbf{x}_{t'}) \right]_{t,t' \in \mathbb{T}_{\leq s}}, & \breve{\bar{\mathbf{K}}}_{j,s} &= \left[ \breve{\bar{K}}_{j,s}(\mathbf{x}_t, \mathbf{x}_{t'}) \right]_{t,t' \in \mathbb{T}_{\leq s}} \\
\hat{\bar{\mathbf{k}}}_{i,s}(\mathbf{x}) &= \left[ \hat{\bar{K}}_{i,s}(\mathbf{x}_t, \mathbf{x}) \right]_{t \in \mathbb{T}_{\leq s}}, & \breve{\bar{\mathbf{k}}}_{j,s}(\mathbf{x}) &= \left[ \breve{\bar{K}}_{j,s}(\mathbf{x}_t, \mathbf{x}) \right]_{t \in \mathbb{T}_{\leq s}} \\
\bar{\mathbf{y}}_s &= \left[ f^\star(\mathbf{x}_t) + \nu_t \right]_{t \in \mathbb{T}_{\leq s}}, & \bar{\mathbf{f}}_s &= \left[ f^\star(\mathbf{x}_t) \right]_{t \in \mathbb{T}_{\leq s}}
\end{aligned}
$$

We also occasionally use:

$$\mathbf{y}_t = \left[ f^\star(\mathbf{x}_{t'}) + \nu_{t'} \right]_{t' \leq t}, \quad \mathbf{f}_t = \left[ f^\star(\mathbf{x}_{t'}) \right]_{t' \leq t}$$

As this is a batch algorithm we are concerned with the instantaneous regret for the batches, not the individual experiments therein. The instantaneous regret for batch $s$ is:

$$\bar{r}_s = \min_{t \in \mathbb{T}_s} r_t, \quad r_t = f^\star(\mathbf{x}^\star) - f^\star(\mathbf{x}_t)$$

where $r_t$ is the instantaneous regret for experiment $t$. The cumulative regret up to and including batch $S$ is:

$$\bar{R}_S = \sum_{s \in \mathbb{N}_S + 1} \bar{r}_s$$

We do not assume $f^\star$ is drawn from any of the GP models for humans or AIs, so the problem mis-specified. So, borrowing from Bogunovic & Krause (2021), we assume humans and AIs attempt to maximise the "closest" (best-in-class) function to $f^\star$ in the respective hypothesis spaces, using the shorthand $\hat{\mathcal{H}}_{i,s} = \mathcal{H}_{\hat{K}_{i,s}}$, $\breve{\mathcal{H}}_{j,s} = \mathcal{H}_{\breve{K}_{j,s}}$:

$$\hat{\mathbb{H}}_{i,s} = \left\{ f \in \hat{\mathcal{H}}_{i,s} \middle| \|f\|_{\hat{\mathcal{H}}_{i,s}} \le \hat{B}_i \right\}$$
$$\breve{\mathbb{H}}_{j,s} = \left\{ f \in \breve{\mathcal{H}}_{j,s} \middle| \|f\|_{\breve{\mathcal{H}}_{j,s}} \le \breve{B}_j \right\}$$

where the closest to optimal approximations of $f^\star$ in the hypothesis spaces are:

$$\hat{\hat{f}}_{i,s}^{\#} = \operatorname*{argmin}_{f \in \hat{\mathbb{H}}_{i,s}} \|f - f^\star\|_\infty, \quad \breve{\breve{f}}_{j,s}^{\#} = \operatorname*{argmin}_{f \in \breve{\mathbb{H}}_{j,s}} \|f - f^\star\|_\infty$$

As is usual in practice, kernels may be updated when the GP models are updated, typically using max-log-likelihood for AIs or something more radical for the humans, which modifies the corresponding RKHSs. The difference between the best-in-class approximations and $f^\star$ for batch $s$ are assumed bounded as:

$$\left\| \hat{\hat{f}}_{i,s}^{\#} - f^\star \right\|_\infty \le \hat{\hat{\epsilon}}_{i,s}, \quad \left\| \breve{\breve{f}}_{j,s}^{\#} - f^\star \right\|_\infty \le \breve{\breve{\epsilon}}_{j,s}$$

After each batch $s$, we have the (nominal) GP models built on (nominal) observations of $\hat{\hat{f}}_{i,s}^{\#}$ and $\breve{\breve{f}}_{j,s}^{\#}$, which have the same variance as the (real) models (the posterior variance is independent of $\mathbf{y}$) but different posterior means:

$$\hat{\hat{\mu}}_{i,s}^{\#}(\mathbf{x}) = \hat{\hat{\boldsymbol{\alpha}}}_{i,s}^{\#\mathrm{T}} \hat{\hat{\mathbf{k}}}_{i,s}(\mathbf{x}), \quad \breve{\breve{\mu}}_{j,s}^{\#}(\mathbf{x}) = \breve{\breve{\boldsymbol{\alpha}}}_{j,s}^{\#\mathrm{T}} \breve{\mathbf{k}}_{j,s}(\mathbf{x})$$

We assume that test points are generated, either nominally (for humans) or directly (for AIs), EC-GP-UCB style (Bogunovic & Krause, 2021), from a sequence of interleaved $\beta_t$ sequences by the relevant human $i = t - \hat{t}$ if $i \in \hat{\mathbb{M}}$ or AI $j = t - \breve{t}$ if $j \in \breve{\mathbb{M}}$ in batch $s_t$, so:

$$\beta_t = \begin{cases} \hat{\beta}_{i,\underline{t}} & \text{if } i = t - \hat{\underline{t}} \in \hat{\mathbb{M}} \\ \breve{\beta}_{j,\underline{t}} & \text{if } j = t - \breve{\underline{t}} \in \breve{\mathbb{M}} \end{cases}$$

$$\epsilon_t = \begin{cases} \hat{\epsilon}_{i,\underline{t}} & \text{if } i = t - \hat{\underline{t}} \in \hat{\mathbb{M}} \\ \breve{\epsilon}_{j,\underline{t}} & \text{if } j = t - \breve{\underline{t}} \in \breve{\mathbb{M}} \end{cases}$$

$$\mu_{t-1}(\mathbf{x}) = \begin{cases} \hat{\bar{\mu}}_{i,s_t-1}(\mathbf{x}) & \text{if } i = t - \hat{\underline{t}} \in \hat{\mathbb{M}} \\ \breve{\bar{\mu}}_{j,s_t-1}(\mathbf{x}) & \text{if } j = t - \breve{\underline{t}} \in \breve{\mathbb{M}} \end{cases}$$

$$\sigma_{t-1}(\mathbf{x}) = \begin{cases} \hat{\bar{\sigma}}_{i,s_t-1}(\mathbf{x}) & \text{if } i = t - \hat{\underline{t}} \in \hat{\mathbb{M}} \\ \breve{\bar{\sigma}}_{j,s_t-1}(\mathbf{x}) & \text{if } j = t - \breve{\underline{t}} \in \breve{\mathbb{M}} \end{cases}$$

using the acquisition function:

$$\alpha_t(\mathbf{x}) = \mu_{t-1}(\mathbf{x}) + \left( \sqrt{\beta_t} + \tfrac{\epsilon_t}{\sigma}\sqrt{\underline{t}} \right) \sigma_{t-1}(\mathbf{x})$$

Generally we cannot control the human's exploitation/exploration trade-off sequence $\hat{\beta}_{i,t}$, but we assume humans are conservative, so the sequence may be assumed small. We use the AI trade-off sequence $\breve{\beta}_{j,t}$, which we do control, to compensate for the conservative tendencies of the humans involved. We do however assume that the humans include at least some exploration in their decisions on the understanding that their knowledge is not, in fact, perfect, so $\hat{\beta}_{i,t} \ge 0$ (note that a human who over-estimates their abilities may fail to meet this requirement, so care is required to avoid this). For reasons which will become apparent, we assume that:

$$\hat{\beta}_{i,\underline{t}_s} \in \left( \hat{\bar{\zeta}}_{i,s\downarrow} \hat{\chi}_{i,s}, \hat{\bar{\zeta}}_{i,s\uparrow} \hat{\chi}_{i,s} \right)$$
$$\breve{\beta}_{j,\underline{t}_s} \in \left( \breve{\bar{\zeta}}_{j,s\downarrow} \breve{\chi}_{j,s}, \breve{\bar{\zeta}}_{j,s\uparrow} \breve{\chi}_{j,s} \right)$$

where $\hat{\breve{\zeta}}_{i,s\downarrow} \leq 1$, which we will see makes the humans over-exploitative, and $\breve{\zeta}_{j,s\downarrow} \geq 1$, which we will see makes the AIs over-explorative, and:

$$\hat{\breve{\chi}}_{i,s} = \left( \frac{\Sigma}{\sqrt{\sigma}} \sqrt{2\ln\left(\frac{1}{\delta}\right) + 1 + \hat{\breve{\gamma}}_{i,s}} + \left\| \hat{\breve{f}}_{i,s}^{\#} \right\|_{\hat{\breve{\mathcal{H}}}_{i,s}} \right)^2$$

$$\breve{\chi}_{j,s} = \left( \frac{\Sigma}{\sqrt{\sigma}} \sqrt{2\ln\left(\frac{1}{\delta}\right) + 1 + \breve{\gamma}_{j,s}} + \left\| \breve{f}_{j,s}^{\#} \right\|_{\breve{\mathcal{H}}_{j,s}} \right)^2$$

where $\hat{\breve{\gamma}}_{i,s}$ and $\breve{\gamma}_{j,s}$ are, respectively, the max-information-gain terms for humans $i \in \hat{\mathbb{M}}$ and AIs $j \in \breve{\mathbb{M}}$, as will be described shortly.

## A.2 NOTES ON MAXIMUM INFORMATION GAIN

We are assuming $p$ GPs, each of which will have a different information gain that is a function of its kernel. All have the same dataset and get updated at the batch boundary. Thus, after $S$ batches, if we consider human $i \in \hat{\mathbb{M}}$:

$$\begin{aligned}
\hat{I}_i\left(\mathbf{y}_{Sp} : \mathbf{f}_{Sp}\right) &= H\left(\mathbf{y}_{Sp}\right) - \tfrac{1}{2}\ln\left|2\pi e \hat{\sigma}^2 \mathbf{I}_{Sp}\right| \\
&= H\left(\mathbf{y}_{Sp-1}\right) + H\left(y_{Sp} | \mathbf{y}_{(S-1)p}\right) - \tfrac{1}{2}\ln\left|2\pi e \hat{\sigma}^2 \mathbf{I}_{Sp}\right| \\
&= H\left(\mathbf{y}_{Sp-2}\right) + H\left(y_{Sp} | \mathbf{y}_{(S-1)p}\right) + H\left(y_{Sp-1} | \mathbf{y}_{(S-1)p}\right) - \tfrac{1}{2}\ln\left|2\pi e \hat{\sigma}^2 \mathbf{I}_{Sp}\right| \\
&= \ldots \\
&= H\left(\mathbf{y}_{(S-1)p}\right) + \sum_{t\in\mathbb{T}_S} H\left(y_t | \mathbf{y}_{t-1}\right) - \tfrac{1}{2}\ln\left|2\pi e \hat{\sigma}^2 \mathbf{I}_{Sp}\right| \\
&= H\left(\mathbf{y}_{(S-2)p}\right) + \sum_{t\in\mathbb{T}_S} H\left(y_t | \mathbf{y}_{t-1}\right) + \sum_{t\in\mathbb{T}_{S-1}} H\left(y_t | \mathbf{y}_{t-1}\right) - \tfrac{1}{2}\ln\left|2\pi e \hat{\sigma}^2 \mathbf{I}_{Sp}\right| \\
&= \ldots \\
&= \sum_{t\in\mathbb{T}_{\leq S}} H\left(y_t | \mathbf{y}_{t-1}\right) - \tfrac{1}{2}\ln\left|2\pi e \hat{\sigma}^2 \mathbf{I}_{Sp}\right| \\
&= \tfrac{1}{2} \sum_{t\in\mathbb{T}_{\leq S}} \ln\left(2\pi e \left(\hat{\sigma}^2 + \hat{\sigma}^{-2} \hat{\sigma}_{i,t-1}^2\left(\mathbf{x}_t\right)\right)\right) - \tfrac{1}{2}\ln\left|2\pi e \hat{\sigma}^2 \mathbf{I}_T\right| \\
&= \tfrac{1}{2} \sum_{t\in\mathbb{T}_{\leq S}} \ln\left(1 + \hat{\sigma}^{-2} \hat{\sigma}_{i,t-1}^2\left(\mathbf{x}_t\right)\right)
\end{aligned}$$

where we have used that $\mathbf{x}_1, \ldots, \mathbf{x}_{sp}$ are deterministic conditioned on $\mathbf{y}_{(s-1)p}$, and that the variances do not depend on $\mathbf{y}$. The derivation for AI $j$ is essentially identical. In summary, therefore, the information gain is:

$$\hat{I}_i\left(\mathbf{y}_{Sp} : \mathbf{f}_{Sp}\right) = \tfrac{1}{2} \sum_{t\in\mathbb{T}_{\leq S}} \ln\left(1 + \hat{\sigma}^{-2} \hat{\sigma}_{i,t-1}^2\left(\mathbf{x}_t\right)\right) \leq \hat{\gamma}_{i,S}$$

$$\breve{I}_j\left(\mathbf{y}_{Sp} : \mathbf{f}_{Sp}\right) = \tfrac{1}{2} \sum_{t\in\mathbb{T}_{\leq S}} \ln\left(1 + \breve{\sigma}^{-2} \breve{\sigma}_{j,t-1}^2\left(\mathbf{x}_t\right)\right) \leq \breve{\gamma}_{j,S}$$

where $\hat{\gamma}_{i,S}$ and $\breve{\gamma}_{j,S}$ are, respectively, the maximum information gains for human $i$ and AI $j$ over $S$ batches.

## A.3 NOTES ON THE HUMAN MODEL

We posit that every human $i \in \hat{\mathbb{M}}$ has an *evolving* model of the system:

$$\hat{\bar{f}}_{i,s}\left(\mathbf{x}\right) = \hat{\bar{g}}_{i,s}\left(\hat{\bar{\mathbf{p}}}_{i,s}\left(\mathbf{x}\right)\right)$$

where $\hat{\bar{g}}_{i,s}$ is in some sense "simple" and $\hat{\bar{\mathbf{p}}}_{i,s} : \mathbb{R}^n \to \mathbb{R}^{\hat{\bar{m}}_{i,s}}$. This fits into the above scheme if we let the form of $\hat{\bar{g}}_{i,s}$ dictate the kernel $\hat{\bar{K}}_{i,s}$. For example if we know that $\hat{\bar{g}}_{i,s}$ is linear - *i.e.,* the human is known to be using some heuristic model of the form:

$$\hat{\bar{f}}_{i,s}\left(\mathbf{x}\right) = \hat{\bar{\mathbf{w}}}_{i,s}^{\mathrm{T}} \hat{\bar{\mathbf{p}}}_{i,s}\left(\mathbf{x}\right)$$

then we can use a GP with a linear-derived kernel:

$$\hat{\bar{K}}_{i,s}\left(\mathbf{x}, \mathbf{x}'\right) = \hat{\bar{\mathbf{p}}}_{i,s}^{\mathrm{T}}\left(\mathbf{x}\right) \hat{\bar{\mathbf{p}}}_{i,s}\left(\mathbf{x}'\right)$$

Similarly if the human is using a model $\hat{\bar{g}}_{i,s}$ that can be captured by a $d^{\text{th}}$-order polynomial model:

$$\hat{\bar{f}}_{i,s}\left(\mathbf{x}\right) = \hat{\bar{\mathbf{w}}}_{i,s}^{\mathrm{T}}\left[\sqrt{\binom{d}{q}}\hat{\bar{\mathbf{p}}}_{i,s}^{\otimes q}\left(\mathbf{x}\right)\right]_{q\in\mathbb{N}_{d+1}}$$

then we can use a GP with a polynomial-derived kernel:

$$\hat{\bar{K}}_{i,s}\left(\mathbf{x}, \mathbf{x}'\right) = \left(1 + \hat{\bar{\mathbf{p}}}_{i,s}^{\mathrm{T}}\left(\mathbf{x}\right) \hat{\bar{\mathbf{p}}}_{i,s}\left(\mathbf{x}'\right)\right)^{d}$$

Alternatively, if $\hat{\bar{g}}_{i,s}$ is more vague (*i.e.,* the researcher knows that the factors $\hat{\bar{\mathbf{p}}}_{i,s}$ are important but not the exact form of the relationship) then we might use a GP assuming a distance-based model:

$$\hat{\bar{K}}_{i,s}\left(\mathbf{x}, \mathbf{x}'\right) = \exp\left(-\tfrac{1}{2l}\left\|\hat{\bar{\mathbf{p}}}_{i,s}\left(\mathbf{x}\right) - \hat{\bar{\mathbf{p}}}_{i,s}\left(\mathbf{x}'\right)\right\|_{2}^{2}\right)$$

or some similarly generic model that captures the worst-case behaviour of the human, along, hopefully, with some insight into the thought processes used by them.

With regard to maximum information gain, because the human models are evolving, $\hat{\bar{m}}_{i,s}$ will change with $s$, so it is convenient to define define $\hat{\bar{m}}_{i,s\uparrow} = \max_{t\in\mathbb{T}_s}\hat{\bar{m}}_{i,s}$ to capture the worst-case feature-space dimensionality over $S$ batches. We can then bound the maximum information gain for the human as the worst-case of these models over all $S$ batches. So for example, depending on the specifics of $\hat{\bar{g}}_{i,s}$, we have (Srinivas et al., 2012; Scetbon & Harchaoui, 2021):

$$
\begin{aligned}
\text{Linear:} \quad & \hat{\bar{\gamma}}_{i,s} = \mathcal{O}\left(\ln\left(Sp\right)\right) \\
\text{Polynomial:} \quad & \hat{\bar{\gamma}}_{i,s} = \mathcal{O}\left(\ln\left(Sp\right)\right) \\
\text{Squared-Exponential:} \quad & \hat{\bar{\gamma}}_{i,s} = \mathcal{O}\left(\ln^{\hat{\bar{m}}_{i,s\uparrow}+1}\left(Sp\right)\right)
\end{aligned}
$$

In general we assume that the asymptotic behaviour of the human maximum information gain converges more quickly than that of the machine models. This makes intuitive sense of the human is applying a linear or polynomial heuristic model $\hat{\bar{g}}_{i,s}$, which is captured by a linear or polynomial kernel, while the machines use more general GP models with SE or Matern type kernels (as is common practice). Thus in this case the human has a better behaved maximum information gain at the cost of a potentially non-zero gap $\hat{\bar{\epsilon}}_{i,s}$ (presumably trending to 0 as the human gains improved insight into the problem and evolves their model to better match the problem), while the machine has a worse behaved maximum information gain but zero gap (assuming a universal kernel like an SE kernel).

## A.4 MATHEMATICAL PRELIMINARIES

We use $p$-norms extensively, where:

$$
\begin{aligned}
\|\mathbf{x}\|_p &= \left(\sum_i |x_i|^p\right)^{\frac{1}{p}} && \text{where } \mathbf{x} \in \mathbb{R}^n \\
\|f\|_p &= \left(\int_{\mathbf{x}} |f\left(\mathbf{x}\right)|^p \, d\mathbf{x}\right)^{\frac{1}{p}} && \text{where } f : \mathbb{R}^n \to \mathbb{R}
\end{aligned}
$$

with the extensions:
$$
\begin{aligned}
\|\mathbf{x}\|_{\infty} &= \max_i |x_i| \\
\|\mathbf{x}\|_{-\infty} &= \min_i |x_i|
\end{aligned}
$$
$$
\begin{aligned}
\|f\|_{\infty} &= \inf\left\{C \geq 0 : |f\left(\mathbf{x}\right)| \leq C \text{ for almost all } \mathbf{x} \in \mathbb{R}^n\right\} \\
\|f\|_{-\infty} &= \sup\left\{C \geq 0 : |f\left(\mathbf{x}\right)| \geq C \text{ for almost all } \mathbf{x} \in \mathbb{R}^n\right\}
\end{aligned}
$$

This is only a norm for $p \in [1, \infty]$ (though we may occasionally refer to it as such in a loose sense), but is well defined for $p \in [-\infty, 0) \cup (0, \infty]$. It is not difficult to see that:

$$
\begin{aligned}
\|\mathbf{x}\|_p &\geq \|\mathbf{x}\|_{p'}, && \forall p, p' \in (0, \infty], p \leq p' \\
\|\mathbf{x}\|_p &\geq \|\mathbf{x}\|_{p'}, && \forall p, p' \in [-\infty, 0), p \leq p' \\
\|\mathbf{x}\|_p &\geq \|\mathbf{x}\|_{p'}, && \forall p \in (0, \infty], p' \in [-\infty, 0)
\end{aligned}
$$

(the final inequality follows from the first two, noting that $\min_i |x_i| < \max_i |x_i|$, and similarly for functions); and furthermore:

$$\|\mathbf{x}\|_p \leq n^{\frac{1}{p} - \frac{1}{p'}} \|\mathbf{x}\|_{p'} \quad \forall 1 \leq p \leq p' \leq \infty$$

when $\mathbf{x} \in \mathbb{R}^n$. We also use the following inequalities ($\odot$ is the Hadamaard product):

$$\left. \begin{array}{l} \|\mathbf{x} \odot \mathbf{x}' \odot \ldots \odot \mathbf{x}''''\|_r \leq \|\mathbf{x}\|_p \|\mathbf{x}'\|_{p'} \cdots \|\mathbf{x}''''\|_{p''''} \\ \|f \odot f' \odot \ldots \odot f''''\|_r \leq \|f\|_p \|f'\|_{p'} \cdots \|f''''\|_{p''''} \end{array} \right\} \text{ General. Hölder inequality}$$

$$\left. \begin{array}{l} \|\mathbf{x} \odot \mathbf{x}'\|_1 \geq \|\mathbf{x}\|_{\frac{1}{q}} \|\mathbf{x}'\|_{\frac{-1}{q-1}} \\ \|f \odot f'\|_1 \geq \|f\|_{\frac{1}{q}} \|f'\|_{\frac{-1}{q-1}} \end{array} \right\} \text{ Reverse Hölder inequality}$$

where $\frac{1}{p} + \frac{1}{p'} + \ldots + \frac{1}{p''''} = \frac{1}{r}$ and $r, p, p', \ldots, p'''' \in (0, \infty]$, with the convention $\frac{1}{\infty} = 0$; and $q \in (0, \infty)$.

We also use generalised (power) mean (Pečarić, 1991; Pyun, 1974), defined as:

$$M_\theta\left(\{a_1, a_2, \ldots a_n\}\right) = \begin{cases} \min_i a_i & \text{if } \theta = -\infty \\ \left(\frac{1}{n} \sum_i a_i^\theta\right)^{\frac{1}{\theta}} & \text{if } \theta \in (-\infty, 0) \\ \left(\prod_i a_i\right)^{\frac{1}{n}} & \text{if } \theta = 0 \\ \left(\frac{1}{n} \sum_i a_i^\theta\right)^{\frac{1}{\theta}} & \text{if } \theta \in (0, \infty) \\ \max_i a_i & \text{if } \theta = \infty \end{cases}$$

where $\theta \in [-\infty, \infty]$. For example $\theta = -1, 0, 1$ correspond, respectively, to the harmonic, geometric and arithmetic means. Note that, for all $\theta, \theta' \in [-\infty, \infty]$, $\theta \leq \theta'$:

$$M_\theta\left(\{a_1, a_2, \ldots a_n\}\right) \leq M_{\theta'}\left(\{a_1, a_2, \ldots a_n\}\right)$$

With a minor abuse of notation, we define:

$$M_\theta\left(\mathbf{a}\right) = M_\theta\left(\{a_1, a_2, \ldots a_n\}\right)$$

for $\mathbf{a} \in \mathbb{R}^n$, so that, for $\mathbf{a} \in \mathbb{R}^n_+$, we have the connection to the $p$-norms:

$$M_\theta\left(\mathbf{a}\right) = \begin{cases} n^{-\frac{1}{\theta}} \|\mathbf{a}\|_\theta & \text{if } \theta \in [-\infty, 0) \cup (0, \infty] \\ \left(\prod_i a_i\right)^{\frac{1}{n}} & \text{if } \theta = 0 \end{cases}$$

The following result is central to our proof:[4]

**Lemma 2.** *Let* $\mathbf{a}, \mathbf{b} \in \mathbb{R}^n_+$, $z \in [-\infty, \infty]$, $q, q' \in [1, \infty]$, $\frac{1}{q} + \frac{1}{q'} = 1$, $\frac{1}{\infty} = 0$. *Then we have the following bounds on* $M_{-\infty}(\mathbf{a} \odot \mathbf{b})$:

$$\begin{array}{ll} M_{-\infty}\left(\mathbf{a} \odot \mathbf{b}\right) & \leq M_{-z}\left(\mathbf{a}\right) M_z\left(\mathbf{b}\right) \\ M_{-\infty}\left(\mathbf{a} \odot \mathbf{b}\right) & \leq M_{|zq|}\left(\mathbf{a}\right) M_{|zq'|}\left(\mathbf{b}\right) \\ M_{-\infty}\left(\mathbf{a} \odot \mathbf{b}\right) & \leq M_{-|z|}\left(\mathbf{a}\right) M_{\frac{q}{q-1}|z|}\left(\mathbf{b}\right) \end{array}$$

*Proof.* Let us suppose that, unlike in the theorem, $z \in (-\infty, 0) \cup (0, \infty)$. Then, using the generalised mean inequality:

$$M_{-\infty}\left(\mathbf{a} \odot \mathbf{b}\right) \leq M_z\left(\mathbf{a} \odot \mathbf{b}\right) = n^{-\frac{1}{z}} \|\mathbf{a} \odot \mathbf{b}\|_z = n^{-\frac{1}{z}} \|\mathbf{a}^{\odot z} \odot \mathbf{b}^{\odot z}\|_1^{\frac{1}{z}}$$

If $z > 0$ then, using Hölder's inequality:

$$\begin{aligned} M_{-\infty}\left(\mathbf{a} \odot \mathbf{b}\right) &= n^{-\frac{1}{|z|}} \|\mathbf{a}^{\odot |z|} \odot \mathbf{b}^{\odot |z|}\|_1^{\frac{1}{|z|}} \\ &\leq n^{-\frac{1}{|z|}} \|\mathbf{a}^{\odot |z|}\|_q^{\frac{1}{|z|}} \|\mathbf{b}^{\odot |z|}\|_{q'}^{\frac{1}{|z|}} \\ &= n^{-\frac{1}{|z|}} \|\mathbf{a}^{\odot |zq|}\|_1^{\frac{1}{|zq|}} \|\mathbf{b}^{\odot |zq'|}\|_1^{\frac{1}{|zq'|}} \\ &= n^{-\frac{1}{|z|}} n^{\frac{1}{|zq|}} n^{\frac{1}{|zq'|}} \|\tfrac{1}{n}\mathbf{a}^{\odot |zq|}\|_1^{\frac{1}{|zq|}} \|\tfrac{1}{n}\mathbf{b}^{\odot |zq'|}\|_1^{\frac{1}{|zq'|}} \\ &= n^{-\frac{1}{|zq'|}} \|\mathbf{a}\|_{|zq|} \, n^{-\frac{1}{|zq'|}} \|\mathbf{b}\|_{|zq'|} \\ &= M_{|zq|}\left(\mathbf{a}\right) M_{|zq'|}\left(\mathbf{b}\right) \end{aligned}$$

---

[4]This result may be well known, but we have been unable to find it in the literature.

If $z < 0$ then, using the reverse Hölder inequality (noting the negative exponent):

$$
\begin{aligned}
M_{-\infty}\left(\mathbf{a}\odot\mathbf{b}\right) &= n^{\frac{1}{|z|}}\left\|\mathbf{a}^{\odot-|z|}\odot\mathbf{b}^{\odot-|z|}\right\|_1^{-\frac{1}{|z|}} \\
&\leq n^{\frac{1}{|z|}}\left\|\mathbf{a}^{\odot-|z|}\right\|_{\frac{1}{q}}^{-\frac{1}{|z|}}\left\|\mathbf{b}^{\odot-|z|}\right\|_{-\frac{1}{q-1}}^{-\frac{1}{q-1}} \\
&= n^{\frac{1}{|z|}}\left\|\mathbf{a}^{\odot-\frac{|z|}{q}}\right\|_1^{-\frac{q}{|z|}}\left\|\mathbf{b}^{\odot\frac{|z|}{q-1}}\right\|_1^{\frac{q-1}{|z|}} \\
&= n^{\frac{1}{|z|}}n^{-\frac{q}{|z|}}n^{\frac{q-1}{|z|}}\left\|\frac{1}{n}\mathbf{a}^{\odot-\frac{|z|}{q}}\right\|_1^{-\frac{q}{|z|}}\left\|\frac{1}{n}\mathbf{b}^{\odot\frac{|z|}{q-1}}\right\|_1^{\frac{q-1}{|z|}} \\
&= n^{\frac{q}{|z|}}\left\|\mathbf{a}\right\|_{-\frac{|z|}{q}}n^{-\frac{q-1}{|z|}}\left\|\mathbf{b}\right\|_{\frac{|z|}{q-1}} \\
&= M_{-\frac{|z|}{q}}\left(\mathbf{a}\right)M_{\frac{|z|}{q-1}}\left(\mathbf{b}\right)
\end{aligned}
$$

It is instructive to let $r = |z|/q$. Then the most recent bound becomes:

$$
M_{-\infty}\left(\mathbf{a}\odot\mathbf{b}\right) \leq M_{-r}\left(\mathbf{a}\right)M_{\frac{q}{q-1}r}\left(\mathbf{b}\right)
$$

which in the limit $q \to \infty$ simplifies to:

$$
M_{-\infty}\left(\mathbf{a}\odot\mathbf{b}\right) \leq M_{-r}\left(\mathbf{a}\right)M_r\left(\mathbf{b}\right)
$$

Finally, using the definitions:

$$
M_{-\infty}\left(\mathbf{a}\odot\mathbf{b}\right) \leq M_0\left(\mathbf{a}\odot\mathbf{b}\right) = M_0\left(\mathbf{a}\right)M_0\left(\mathbf{b}\right)
$$

completing the proof. $\qquad\square$

## A.5 BOGUNOVIC'S LEMMA

We have the following from Bogunovic & Krause (2021):

$$
\left|\mu_{t-1}^{\#}\left(\mathbf{x}\right) - \mu_{t-1}\left(\mathbf{x}\right)\right| \leq \frac{\epsilon_t}{\sigma}\sqrt{\underline{t}}\sigma_{t-1}\left(\mathbf{x}\right) \tag{9}
$$

This follows from (Bogunovic & Krause, 2021, Lemma 2) using that all models satisfy this result, and that the model is posterior on the observations up to and including the previous batch. It follows from this and the definitions that, for all $\mathbf{x} \in \mathbb{X}$:

$$
\begin{aligned}
\left|\hat{\bar{\mu}}_{i,s-1}\left(\mathbf{x}\right) - f^\star\left(\mathbf{x}\right)\right| &\leq \left|\hat{\bar{\mu}}_{i,s-1}^{\#}\left(\mathbf{x}\right) - \hat{\bar{f}}_{i,s-1}^{\#}\left(\mathbf{x}\right)\right| + \frac{\hat{\epsilon}_{i,\underline{t}_s}}{\sigma}\sqrt{\underline{t}_s}\hat{\bar{\sigma}}_{i,s-1}\left(\mathbf{x}\right) + \hat{\epsilon}_{i,\underline{t}_s} \\
\left|\breve{\bar{\mu}}_{j,s-1}\left(\mathbf{x}\right) - f^\star\left(\mathbf{x}\right)\right| &\leq \left|\breve{\bar{\mu}}_{j,s-1}^{\#}\left(\mathbf{x}\right) - \breve{\bar{f}}_{j,s-1}^{\#}\left(\mathbf{x}\right)\right| + \frac{\breve{\epsilon}_{j,\underline{t}_s}}{\sigma}\sqrt{\underline{t}_s}\breve{\bar{\sigma}}_{j,s-1}\left(\mathbf{x}\right) + \breve{\epsilon}_{j,\underline{t}_s}
\end{aligned} \tag{10}
$$

## A.6 THE REGRET BOUND

We begin with the following uncertainty bound, which is largely based on the bound (Chowdhury & Gopalan, 2017, Theorem 2) and analogous to (Srinivas et al., 2012, Lemma 5.1) and (Bogunovic & Krause, 2021, Lemma 1):

**Lemma 3.** *Let $\delta \in (0,1)$. Assume noise variables are $\Sigma$-sub-Gaussian. Let:*

$$
\begin{aligned}
\hat{\bar{\chi}}_{i,s} &= \left(\frac{\Sigma}{\sqrt{\hat{\sigma}}}\sqrt{2\ln\left(\frac{1}{\delta}\right) + 1 + \hat{\bar{\gamma}}_{i,s}} + \left\|\hat{\bar{f}}_{i,s}^{\#}\right\|_{\hat{\bar{\mathcal{H}}}_{i,s}}\right)^2 \\
\breve{\bar{\chi}}_{j,s} &= \left(\frac{\Sigma}{\sqrt{\breve{\sigma}}}\sqrt{2\ln\left(\frac{1}{\delta}\right) + 1 + \breve{\bar{\gamma}}_{j,s}} + \left\|\breve{\bar{f}}_{j,s}^{\#}\right\|_{\breve{\bar{\mathcal{H}}}_{j,s}}\right)^2
\end{aligned} \tag{11}
$$

*Then, for all $i \in \hat{\mathbb{M}}$ and $j \in \breve{\mathbb{M}}$:*

$$\Pr\left\{\forall s, \forall \mathbf{x} \in \mathbb{X}, \left|\hat{\bar{\mu}}_{i,s-1}(\mathbf{x}) - f^\star(\mathbf{x})\right| \leq \ldots\right.$$

$$\left.\left(\sqrt{\hat{\bar{\chi}}_{i,s}} + \frac{\hat{\epsilon}_{i,\underline{t}_s}}{\hat{\bar{\sigma}}}\sqrt{\underline{t}_s}\right)\hat{\bar{\sigma}}_{i,s-1}(\mathbf{x}) + \hat{\epsilon}_{i,\underline{t}_s}\right\} \geq 1 - \delta$$

$$\Pr\left\{\forall s, \forall \mathbf{x} \in \mathbb{X}, \left|\breve{\bar{\mu}}_{j,s-1}(\mathbf{x}) - f^\star(\mathbf{x})\right| \leq \ldots\right.$$

$$\left.\left(\sqrt{\breve{\bar{\chi}}_{j,s}} + \frac{\breve{\epsilon}_{j,\underline{t}_s}}{\breve{\bar{\sigma}}}\sqrt{\underline{t}_s}\right)\breve{\bar{\sigma}}_{j,s-1}(\mathbf{x}) + \breve{\epsilon}_{j,\underline{t}_s}\right\} \geq 1 - \delta$$

*Proof.* We start with (Chowdhury & Gopalan, 2017, Theorem 2). This states that, in our setting, with probability $\geq 1 - \delta$, simultaneously for all $s \geq 1$ and $\mathbf{x} \in D$:

$$\left|\hat{\mu}^{\#}_{i,s-1}(\mathbf{x}) - \hat{\bar{f}}^{\#}_{i,s-1}(\mathbf{x})\right| \leq \hat{\bar{\sigma}}^{-1}_{i,s-1}(\mathbf{x})\sqrt{\hat{\bar{\chi}}_{i,s}}$$

$$\left|\breve{\mu}^{\#}_{j,s-1}(\mathbf{x}) - \breve{\bar{f}}^{\#}_{j,s-1}(\mathbf{x})\right| \leq \breve{\bar{\sigma}}^{-1}_{j,s-1}(\mathbf{x})\sqrt{\breve{\bar{\chi}}_{j,s}}$$

for $\hat{\bar{\chi}}_{i,s}, \breve{\bar{\chi}}_{j,s}$ as specified by (11). Next, recall (10):

$$\left|\hat{\bar{\mu}}_{i,s-1}(\mathbf{x}) - f^\star(\mathbf{x})\right| \leq \left|\hat{\mu}^{\#}_{i,s-1}(\mathbf{x}) - \hat{\bar{f}}^{\#}_{i,s-1}(\mathbf{x})\right| + \frac{\hat{\epsilon}_{i,s}}{\hat{\bar{\sigma}}}\sqrt{\underline{t}_s}\hat{\bar{\sigma}}_{i,s-1}(\mathbf{x}) + \hat{\epsilon}_{i,s}$$

$$\left|\breve{\bar{\mu}}_{j,s-1}(\mathbf{x}) - f^\star(\mathbf{x})\right| \leq \left|\breve{\mu}^{\#}_{j,s-1}(\mathbf{x}) - \breve{\bar{f}}^{\#}_{j,s-1}(\mathbf{x})\right| + \frac{\breve{\epsilon}_{j,s}}{\breve{\bar{\sigma}}}\sqrt{\underline{t}_s}\breve{\bar{\sigma}}_{j,s-1}(\mathbf{x}) + \breve{\epsilon}_{j,s}$$

and the result follows. $\qquad\square$

**Remark:** Alternatively, following (Fiedler et al., 2021, Theorem 1) we can use:

$$\hat{\bar{\chi}}_{i,s} \geq \left(\frac{\Sigma}{\sqrt{\hat{\bar{\sigma}}}}\sqrt{2\ln\left(\frac{1}{\delta}\right) + \ln\left(\det\left(\hat{\bar{\mathbf{K}}}_{i,s-1} + \hat{\bar{\sigma}}^2\mathbf{I}\right)\right)} + \left\|\hat{\bar{f}}^{\#}_{i,s}\right\|_{\hat{\bar{\mathcal{H}}}_{i,s}}\right)^2$$

$$\breve{\bar{\chi}}_{j,s} \geq \left(\frac{\Sigma}{\sqrt{\breve{\bar{\sigma}}}}\sqrt{2\ln\left(\frac{1}{\delta}\right) + \ln\left(\det\left(\breve{\bar{\mathbf{K}}}_{j,s-1} + \breve{\bar{\sigma}}^2\mathbf{I}\right)\right)} + \left\|\breve{\bar{f}}^{\#}_{j,s}\right\|_{\breve{\bar{\mathcal{H}}}_{j,s}}\right)^2$$

Using the uncertainty bound above, we obtain our first bound on instantaneous experiment-wise regret based on (Srinivas et al., 2012, Lemma 5.2) with some techniques borrowed from Bogunovic & Krause (2021):

**Lemma 4.** *Let $\delta \in (0,1)$. Assume noise variables are $\Sigma$-sub-Gaussian. Assume that:*

$$\hat{\beta}_{i,\underline{t}_s} \in \left(\hat{\bar{\zeta}}_{i,s\downarrow}\hat{\bar{\chi}}_{i,s}, \hat{\bar{\zeta}}_{i,s\uparrow}\hat{\bar{\chi}}_{i,s}\right)$$

$$\breve{\beta}_{j,\underline{t}_s} = \left(\breve{\bar{\zeta}}_{j,s\downarrow}\breve{\bar{\chi}}_{j,s}, \breve{\bar{\zeta}}_{j,s\uparrow}\breve{\bar{\chi}}_{j,s}\right)$$

*where:*

$$\hat{\bar{\chi}}_{i,s} = \left(\frac{\Sigma}{\sqrt{\hat{\bar{\sigma}}}}\sqrt{2\ln\left(\frac{1}{\delta}\right) + 1 + \hat{\bar{\gamma}}_{i,s}} + \left\|\hat{\bar{f}}^{\#}_{i,s}\right\|_{\hat{\bar{\mathcal{H}}}_{i,s}}\right)^2$$

$$\breve{\bar{\chi}}_{j,s} = \left(\frac{\Sigma}{\sqrt{\breve{\bar{\sigma}}}}\sqrt{2\ln\left(\frac{1}{\delta}\right) + 1 + \breve{\bar{\gamma}}_{j,s}} + \left\|\breve{\bar{f}}^{\#}_{j,s}\right\|_{\breve{\bar{\mathcal{H}}}_{j,s}}\right)^2$$

*Then, simultaneously for all $s \geq 1$, the instantaneous regret is bounded as:*

$$r_{\underline{t}_s + i} \leq \hat{\bar{r}}_{i,s\uparrow} + \left(1 - \sqrt{\hat{\bar{\zeta}}_{i,s\downarrow}}\right)\sqrt{\hat{\bar{\chi}}_{i,s}}\hat{\bar{\sigma}}_{i,s-1}(\mathbf{x}^\star)$$

$$r_{\underline{t}_s + j} \leq \breve{\bar{r}}_{j,s\uparrow} + \left(1 - \sqrt{\breve{\bar{\zeta}}_{j,s\downarrow}}\right)\sqrt{\breve{\bar{\chi}}_{j,s}}\breve{\bar{\sigma}}_{j,s-1}(\mathbf{x}^\star)$$

*with probability $\geq 1 - \delta$, where:*

$$\hat{\breve{r}}_{i,s\uparrow} = 2 \left( \frac{1+\sqrt{\hat{\breve{\zeta}}_{i,s\uparrow}}}{2} \sqrt{\hat{\breve{\chi}}_{i,s}} + \frac{\hat{\epsilon}_{i,t_s}}{\hat{\breve{\sigma}}} \sqrt{t_s} \right) \hat{\breve{\sigma}}_{i,s-1} \left( \mathbf{x}_{\underline{t}_s + i} \right) + 2\hat{\epsilon}_{i,t_s}$$

$$\breve{r}_{j,s\uparrow} = 2 \left( \frac{1+\sqrt{\breve{\zeta}_{j,s\uparrow}}}{2} \sqrt{\breve{\chi}_{j,s}} + \frac{\breve{\epsilon}_{j,t_s}}{\breve{\sigma}} \sqrt{t_s} \right) \breve{\sigma}_{j,s-1} \left( \mathbf{x}_{\underline{t}_s + j} \right) + 2\breve{\epsilon}_{j,t_s}$$

*Proof.* By the pretext and Lemma 3:

$$f^\star (\mathbf{x}^\star) \leq \hat{\breve{\mu}}_{i,s-1} (\mathbf{x}^\star) + \left( \sqrt{\hat{\breve{\chi}}_{i,s}} + \frac{\hat{\epsilon}_{i,t_s}}{\hat{\breve{\sigma}}} \sqrt{t_s} \right) \hat{\breve{\sigma}}_{i,s-1} (\mathbf{x}^\star) + \hat{\epsilon}_{i,t_s}$$

with probability $\geq 1 - \delta$. By definition $\mathbf{x}_t$ maximises the acquisition function, so:

$$\hat{\breve{\mu}}_{i,s-1} (\mathbf{x}^\star) \leq \hat{\breve{\mu}}_{i,s-1} \left( \mathbf{x}_{\underline{t}_s+i} \right) + \left( \sqrt{\hat{\beta}_{i,t_s}} + \frac{\hat{\epsilon}_{i,t_s}}{\hat{\breve{\sigma}}} \sqrt{t_s} \right) \hat{\breve{\sigma}}_{i,s-1} \left( \mathbf{x}_{\underline{t}_s+i} \right)$$
$$- \left( \sqrt{\hat{\beta}_{i,t_s}} + \frac{\hat{\epsilon}_{i,t_s}}{\hat{\breve{\sigma}}} \sqrt{t_s} \right) \hat{\breve{\sigma}}_{i,s-1} (\mathbf{x}^\star)$$

and hence:

$$f^\star (\mathbf{x}^\star) - f^\star \left( \mathbf{x}_{\underline{t}_s+i} \right) \leq \hat{\breve{\mu}}_{i,\}s-1} \left( \mathbf{x}_{\underline{t}_s+i} \right) + \left( \sqrt{\hat{\beta}_{i,t_s}} + \frac{\hat{\epsilon}_{i,t_s}}{\hat{\breve{\sigma}}} \sqrt{t_s} \right) \hat{\breve{\sigma}}_{i,s-1} \left( \mathbf{x}_{\underline{t}_s+i} \right) + \hat{\epsilon}_{i,t_s} \cdots$$
$$+ \left( \sqrt{\hat{\breve{\chi}}_{i,s}} - \sqrt{\hat{\beta}_{i,t_s}} \right) \hat{\breve{\sigma}}_{i,s-1} (\mathbf{x}^\star) - f^\star \left( \mathbf{x}_{\underline{t}_s+i} \right)$$

It follows that the instantaneous regret is bounded by:

$$r_{\underline{t}_s+i} \leq \hat{\breve{\mu}}_{i,s-1} \left( \mathbf{x}_{\underline{t}_s+i} \right) + \left( \sqrt{\hat{\beta}_{i,t_s}} + \frac{\hat{\epsilon}_{i,t_s}}{\hat{\breve{\sigma}}} \sqrt{t_s} \right) \hat{\breve{\sigma}}_{i,s-1} \left( \mathbf{x}_{\underline{t}_s+i} \right) + \hat{\epsilon}_{i,t_s} \cdots$$
$$+ \left( \sqrt{\hat{\breve{\chi}}_{i,s}} - \sqrt{\hat{\beta}_{i,t_s}} \right) \hat{\breve{\sigma}}_{i,s-1} (\mathbf{x}^\star) - f^\star \left( \mathbf{x}_{\underline{t}_s+i} \right)$$

So, using our assumptions on $\beta$ we find that:

$$r_{\underline{t}_s+i} \leq \hat{\breve{\mu}}_{i,s-1} \left( \mathbf{x}_{\underline{t}_s+i} \right) + \left( \sqrt{\hat{\breve{\zeta}}_{i,s\uparrow}} \sqrt{\hat{\breve{\chi}}_{i,s}} + \frac{\hat{\epsilon}_{i,t_s}}{\hat{\breve{\sigma}}} \sqrt{t_s} \right) \hat{\breve{\sigma}}_{i,s-1} \left( \mathbf{x}_{\underline{t}_s+i} \right) + \hat{\epsilon}_{i,t_s} \cdots$$
$$+ \left( 1 - \sqrt{\hat{\breve{\zeta}}_{i,s\downarrow}} \right) \sqrt{\hat{\breve{\chi}}_{i,s}} \hat{\breve{\sigma}}_{i,s-1} (\mathbf{x}^\star) - f^\star \left( \mathbf{x}_{\underline{t}_s+i} \right)$$

Once again using our pretext and Lemma 3 we see that:

$$f^\star \left( \mathbf{x}_{\underline{t}_s+i} \right) \geq \hat{\breve{\mu}}_{i,s-1} \left( \mathbf{x}_{\underline{t}_s+i} \right) - \left( \sqrt{\hat{\breve{\chi}}_{i,s}} + \frac{\hat{\epsilon}_{i,t_s}}{\hat{\breve{\sigma}}} \sqrt{t_s} \right) \hat{\breve{\sigma}}_{i,s-1} \left( \mathbf{x}_{\underline{t}_s+i} \right) - \hat{\epsilon}_{i,t_s}$$

Hence:

$$r_{\underline{t}_s+i} \leq 2 \left( \frac{1+\sqrt{\hat{\breve{\zeta}}_{i,s\uparrow}}}{2} \sqrt{\hat{\breve{\chi}}_{i,s}} + \frac{\hat{\epsilon}_{i,t_s}}{\hat{\breve{\sigma}}} \sqrt{t_s} \right) \hat{\breve{\sigma}}_{i,s-1} \left( \mathbf{x}_{\underline{t}_s+i} \right) \cdots$$
$$+ \left( 1 - \sqrt{\hat{\breve{\zeta}}_{i,s\downarrow}} \right) \sqrt{\hat{\breve{\chi}}_{i,s}} \hat{\breve{\sigma}}_{i,s-1} (\mathbf{x}^\star) + 2\hat{\epsilon}_{i,t_s}$$

and the desired result follows from the definitions. The proof for AI regret follows by an analogous argument. □

To extend this to usable batchwise instantaneous regret bound we need to deal with the variance terms $\hat{\breve{\sigma}}_{i,s-1}(\mathbf{x}^\star)$, $\breve{\sigma}_{i,s-1}(\mathbf{x}^\star)$ in the above theorem. To do this, in the following theorem we use the generalised (power) mean, and in particular lemma 2, to split the batchwise risk bound into a constraint term containing the free variances and a risk bound that depends only on the various parameters of the problem:

**Lemma 5.** *Fix $\delta > 0$ and $\bar{\theta}_s \in [0, \infty]$.[5] Assume noise variables are $\Sigma$-sub-Gaussian, and that:*

$$
\begin{aligned}
\hat{\beta}_{i,\underline{t}_s} &\in \left( \hat{\bar{\zeta}}_{i,s\downarrow} \hat{\bar{\chi}}_{i,s}, \hat{\bar{\zeta}}_{i,s\uparrow} \hat{\bar{\chi}}_{i,s} \right) \\
\breve{\beta}_{j,\underline{t}_s} &= \left( \breve{\zeta}_{j,s\downarrow} \breve{\bar{\chi}}_{j,s}, \breve{\zeta}_{j,s\uparrow} \breve{\bar{\chi}}_{j,s} \right)
\end{aligned}
$$

*where $\hat{\bar{\zeta}}_{i,s\downarrow} \leq 1$, $\breve{\zeta}_{j,s\downarrow} \geq 1$, and:*

$$
\begin{aligned}
\hat{\bar{\chi}}_{i,s} &= \left( \frac{\Sigma}{\sqrt{\hat{\bar{\sigma}}}} \sqrt{2 \ln\left(\frac{1}{\delta}\right) + 1 + \hat{\bar{\gamma}}_{i,s}} + \left\| \hat{\bar{f}}^{\#}_{i,s} \right\|_{\hat{\mathcal{H}}_{i,s}} \right)^2 \\
\breve{\bar{\chi}}_{j,s} &= \left( \frac{\Sigma}{\sqrt{\breve{\bar{\sigma}}}} \sqrt{2 \ln\left(\frac{1}{\delta}\right) + 1 + \breve{\bar{\gamma}}_{j,s}} + \left\| \breve{\bar{f}}^{\#}_{j,s} \right\|_{\breve{\mathcal{H}}_{j,s}} \right)^2
\end{aligned}
$$

*If:*

$$
M_{\bar{\theta}_s} \left(
\begin{array}{l}
\left\{ \exp\left( \sqrt{\hat{\bar{\chi}}_{i,s}} \left(1 - \sqrt{\hat{\bar{\zeta}}_{i,s\downarrow}}\right)_+ \bar{\sigma}_{s-1\uparrow} \right) : i \in \hat{\mathbb{M}} \right\} \cup \ldots \\
\left\{ \exp\left( \sqrt{\breve{\bar{\chi}}_{j,s}} \left(1 - \sqrt{\breve{\zeta}_{j,s\downarrow}}\right)_- \bar{\sigma}_{s-1\downarrow} \right) : j \in \breve{\mathbb{M}} \right\}
\end{array}
\right) \leq 1
$$

*Then, simultaneously for all $s \geq 1$, the batchwise instantaneous regret is bounded as:*

$$
\bar{r}_s \leq \ln\left( M_{-\bar{\theta}_s}\left( \exp\left( \bar{\mathbf{r}}_{s\uparrow} \right) \right) \right)
$$

*with probability $\geq 1 - \delta$, where:*

$$
\bar{\mathbf{r}}_{s\uparrow} = \left[ \begin{array}{l}
\left[ 2\left( \frac{1+\sqrt{\hat{\bar{\zeta}}_{i,s\uparrow}}}{2} \sqrt{\hat{\bar{\chi}}_{i,s}} + \frac{\hat{\epsilon}_{i,\underline{t}_s}}{\hat{\bar{\sigma}}}\sqrt{\underline{t}_s} \right) \hat{\bar{\sigma}}_{i,s-1}\left( \mathbf{x}_{\hat{\underline{t}}_s + i} \right) + 2\hat{\epsilon}_{i,\underline{t}_s} \right]_{i \in \hat{\mathbb{M}}} \\
\left[ 2\left( \frac{1+\sqrt{\breve{\zeta}_{j,s\uparrow}}}{2} \sqrt{\breve{\bar{\chi}}_{j,s}} + \frac{\breve{\epsilon}_{j,\underline{t}_s}}{\breve{\bar{\sigma}}}\sqrt{\underline{t}_s} \right) \breve{\bar{\sigma}}_{j,s-1}\left( \mathbf{x}_{\breve{\underline{t}}_s + j} \right) + 2\breve{\epsilon}_{j,\underline{t}_s} \right]_{i \in \hat{\mathbb{M}}}
\end{array} \right]
$$

*and:*

$$
\begin{aligned}
\bar{\sigma}_{s-1\uparrow} &= \max\left\{ \max_{i \in \hat{\mathbb{M}}} \left\{ \hat{\bar{\sigma}}_{i,s-1}(\mathbf{x}^\star) \right\}, \max_{j \in \breve{\mathbb{M}}} \left\{ \breve{\bar{\sigma}}_{j,s-1}(\mathbf{x}^\star) \right\} \right\} \\
\bar{\sigma}_{s-1\downarrow} &= \min\left\{ \min_{i \in \hat{\mathbb{M}}} \left\{ \hat{\bar{\sigma}}_{i,s-1}(\mathbf{x}^\star) \right\}, \min_{j \in \breve{\mathbb{M}}} \left\{ \breve{\bar{\sigma}}_{j,s-1}(\mathbf{x}^\star) \right\} \right\}
\end{aligned}
$$

*and $\exp$ is applied elementwise.*

---

[5]The proof is true for $\bar{\theta}_s \in [-\infty, \infty]$, but the negative $\bar{\theta}_s$ case is not of interest here.

*Proof.* It is convenient to re-frame the batch-wise instantaneous regret in log-space, and extract the max variance upper bound. Recalling that $\hat{\breve{\zeta}}_{i,s\downarrow} \leq 1, \breve{\zeta}_{j,s\downarrow} \geq 1$:

$$\bar{r}_s \leq \min \left\{ \min_{i \in \hat{\mathbb{M}}} \left\{ \begin{array}{c} 2\left(\frac{1+\sqrt{\hat{\breve{\zeta}}_{j,s\uparrow}}}{2}\sqrt{\hat{\breve{\chi}}_{i,s}} + \frac{\hat{\epsilon}_{i,\underline{t}_s}}{\hat{\sigma}}\sqrt{\underline{t}_s}\right)\hat{\sigma}_{i,s-1}\left(\mathbf{x}_{\underline{\hat{t}}_s+i}\right) + \dots \\ \left(1-\sqrt{\hat{\breve{\zeta}}_{i,s\downarrow}}\right)\sqrt{\hat{\breve{\chi}}_{i,s}}\hat{\sigma}_{i,s-1}\left(\mathbf{x}^\star\right) + 2\hat{\epsilon}_{i,\underline{t}_s} \end{array} \right\}, \right.$$

$$\left. \min_{j \in \breve{\mathbb{M}}} \left\{ \begin{array}{c} 2\left(\frac{1+\sqrt{\breve{\zeta}_{j,s\uparrow}}}{2}\sqrt{\breve{\chi}_{j,s}} + \frac{\breve{\epsilon}_{j,\underline{t}_s}}{\breve{\sigma}}\sqrt{\underline{t}_s}\right)\breve{\sigma}_{j,s-1}\left(\mathbf{x}_{\underline{t}_s+j}\right) + \dots \\ \left(1-\sqrt{\breve{\zeta}_{j,s\downarrow}}\right)\sqrt{\breve{\chi}_{j,s}}\breve{\sigma}_{j,s-1}\left(\mathbf{x}^\star\right) + 2\breve{\epsilon}_{j,\underline{t}_s} \end{array} \right\} \right\}$$

$$\leq \ln\left( \min \left\{ \min_{i \in \hat{\mathbb{M}}} \left\{ \begin{array}{c} \exp\left(2\left(\frac{1+\sqrt{\hat{\breve{\zeta}}_{i,s\uparrow}}}{2}\sqrt{\hat{\breve{\chi}}_{i,s}} + \frac{\hat{\epsilon}_{i,\underline{t}_s}}{\hat{\sigma}}\sqrt{\underline{t}_s}\right)\hat{\sigma}_{i,s-1}\left(\mathbf{x}_{\underline{\hat{t}}_s+i}\right) + 2\hat{\epsilon}_{i,\underline{t}_s}\right)\dots \\ \exp\left(\sqrt{\hat{\breve{\chi}}_{i,s}}\left(1-\sqrt{\hat{\breve{\zeta}}_{i,s\downarrow}}\right)_+ \bar{\sigma}_{s-1\uparrow}\right) \end{array} \right\}, \right. \right.$$

$$\left. \left. \min_{j \in \breve{\mathbb{M}}} \left\{ \begin{array}{c} \exp\left(2\left(\frac{1+\sqrt{\breve{\zeta}_{j,s\uparrow}}}{2}\sqrt{\breve{\chi}_{j,s}} + \frac{\breve{\epsilon}_{j,\underline{t}_s}}{\breve{\sigma}}\sqrt{\underline{t}_s}\right)\breve{\sigma}_{j,s-1}\left(\mathbf{x}_{\underline{t}_s+j}\right) + 2\breve{\epsilon}_{j,\underline{t}_s}\right)\dots \\ \exp\left(\sqrt{\breve{\chi}_{j,s}}\left(1-\sqrt{\breve{\zeta}_{j,s\downarrow}}\right)_- \bar{\sigma}_{s-1\downarrow}\right) \end{array} \right\} \right\} \right)$$

which may be re-written:

$$\bar{r}_s \leq \ln\left(M_{-\infty}\left(\mathbf{a}\odot\mathbf{b}\right)\right)$$

where:

$$\mathbf{a} = \left[\begin{array}{c} \hat{\mathbf{a}} \\ \breve{\mathbf{a}} \end{array}\right] \in \mathbb{R}_+^p, \quad \mathbf{b} = \left[\begin{array}{c} \hat{\mathbf{b}} \\ \breve{\mathbf{b}} \end{array}\right] \in \mathbb{R}_+^p$$

$$\hat{\mathbf{a}} = \left[\exp\left(2\left(\frac{1+\sqrt{\hat{\breve{\zeta}}_{i,s\uparrow}}}{2}\sqrt{\hat{\breve{\chi}}_{i,s}} + \frac{\hat{\epsilon}_{i,\underline{t}_s}}{\hat{\sigma}}\sqrt{\underline{t}_s}\right)\hat{\sigma}_{i,s-1}\left(\mathbf{x}_{\underline{\hat{t}}_s+i}\right) + 2\hat{\epsilon}_{i,\underline{t}_s}\right)\right]_{i \in \hat{\mathbb{M}}}$$

$$\hat{\mathbf{b}} = \left[\exp\left(\sqrt{\hat{\breve{\chi}}_{i,s}}\left(1-\sqrt{\hat{\breve{\zeta}}_{i,s\downarrow}}\right)_+ \bar{\sigma}_{s-1\uparrow}\right)\right]_{i \in \hat{\mathbb{M}}}$$

$$\breve{\mathbf{a}} = \left[\exp\left(2\left(\frac{1+\sqrt{\breve{\zeta}_{j,s\uparrow}}}{2}\sqrt{\breve{\chi}_{j,s}} + \frac{\breve{\epsilon}_{j,\underline{t}_s}}{\breve{\sigma}}\sqrt{\underline{t}_s}\right)\breve{\sigma}_{j,s-1}\left(\mathbf{x}_{\underline{t}_s+j}\right) + 2\breve{\epsilon}_{j,\underline{t}_s}\right)\right]_{j \in \breve{\mathbb{M}}}$$

$$\breve{\mathbf{b}} = \left[\exp\left(\sqrt{\breve{\chi}_{j,s}}\left(1-\sqrt{\breve{\zeta}_{j,s\downarrow}}\right)_- \bar{\sigma}_{s-1\downarrow}\right)\right]_{j \in \breve{\mathbb{M}}}$$

So, by lemma 2, we have that:

$$\bar{r}_s \leq \ln\left(M_{-\bar{\theta}_s}\left(\mathbf{a}\right)M_{\bar{\theta}_s}\left(\mathbf{b}\right)\right) = \ln\left(M_{-\bar{\theta}_s}\left(\mathbf{a}\right)\right) + \ln\left(M_{\bar{\theta}_s}\left(\mathbf{b}\right)\right)$$

and by assumption $M_{\bar{\theta}_s}\left(\mathbf{b}\right) \leq 1$, so:

$$\bar{r}_s \leq \ln\left(M_{-\bar{\theta}_s}\left(\mathbf{a}\right)\right)$$

and the result follows by the definition. $\square$

The next step is to convert this bound into a bound on the total regret. To obtain such a bound we need some additional assumptions regarding "gap" parameters $\epsilon$ and the explorative/exploitative nature of the humans and AIs. We do this with the following theorem:

**Theorem 6.** *Fix $\delta > 0$ and $\bar{\theta}_s \in [0,\infty]$.[6] Assume noise variables are $\Sigma$-sub-Gaussian, and that:*

$$\hat{\beta}_{i,\underline{t}_s} \in \left(\hat{\breve{\zeta}}_{i,s\downarrow}\hat{\breve{\chi}}_{i,s}, \hat{\breve{\zeta}}_{i,s\uparrow}\hat{\breve{\chi}}_{i,s}\right)$$

$$\breve{\beta}_{j,\underline{t}_s} = \left(\breve{\zeta}_{j,s\downarrow}\breve{\chi}_{j,s}, \breve{\zeta}_{j,s\uparrow}\breve{\chi}_{j,s}\right)$$

---

[6]The proof is true for $\bar{\theta}_s \in [-\infty,\infty]$, but the negative $\bar{\theta}_s$ case is not of interest here.

*where* $\hat{\breve{\zeta}}_{i,s\downarrow} \leq 1$, $\breve{\zeta}_{j,s\downarrow} \geq 1$, *and:*

$$\hat{\breve{\chi}}_{i,s} = \left( \frac{\Sigma}{\sqrt{\hat{\sigma}}} \sqrt{2\ln\left(\frac{1}{\delta}\right) + 1} + \hat{\breve{\gamma}}_{i,s} + \left\| \hat{\breve{f}}_{i,s}^{\#} \right\|_{\hat{\mathcal{H}}_{i,s}} \right)^2$$

$$\breve{\chi}_{j,s} = \left( \frac{\Sigma}{\sqrt{\breve{\sigma}}} \sqrt{2\ln\left(\frac{1}{\delta}\right) + 1} + \breve{\gamma}_{j,s} + \left\| \breve{f}_{j,s}^{\#} \right\|_{\breve{\mathcal{H}}_{j,s}} \right)^2$$

*and:*

$$M_{\bar{\theta}_s} \left( \begin{array}{c} \left\{ \exp\left( \sqrt{\hat{\breve{\chi}}_{i,s}} \left( 1 - \sqrt{\hat{\breve{\zeta}}_{i,s\downarrow}} \right)_+ \bar{\sigma}_{s-1\uparrow} \right) : i \in \hat{\mathbb{M}} \right\} \cup \ldots \\ \left\{ \exp\left( \sqrt{\breve{\chi}}_{j,s} \left( 1 - \sqrt{\breve{\zeta}}_{j,s\downarrow} \right)_- \bar{\sigma}_{s-1\downarrow} \right) : j \in \breve{\mathbb{M}} \right\} \end{array} \right) \leq 1$$

*for all s, where:*

$$\bar{\sigma}_{s-1\uparrow} = \max\left\{ \max_{i\in\hat{\mathbb{M}}} \left\{ \hat{\bar{\sigma}}_{i,s-1}(\mathbf{x}^\star) \right\}, \max_{j\in\breve{\mathbb{M}}} \left\{ \breve{\bar{\sigma}}_{j,s-1}(\mathbf{x}^\star) \right\} \right\}$$

$$\bar{\sigma}_{s-1\downarrow} = \min\left\{ \min_{i\in\hat{\mathbb{M}}} \left\{ \hat{\bar{\sigma}}_{i,s-1}(\mathbf{x}^\star) \right\}, \min_{j\in\breve{\mathbb{M}}} \left\{ \breve{\bar{\sigma}}_{j,s-1}(\mathbf{x}^\star) \right\} \right\}$$

*If* $\hat{\epsilon}_{i,\underline{t}_s}, \breve{\epsilon}_{j,\underline{t}_s} = \mathcal{O}(s^{-q})$ *for some* $q \in (\frac{1}{2}, \infty)$ *then, with probability* $\geq 1 - \delta$*:*

$$\frac{1}{S}\bar{R}_S \leq \ln\left( M_{-\bar{\theta}_\downarrow} \left( \exp\left( \left\{ \left\{ \sqrt{8\hat{\sigma}^2 \left( \sqrt{\hat{\breve{\chi}}_{i,S}} + \mathcal{O}(1) \right)^2 \hat{C}_{i,2} \frac{1}{S}\hat{\breve{\gamma}}_{i,S} + \mathcal{O}(1)} \middle| i \in \hat{\mathbb{M}} \right\} \cup \ldots \right.\right.\right.$$
$$\left.\left.\left. \left\{ \sqrt{8\breve{\sigma}^2 \left( \frac{1+\sqrt{\breve{\zeta}_{j\sim}}}{2} \sqrt{\breve{\chi}_{j,S}} + \mathcal{O}(1) \right)^2 \breve{C}_{j,2} \frac{1}{S}\breve{\gamma}_{j,S} + \mathcal{O}(1)} \middle| j \in \breve{\mathbb{M}} \right\} \right\} \right) \right) \right)$$

*where* $\hat{C}_{i,2} = \max_s \frac{\hat{\sigma}^{-2}\hat{K}_{i,s\uparrow}}{\ln\left(1+\hat{\sigma}^{-2}\hat{K}_{i,s\uparrow}\right)}$ *and* $\breve{C}_{j,2} = \max_s \frac{\breve{\sigma}^{-2}\breve{K}_{j,s\uparrow}}{\ln\left(1+\breve{\sigma}^{-2}\breve{K}_{j,\}s\uparrow}\right)}$*; and:*

$$\bar{\theta}_\downarrow = \min_{s\in\mathbb{N}_S+1} \bar{\theta}_s$$
$$\breve{\zeta}_{j\sim} = \max_{s\in\mathbb{N}_S+1} \breve{\zeta}_{j,s}$$

*Proof.* Using the assumptions and Lemma 5, we have that, with probability $\geq 1-\delta$, simultaneously for all $s \geq 1$:

$$\bar{r}_s \leq \ln\left( M_{-\theta_s}\left( \exp(\bar{\mathbf{r}}_{s\uparrow}) \right) \right) \leq \ln\left( M_{-\bar{\theta}_s}\left( \exp(\bar{\mathbf{r}}_{s\uparrow}) \right) \right)$$

where $\bar{\mathbf{r}}_{s\uparrow} \geq \mathbf{0}$. Using the definition of $M_{-\bar{\theta}_s}$:

$$\sum_{s\in\mathbb{N}_S+1} \bar{r}_s \leq \sum_{s\in\mathbb{N}_S+1} \ln\left( M_{-\bar{\theta}_\downarrow}\left( \exp(\bar{\mathbf{r}}_{s\uparrow}) \right) \right)$$

$$= \ln\left( \prod_{s\in\mathbb{N}_S+1} M_{-\bar{\theta}_\downarrow}\left( \exp(\bar{\mathbf{r}}_{s\uparrow}) \right) \right)$$

$$= \ln\left( \prod_{s\in\mathbb{N}_S+1} \left( \frac{1}{p} \left\| \exp(\bar{\mathbf{r}}_{s\uparrow}) \right\|_{-\bar{\theta}_\downarrow} \right)^{-\frac{1}{\bar{\theta}_\downarrow}} \right)$$

$$= \ln\left( \left( \prod_{s\in\mathbb{N}_S+1} p^{\frac{S}{\bar{\theta}_\downarrow}} \left\| (\exp(\bar{\mathbf{r}}_{s\uparrow}))^{\odot-\bar{\theta}_\downarrow} \right\|_1^{-\frac{1}{\bar{\theta}_\downarrow}} \right) \right)$$

$$= \ln\left( \left( \prod_{s\in\mathbb{N}_S+1} p^{\frac{S}{\bar{\theta}_\downarrow}} \left( \left\| \left( (\exp(\bar{\mathbf{r}}_{s\uparrow}))^{\odot-\frac{\bar{\theta}_\downarrow}{S}} \right)^{\odot S} \right\|_1^{\frac{1}{S}} \right)^{-\frac{S}{\bar{\theta}_\downarrow}} \right) \right)$$

$$= \ln\left( \left( \prod_{s\in\mathbb{N}_S+1} p^{\frac{S}{\bar{\theta}_\downarrow}} \left\| (\exp(\bar{\mathbf{r}}_{s\uparrow}))^{\odot-\frac{\bar{\theta}_\downarrow}{S}} \right\|_S^{-\frac{S}{\bar{\theta}_\downarrow}} \right) \right)$$

Using the generalised Hölder inequality and the definition of $M_{-\bar{\theta}_\downarrow}$:

$$
\begin{aligned}
\sum_{s\in\mathbb{N}_S+1} \bar{r}_s &\leq \ln\left(\left(\prod_{s\in\mathbb{N}_S+1} p^{\frac{S}{\bar{\theta}_\downarrow}} \left\|(\exp(\bar{\mathbf{r}}_{s\uparrow}))^{\odot-\frac{\bar{\theta}_\downarrow}{S}}\right\|_S\right)^{-\frac{S}{\bar{\theta}_\downarrow}}\right)\\
&\leq \ln\left(\left(\left\|\bigodot_{s\in\mathbb{N}_S+1} p^{\frac{S}{\bar{\theta}_\downarrow}}\left(\exp(\bar{\mathbf{r}}_{s\uparrow})\right)^{\odot-\frac{\bar{\theta}_\downarrow}{S}}\right\|_1\right)^{-\frac{S}{\bar{\theta}_\downarrow}}\right)\\
&= \ln\left(\left(\left(\frac{1}{p}\left\|\left(\bigodot_{s\in\mathbb{N}_S+1}\exp(\bar{\mathbf{r}}_{s\uparrow})\right)^{\odot-\frac{\bar{\theta}_\downarrow}{S}}\right\|_1\right)^{-\frac{S}{\bar{\theta}_\downarrow}}\right)\right)\\
&= \ln\left(M_{-\frac{\bar{\theta}_\downarrow}{S}}\left(\bigodot_{s\in\mathbb{N}_S+1}\exp(\bar{\mathbf{r}}_{s\uparrow})\right)\right)
\end{aligned}
$$

and so, again recalling that $\bar{\mathbf{r}}_{s\uparrow}\geq\mathbf{0}$:

$$
\begin{aligned}
\sum_{s\in\mathbb{N}_S+1} \bar{r}_s &\leq \ln\left(M_{-\frac{\bar{\theta}_\downarrow}{S}}\left(\bigodot_{s\in\mathbb{N}_S+1}\exp(\bar{\mathbf{r}}_{s\uparrow})\right)\right)\\
&= \ln\left(M_{-\frac{\bar{\theta}_\downarrow}{S}}\left(\exp\left(\sum_{s\in\mathbb{N}_S+1}\bar{\mathbf{r}}_{s\uparrow}\right)\right)\right)
\end{aligned}
$$

Next, using that $\chi$ is increasing with $s$, and noting our restricted range in $\zeta$, note that:

$$
\begin{aligned}
\hat{\bar{r}}_{i,s\uparrow}^2 &\leq 4\left(\left(\sqrt{\hat{\bar{\chi}}_{i,S}}+\frac{\hat{\epsilon}_{i,t_s}}{\hat{\bar{\sigma}}}\sqrt{\hat{t}_s}\right)\hat{\bar{\sigma}}_{i,s-1}\left(\mathbf{x}_{\hat{t}_s+i}\right)+\hat{\epsilon}_{i,t_s}\right)^2\\
&\leq 8\max\left\{\left(\sqrt{\hat{\bar{\chi}}_{i,S}}+\frac{\hat{\epsilon}_{i,t_s}}{\hat{\bar{\sigma}}}\sqrt{\hat{t}_s}\right)^2\hat{\bar{\sigma}}_{i,s-1}^2\left(\mathbf{x}_{\hat{t}_s+i}\right),\hat{\epsilon}_{i,t_s}^2\right\}\\
\breve{\bar{r}}_{j,s\uparrow}^2 &\leq 4\left(\left(\frac{1+\sqrt{\breve{\bar{\zeta}}_{j\sim}}}{2}\sqrt{\breve{\bar{\chi}}_{j,S}}+\frac{\breve{\epsilon}_{j,t_s}}{\breve{\bar{\sigma}}}\sqrt{\breve{t}_s}\right)\breve{\bar{\sigma}}_{j,s-1}\left(\mathbf{x}_{\breve{t}_s+j}\right)+\breve{\epsilon}_{j,t_s}\right)^2\\
&\leq 8\max\left\{\left(\frac{1+\sqrt{\breve{\bar{\zeta}}_{j\sim}}}{2}\sqrt{\breve{\bar{\chi}}_{j,S}}+\frac{\breve{\epsilon}_{j,t_s}}{\breve{\bar{\sigma}}}\sqrt{\breve{t}_s}\right)^2\breve{\bar{\sigma}}_{j,s-1}^2\left(\mathbf{x}_{\breve{t}_s+j}\right),\breve{\epsilon}_{j,t_s}^2\right\}
\end{aligned}
$$

and so:

$$
\begin{aligned}
\sum_{s\in\mathbb{N}_S+1}\hat{\bar{r}}_{i,s\uparrow}^2 &\leq 8\sum_{s\in\mathbb{N}_S+1}\max\left\{\left(\sqrt{\hat{\bar{\chi}}_{i,S}}+\frac{\hat{\epsilon}_{i,t_s}}{\hat{\bar{\sigma}}}\sqrt{\hat{t}_s}\right)^2\hat{\bar{\sigma}}_{i,s-1}^2\left(\mathbf{x}_{\hat{t}_s+i}\right),\hat{\epsilon}_{i,t_s}^2\right\}\\
&\leq 8\sum_{s\in\mathbb{N}_S+1}\left(\sqrt{\hat{\bar{\chi}}_{i,S}}+\frac{\hat{\epsilon}_{i,t_s}}{\hat{\bar{\sigma}}}\sqrt{\hat{t}_s}\right)^2\hat{\bar{\sigma}}_{i,s-1}^2\left(\mathbf{x}_{\hat{t}_s+i}\right)+8\sum_{s\in\mathbb{N}_S+1}\hat{\epsilon}_{i,t_s}^2\\
\sum_{s\in\mathbb{N}_S+1}\breve{\bar{r}}_{j,s\uparrow}^2 &\leq 8\sum_{s\in\mathbb{N}_S+1}\max\left\{\left(\frac{1+\sqrt{\breve{\bar{\zeta}}_{j\sim}}}{2}\sqrt{\breve{\bar{\chi}}_{j,S}}+\frac{\breve{\epsilon}_{j,t_s}}{\breve{\bar{\sigma}}}\sqrt{\breve{t}_s}\right)^2\breve{\bar{\sigma}}_{j,s-1}^2\left(\mathbf{x}_{\breve{t}_s+j}\right),\breve{\epsilon}_{j,t_s}^2\right\}\\
&\leq 8\sum_{s\in\mathbb{N}_S+1}\left(\frac{1+\sqrt{\breve{\bar{\zeta}}_{j\sim}}}{2}\sqrt{\breve{\bar{\chi}}_{j,S}}+\frac{\breve{\epsilon}_{j,t_s}}{\breve{\bar{\sigma}}}\sqrt{\breve{t}_s}\right)^2\breve{\bar{\sigma}}_{j,s-1}^2\left(\mathbf{x}_{\breve{t}_s+j}\right)+8\sum_{s\in\mathbb{N}_S+1}\breve{\epsilon}_{j,t_s}^2
\end{aligned}
$$

Recalling our assumption $\hat{\epsilon}_{i,t_s},\breve{\epsilon}_{j,t_s}=\mathcal{O}(s^{-q})$ for some $q>\frac{1}{2}$ we find that, as $\sum_{s\in\mathbb{N}}\frac{1}{s^{2\hat{q}_i}}=\zeta(2\hat{q}_i)<\infty$ ($\zeta$ here is the Reimann zeta function):

$$
\begin{aligned}
\sum_{s\in\mathbb{N}_S+1}\hat{\bar{r}}_{i,s\uparrow}^2 &\leq 8\sum_{s\in\mathbb{N}_S+1}\left(\sqrt{\hat{\bar{\chi}}_{i,S}}+\mathcal{O}(1)\right)^2\hat{\bar{\sigma}}_{i,s-1}^2\left(\mathbf{x}_{\hat{t}_s+i}\right)+\mathcal{O}(1)\\
\sum_{s\in\mathbb{N}_S+1}\breve{\bar{r}}_{j,s\uparrow}^2 &\leq 8\sum_{s\in\mathbb{N}_S+1}\left(\frac{1+\sqrt{\breve{\bar{\zeta}}_{j\sim}}}{2}\sqrt{\breve{\bar{\chi}}_{j,S}}+\mathcal{O}(1)\right)^2\breve{\bar{\sigma}}_{j,s-1}^2\left(\mathbf{x}_{\breve{t}_s+j}\right)+\mathcal{O}(1)
\end{aligned}
$$

Now, by the standard procedure:

$$
\begin{aligned}
\sum_{s\in\mathbb{N}_S+1}\hat{\sigma}^{-2}\hat{\sigma}^2_{i,s-1}\left(\mathbf{x}_{\underline{t}_s+i}\right) &\le \hat{C}_{i,2}\sum_{s\in\mathbb{N}_S+1}\ln\left(1+\hat{\sigma}^{-2}\hat{\sigma}^2_{i,s-1}\left(\mathbf{x}_{\underline{t}_s+i}\right)\right)\\
&\le \hat{C}_{i,2}\sum_{t\in\mathbb{T}_{\le S}}\ln\left(1+\hat{\sigma}^{-2}\hat{\sigma}^2_{i,s_t-1}\left(\mathbf{x}_t\right)\right)\\
&\le \hat{C}_{i,2}\hat{\gamma}_{i,S}\\
\sum_{s\in\mathbb{N}_S+1}\breve{\sigma}^{-2}\breve{\sigma}^2_{j,s-1}\left(\mathbf{x}_{\underline{t}_s+j}\right) &\le \breve{C}_{j,2}\sum_{s\in\mathbb{N}_S+1}\ln\left(1+\breve{\sigma}^{-2}\breve{\sigma}^2_{j,s-1}\left(\mathbf{x}_{\underline{t}_s+j}\right)\right)\\
&\le \breve{C}_{j,2}\sum_{t\in\mathbb{T}_{\le S}}\ln\left(1+\breve{\sigma}^{-2}\breve{\sigma}^2_{j,s_t-1}\left(\mathbf{x}_t\right)\right)\\
&\le \breve{C}_{j,2}\breve{\gamma}_{j,S}
\end{aligned}
$$

where:

$$
\begin{aligned}
\hat{C}_{i,2} &= \max_{s\in\mathbb{N}_S+1}\frac{\hat{\sigma}^{-2}\hat{K}_{i,s\uparrow}}{\ln\left(1+\hat{\sigma}^{-2}\hat{K}_{i,s\uparrow}\right)}\\
\breve{C}_{j,2} &= \max_{s\in\mathbb{N}_S+1}\frac{\breve{\sigma}^{-2}\breve{K}_{j,s\uparrow}}{\ln\left(1+\breve{\sigma}^{-2}\breve{K}_{j,s\uparrow}\right)}
\end{aligned}
$$

and so:

$$
\begin{aligned}
\sum_{s\in\mathbb{N}_S+1}\hat{r}^2_{i,s\uparrow} &\le 8\hat{\sigma}^2\left(\sqrt{\hat{\chi}_{i,S}}+\mathcal{O}\left(1\right)\right)^2\hat{C}_{i,2}\hat{\gamma}_{i,S}+\mathcal{O}\left(1\right)\\
\sum_{s\in\mathbb{N}_S+1}\breve{r}^2_{j,s\uparrow} &\le 8\breve{\sigma}^2\left(\frac{1+\sqrt{\breve{\zeta}_{j\sim}}}{2}\sqrt{\breve{\chi}_{j,S}}+\mathcal{O}\left(1\right)\right)^2\breve{C}_{j,2}\breve{\gamma}_{j,S}+\mathcal{O}\left(1\right)
\end{aligned}
$$

Recalling that $\|\cdot\|_1\le\sqrt{S}\|\cdot\|_2$ (in $S$ dimensions):

$$
\begin{aligned}
\sum_{s\in\mathbb{N}_S+1}\hat{r}_{i,s\uparrow} &\le \sqrt{S}\sqrt{8\hat{\sigma}^2\left(\sqrt{\hat{\chi}_{i,S}}+\mathcal{O}\left(1\right)\right)^2\hat{C}_{i,2}\hat{\gamma}_{i,S}+\mathcal{O}\left(1\right)}\\
\sum_{s\in\mathbb{N}_S+1}\breve{r}_{j,s\uparrow} &\le \sqrt{S}\sqrt{18\breve{\sigma}^2\left(\frac{1+\sqrt{\breve{\zeta}_{j\sim}}}{2}\sqrt{\breve{\chi}_{j,S}}+\mathcal{O}\left(1\right)\right)^2\breve{C}_{j,2}\breve{\gamma}_{j,S}+\mathcal{O}\left(1\right)}
\end{aligned}
$$

and so:

$$
\begin{aligned}
&\sum_{s\in\mathbb{N}_S+1}\hat{r}_{i,s\uparrow}+\sum_{s\in\mathbb{N}_S+1}\breve{r}_{j,s\uparrow}\ldots\\
&\le \ln\left(M_{-\frac{\bar{\theta}_\downarrow}{S}}\left(\exp\left(\left\{\left\{\sqrt{S}\sqrt{8\hat{\sigma}^2\left(\mathcal{O}\left(1\right)+\sqrt{\hat{\chi}_{i,S}}\right)^2\hat{C}_{i,2}\hat{\gamma}_{i,S}+\mathcal{O}\left(1\right)}\Big| i\in\hat{\mathbb{M}}\right\}\cup\ldots\right.\right.\right.\right.\\
&\qquad\left.\left.\left.\left.\left\{\sqrt{S}\sqrt{8\breve{\sigma}^2\left(\mathcal{O}\left(1\right)+\frac{1+\sqrt{\breve{\zeta}_{j\sim}}}{2}\sqrt{\breve{\chi}_{j,S}}\right)^2\breve{C}_{j,2}\breve{\gamma}_{j,S}+\mathcal{O}\left(1\right)}\Big| j\in\breve{\mathbb{M}}\right\}\right\}\right)\right)\right)
\end{aligned}
$$

Finally, noting that, for $\mathbf{a}>\mathbf{0}$:

$$
\begin{aligned}
\tfrac{1}{S}\ln\left(M_{-\frac{\bar{\theta}_\downarrow}{S}}\left(\exp\left(\mathbf{a}\right)\right)\right) &= \ln\left(M^{\frac{1}{S}}_{-\frac{\bar{\theta}_\downarrow}{S}}\left(\exp\left(\mathbf{a}\right)\right)\right)\\
&= \ln\left(\left(\tfrac{1}{p}\sum_i\left(e^{a_i}\right)^{-\frac{\bar{\theta}_\downarrow}{S}}\right)^{-\frac{1}{\bar{\theta}_\downarrow}}\right)\\
&= \ln\left(\left(\tfrac{1}{p}\sum_i\left(e^{\frac{1}{S}a_i}\right)^{-\bar{\theta}_\downarrow}\right)^{-\frac{1}{\bar{\theta}_\downarrow}}\right)\\
&= \ln\left(M_{-\bar{\theta}_\downarrow}\left(\exp\left(\tfrac{1}{S}\mathbf{a}\right)\right)\right)
\end{aligned}
$$

we obtain:

$$\frac{1}{S}\left(\sum_{s\in\mathbb{N}_S+1}\hat{\tilde{r}}_{i,s\uparrow}+\sum_{s\in\mathbb{N}_S+1}\check{\tilde{r}}_{j,s\uparrow}\right)\dots$$

$$\leq\ln\left(M_{-\bar{\theta}_\downarrow}\left(\exp\left(\left\{\left\{\frac{1}{\sqrt{S}}\sqrt{8\hat{\tilde{\sigma}}^2\left(\mathcal{O}\left(1\right)+\sqrt{\hat{\tilde{\chi}}_{i,S}}\right)^2\hat{C}_{i,2}\hat{\tilde{\gamma}}_{i,S}+\mathcal{O}\left(1\right)}\middle|\,i\in\hat{\mathbb{M}}\right\}\bigcup\dots\right.\right.\right.$$

$$\left.\left.\left.\left\{\frac{1}{\sqrt{S}}\sqrt{8\check{\tilde{\sigma}}^2\left(\mathcal{O}\left(1\right)+\frac{1+\sqrt{\check{\tilde{\zeta}}_{j\sim}}}{2}\sqrt{\check{\tilde{\chi}}_{j,S}}\right)^2\check{C}_{j,2}\check{\tilde{\gamma}}_{j,S}+\mathcal{O}\left(1\right)}\middle|\,j\in\check{\mathbb{M}}\right\}\right\}\right)\right)\right)$$

as required. □

Finally, we consider the bounds on the posterior variance at $\mathbf{x}^\star$. We have the following result:

**Lemma 7.** *Fix $j'\in\check{\mathbb{M}}$. We have the bounds for all $i\in\hat{\mathbb{M}}$, $j\in\check{\mathbb{M}}$:*

$$\hat{\tilde{\sigma}}^\star_{i,s\downarrow}\ \leq\hat{\tilde{\sigma}}_{i,s}\left(\mathbf{x}^\star\right)\ \leq\hat{\tilde{\sigma}}^\star_{i,s\uparrow}$$
$$\check{\tilde{\sigma}}^\star_{j,s\downarrow}\ \leq\check{\tilde{\sigma}}_{j,s}\left(\mathbf{x}^\star\right)\ \leq\check{\tilde{\sigma}}^\star_{j,s\uparrow}$$

*where:*

$$\hat{\tilde{\sigma}}^{\star2}_{i,s\downarrow}\ =\frac{\hat{\tilde{\sigma}}^2+2ps\hat{\tilde{D}}_{i,s,\downarrow}-ps\frac{1}{\hat{\tilde{K}}^{\star\star}_{i,s}}\hat{\tilde{D}}^2_{i,s,\downarrow}}{\hat{\tilde{\sigma}}^2+ps\hat{\tilde{K}}^{\star\star}_{i,s}}\hat{\tilde{K}}^{\star\star}_{i,s}$$

$$\check{\tilde{\sigma}}^{\star2}_{j,s\downarrow}\ =\frac{\check{\tilde{\sigma}}^2+2ps\check{\tilde{D}}_{j,s,\downarrow}-ps\frac{1}{\check{\tilde{K}}^{\star\star}_{j,s}}\check{\tilde{D}}^2_{j,s,\downarrow}}{\check{\tilde{\sigma}}^2+ps\check{\tilde{K}}^{\star\star}_{j,s}}\check{\tilde{K}}^{\star\star}_{j,s}$$

$$\hat{\tilde{\sigma}}^{\star2}_{i,s\uparrow}\ =\frac{\hat{\tilde{\sigma}}^2+2\sum_{t\in\mathbb{T}_{\leq s}}\hat{\tilde{D}}_{i,s,t}-\frac{1}{\hat{\tilde{K}}^{\star\star}_{i,s}}\sum_{t\in\mathbb{T}_{\leq s}}\hat{\tilde{D}}^2_{i,s,t}}{\hat{\tilde{\sigma}}^2+ps\hat{\tilde{K}}^{\star\star}_{i,s}}\hat{\tilde{K}}^{\star\star}_{i,s}$$

$$\check{\tilde{\sigma}}^{\star2}_{j,s\uparrow}\ =\frac{\check{\tilde{\sigma}}^2+2\sum_{t\in\mathbb{T}_{\leq s}}\check{\tilde{D}}_{j,s,t}-\frac{1}{\check{\tilde{K}}^{\star\star}_{j,s}}\sum_{t\in\mathbb{T}_{\leq s}}\check{\tilde{D}}^2_{j,s,t}}{\check{\tilde{\sigma}}^2+ps\check{\tilde{K}}^{\star\star}_{j,s}}\check{\tilde{K}}^{\star\star}_{j,s}$$

*and:*

$$\hat{\tilde{K}}^{\star\star}_{i,s}=\hat{\tilde{K}}_{i,s}\left(\mathbf{x}^\star,\mathbf{x}^\star\right),\qquad\check{\tilde{K}}^{\star\star}_{j,s}=\check{\tilde{K}}_{j,s}\left(\mathbf{x}^\star,\mathbf{x}^\star\right)$$
$$\hat{\tilde{K}}_{i,s,t,\star}=\hat{\tilde{K}}_{i,s}\left(\mathbf{x}^\star,\mathbf{x}_t\right),\quad\check{\tilde{K}}_{j,s,t,\star}=\check{\tilde{K}}_{j,s}\left(\mathbf{x}^\star,\mathbf{x}_t\right)$$
$$\hat{\tilde{K}}_{i,s,t,t}=\hat{\tilde{K}}_{i,s}\left(\mathbf{x}_t,\mathbf{x}_t\right),\quad\check{\tilde{K}}_{j,s,t,t}=\check{\tilde{K}}_{j,s}\left(\mathbf{x}_t,\mathbf{x}_t\right)$$

*and:*

$$\left|\hat{\tilde{K}}_{i,s,t,t}\right|=\hat{\tilde{K}}^{\star\star}_{i,s}-\hat{\tilde{D}}_{i,s,t},\qquad\left|\check{\tilde{K}}_{j,s,t,t}\right|=\check{\tilde{K}}^{\star\star}_{j,s}-\check{\tilde{D}}_{j,s,t}$$
$$\hat{\tilde{D}}_{i,s,\downarrow}=\min_{t\in\mathbb{T}_{\leq s}}\left\{\hat{\tilde{D}}_{i,s,t}\right\},\qquad\check{\tilde{D}}_{i,s,\downarrow}=\min_{t\in\mathbb{T}_{\leq s}}\left\{\check{\tilde{D}}_{i,s,t}\right\}$$

*Proof.* Using the definition of GP posterior variance and $K$ bounds we can minimise the posterior variance within the constraints given as:

$$\hat{\tilde{\sigma}}^2_{i,s}\left(\mathbf{x}^\star\right)\geq\hat{\tilde{K}}^{\star\star}_{i,s}-\hat{\tilde{K}}^2_{i,s,\uparrow,\star}\mathbf{1}^{\mathrm{T}}_{ps}\left(\hat{\tilde{K}}_{i,s\uparrow}\mathbf{1}_{ps}\mathbf{1}^{\mathrm{T}}_{ps}+\hat{\tilde{\sigma}}^2\mathbf{I}_{ps}\right)^{-1}\mathbf{1}_{ps}$$

$$=\hat{\tilde{K}}^{\star\star}_{i,s}\left(1-\frac{\hat{\tilde{K}}^2_{i,s,\uparrow,\star}}{\hat{\tilde{K}}^{\star\star}_{i,s}}\frac{ps}{\hat{\tilde{\sigma}}^2+ps\hat{\tilde{K}}_{i,s\uparrow}}\right)$$

$$=\hat{\tilde{K}}^{\star\star}_{i,s}\left(1-\frac{ps\hat{\tilde{K}}^2_{i,s,\uparrow,\star}}{\hat{\tilde{\sigma}}^2\hat{\tilde{K}}^{\star\star}_{i,s}+ps\hat{\tilde{K}}_{i,s\uparrow}\hat{\tilde{K}}^{\star\star}_{i,s}}\right)$$

$$=\hat{\tilde{K}}^{\star\star}_{i,s}\left(\frac{\hat{\tilde{\sigma}}^2\hat{\tilde{K}}^{\star\star}_{i,s}+ps\hat{\tilde{K}}_{i,s\uparrow}\hat{\tilde{K}}^{\star\star}_{i,s}-ps\hat{\tilde{K}}^2_{i,s,\uparrow,\star}}{\hat{\tilde{\sigma}}^2\hat{\tilde{K}}^{\star\star}_{i,s}+ps\hat{\tilde{K}}_{i,s\uparrow}\hat{\tilde{K}}^{\star\star}_{i,s}}\right)$$

$$=\hat{\tilde{K}}^{\star\star}_{i,s}\left(\frac{\hat{\tilde{\sigma}}^2\hat{\tilde{K}}^{\star\star}_{i,s}+ps\hat{\tilde{K}}_{i,s\uparrow}\hat{\tilde{K}}^{\star\star}_{i,s}-ps\hat{\tilde{K}}^{\star\star2}_{i,s}+2ps\hat{\tilde{K}}^{\star\star}_{i,s}D_{i,s,\downarrow}-psD^2_{i,s,\downarrow}}{\hat{\tilde{\sigma}}^2\hat{\tilde{K}}^{\star\star}_{i,s}+ps\hat{\tilde{K}}_{i,s\uparrow}\hat{\tilde{K}}^{\star\star}_{i,s}}\right)$$

$$\geq\hat{\tilde{K}}^{\star\star}_{i,s}\left(\frac{\hat{\tilde{\sigma}}^2+2psD_{i,s,\downarrow}-ps\frac{1}{\hat{\tilde{K}}^{\star\star}_{i,s}}D^2_{i,s,\downarrow}}{\hat{\tilde{\sigma}}^2+ps\hat{\tilde{K}}_{i,s\uparrow}}\right)$$

and likewise for the AI posterior variances. For the upper bound, we may pessimise the bound by first using the maximum eigenvalue:

$$\hat{\bar{\sigma}}_{i,s}^2\left(\mathbf{x}^\star\right) \leq \hat{\bar{K}}_{i,s}^{\star\star} - \frac{\sum_{t\in\mathbb{T}_{\leq s}}\hat{\bar{K}}_{i,s,t,\star}^2}{\lambda_{\max}\left(\hat{\bar{\mathbf{K}}}_{i,s}\right)+\hat{\bar{\sigma}}^2}$$

and by Gershgorin's circle theorem $\lambda_{\max}(\hat{\bar{\mathbf{K}}}_{i,s}) \leq \sum_{t\in\mathbb{T}_{\leq s}}\hat{\bar{K}}_{i,s,t_s,t_s}$, so:

$$\begin{aligned}
\hat{\bar{\sigma}}_{i,s}^2\left(\mathbf{x}^\star\right) &\leq \hat{\bar{K}}_{i,s}^{\star\star} - \frac{\sum_{t\in\mathbb{T}_{\leq s}}\hat{\bar{K}}_{i,s,t,\star}^2}{\hat{\bar{\sigma}}^2+\sum_{t\in\mathbb{T}_{\leq s}}\hat{\bar{K}}_{i,s,t_s,t_s}} \\
&= \frac{\hat{\bar{\sigma}}^2\hat{\bar{K}}_{i,s}^{\star\star}+\sum_{t\in\mathbb{T}_{\leq s}}\hat{\bar{K}}_{i,s,t_s,t_s}\hat{\bar{K}}_{i,s}^{\star\star}-\sum_{t\in\mathbb{T}_{\leq s}}\hat{\bar{K}}_{i,s,t,\star}^2}{\hat{\bar{\sigma}}^2\hat{\bar{K}}_{i,s}^{\star\star}+\sum_{t\in\mathbb{T}_{\leq s}}\hat{\bar{K}}_{i,s,t_s,t_s}\hat{\bar{K}}_{i,s}^{\star\star}}\hat{\bar{K}}_{i,s}^{\star\star}
\end{aligned}$$

Now, we have that $\hat{\bar{K}}_{i,s,t,\star} = \hat{\bar{K}}_{i,s}^{\star\star} - \hat{\bar{D}}_{i,s,t}$, where $\hat{\bar{D}}_{i,s,t} \in [0, \hat{\bar{K}}_{i,s}^{\star\star}]$, so:

$$\begin{aligned}
\hat{\bar{\sigma}}_{i,s}^2\left(\mathbf{x}^\star\right) &\leq \frac{\hat{\bar{\sigma}}^2\hat{\bar{K}}_{i,s}^{\star\star}+\sum_{t\in\mathbb{T}_{\leq s}}\hat{\bar{K}}_{i,s,t_s,t_s}\hat{\bar{K}}_{i,s}^{\star\star}-ps\hat{\bar{K}}_{i,s}^{\star\star2}+2\hat{\bar{K}}_{i,s}^{\star\star}\sum_{t\in\mathbb{T}_{\leq s}}\hat{\bar{D}}_{i,s,t}-\sum_{t\in\mathbb{T}_{\leq s}}\hat{\bar{D}}_{i,s,t}^2}{\hat{\bar{\sigma}}^2\hat{\bar{K}}_{i,s}^{\star\star}+\sum_{t\in\mathbb{T}_{\leq s}}\hat{\bar{K}}_{i,s,t_s,t_s}\hat{\bar{K}}_{i,s}^{\star\star}}\hat{\bar{K}}_{i,s}^{\star\star} \\
&\leq \frac{\hat{\bar{\sigma}}^2\hat{\bar{K}}_{i,s}^{\star\star}+\sum_{t\in\mathbb{T}_{\leq s}}\hat{\bar{K}}_{i,s,t_s,t_s}\hat{\bar{K}}_{i,s}^{\star\star}-ps\hat{\bar{K}}_{i,s}^{\star\star2}+2\hat{\bar{K}}_{i,s}^{\star\star}\sum_{t\in\mathbb{T}_{\leq s}}\hat{\bar{D}}_{i,s,t}-\sum_{t\in\mathbb{T}_{\leq s}}\hat{\bar{D}}_{i,s,t}^2}{\hat{\bar{\sigma}}^2\hat{\bar{K}}_{i,s}^{\star\star}+\sum_{t\in\mathbb{T}_{\leq s}}\hat{\bar{K}}_{i,s,t_s,t_s}\hat{\bar{K}}_{i,s}^{\star\star}}\hat{\bar{K}}_{i,s}^{\star\star} \\
&\leq \frac{\hat{\bar{\sigma}}^2\hat{\bar{K}}_{i,s}^{\star\star}+\sum_{t\in\mathbb{T}_{\leq s}}\hat{\bar{K}}_{i,s}^{\star\star2}-ps\hat{\bar{K}}_{i,s}^{\star\star2}+2\hat{\bar{K}}_{i,s}^{\star\star}\sum_{t\in\mathbb{T}_{\leq s}}\hat{\bar{D}}_{i,s,t}-\sum_{t\in\mathbb{T}_{\leq s}}\hat{\bar{D}}_{i,s,t}^2}{\hat{\bar{\sigma}}^2\hat{\bar{K}}_{i,s}^{\star\star}+\sum_{t\in\mathbb{T}_{\leq s}}\hat{\bar{K}}_{i,s}^{\star\star2}}\hat{\bar{K}}_{i,s}^{\star\star} \\
&\leq \frac{\hat{\bar{\sigma}}^2\hat{\bar{K}}_{i,s}^{\star\star}+2\hat{\bar{K}}_{i,s}^{\star\star}\sum_{t\in\mathbb{T}_{\leq s}}\hat{\bar{D}}_{i,s,t}-\sum_{t\in\mathbb{T}_{\leq s}}\hat{\bar{D}}_{i,s,t}^2}{\hat{\bar{\sigma}}^2\hat{\bar{K}}_{i,s}^{\star\star}+ps\hat{\bar{K}}_{i,s}^{\star\star2}}\hat{\bar{K}}_{i,s}^{\star\star} \\
&= \frac{\hat{\bar{\sigma}}^2+2\sum_{t\in\mathbb{T}_{\leq s}}\hat{\bar{D}}_{i,s,t}-\frac{1}{\hat{\bar{K}}_{i,s}^{\star\star}}\sum_{t\in\mathbb{T}_{\leq s}}\hat{\bar{D}}_{i,s,t}^2}{\hat{\bar{\sigma}}^2+ps\hat{\bar{K}}_{i,s}^{\star\star}}\hat{\bar{K}}_{i,s}^{\star\star}
\end{aligned}$$

and likewise for AI variances. $\qquad\square$

In this theorem the sequence $\hat{\bar{D}}_{i,s,t} = \hat{\bar{K}}_{i,s}(\mathbf{x}^\star,\mathbf{x}^\star)-|\hat{\bar{K}}_{i,s}(\mathbf{x}^\star,\mathbf{x}_t)| \geq 0$ is a proxy for convergence, being minimised for $\mathbf{x}_t = \mathbf{x}^\star$ and increasing as $\mathbf{x}_t$ becomes (a-posterior) less correlated with $\mathbf{x}^\star$. If we consider the case considered in the paper - namely 1 human and 1 AI with zero gap and a trade-off sequence meeting the conditions of GP-UCB - then we know that the AI, operating alone, suffices to ensure convergence; and moreover adding additional observations (the human recommendations) will not prevent this. From here, it is not difficult to see that the upper and lower bounds in the above theorem converge not just to 0 but to one another, and if we further assume that $\hat{\bar{K}}_{i,s}^{\star\star} = \check{\bar{K}}_{j,s}^{\star\star}$ then the ratio of any upper bound on the posterior variance to the lower bound on any posterior variance will convergen to 1.

### A.7 ADDITIONAL SYNTHETIC EXPERIMENTS

In addition to the experimental results mentioned in the main paper, we provide the optimisation performance of our proposed BO-Muse framework on the following synthetic benchmark functions: (a) Matyas-2D, and (b) Rastrigin-5D. The details of the additional synthetic optimisation benchmark functions are mentioned in Table 2.

We maintain the same experimental settings mentioned in the main paper for the additional experiments conducted. For a given $d$ dimensional problem, we use $d+1$ initial observations and optimise for $10 \times d$ iterations i.e., the budget allocated for our synthetic experiments is set to $10 \times d$ function evaluations. The simple regret plots obtained for Matyas-2D function and Rastrigin-5D function after 10 runs with random initialisations are shown in Figure 4a and Figure 4b, respectively.

### A.8 ABLATION STUDIES

#### A.8.1 VARYING DEGREES OF EXPLOITATION-EXPLORATION

We study the sensitivity of the exploitation-exploration parameter $(\hat{\beta}_s)$ in our proposed BO-Muse framework and compare the optimisation performance. We vary the exploitation-exploration param-

| Functions | $f^\star(\mathbf{x})$ | High Level Features |
|---|---|---|
| Matyas-2D | $0.26(x_1^2 + x_2^2) - 0.48(x_1 x_2)$ | $x_1' = x_1^2,\ x_2' = x_2^2$ 
 $x_3' = x_1 x_2$ |
| Rastrigin-5D | $10d + \sum_i^d [x_i^2 - 10\cos(2\pi x_i)]\ \forall i \in \mathbb{N}_5$ | $x_i' = x_i^2\ \forall i \in \mathbb{N}_5$ 
 $x_{j+5}' = \cos x_j\ \forall j \in \mathbb{N}_5$ |

Table 2: Synthetic optimisation benchmark functions. Analytical forms are provided in the second column and the last column depicts the features used by a simulated human expert.

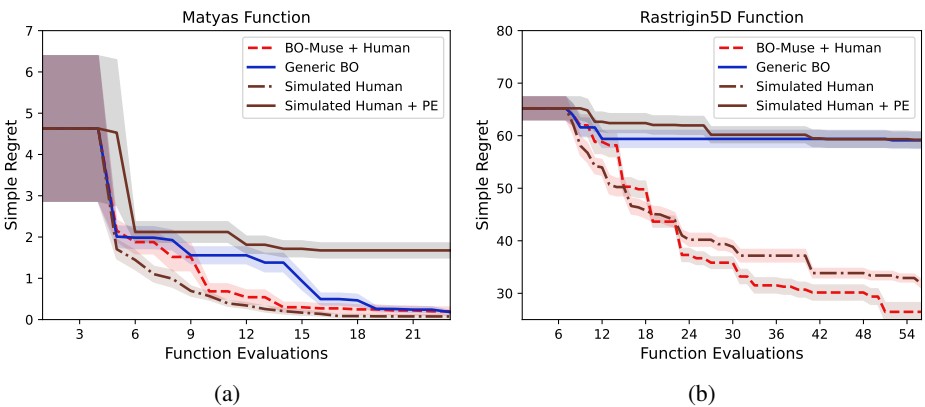

Figure 4: Simple regret vs iterations for synthetic Functions:(a) Matyas-2D (b) Rastrigin-5D

eter ($\hat{\beta}_s$) in the exponent range of $[0,3]$ i.e., $\hat{\beta}_s \in 10^{[0,3]}$ to cover the whole spectrum from over-exploitative experts ($\hat{\beta}_s = 1$) to over-explorative experts ($\hat{\beta}_s = 1000$). We have tuned the Squared Exponential (SE) kernel hyper-parameters of the inherent GP surrogate models using maximum-likelihood estimation. The empirical results obtained for various synthetic functions are depicted in Figure 5. It is evident from the empirical results that BO-Muse teamed up with an expert following more of an exploitation strategy ($\hat{\beta}_s = 1$) has better convergence when compared to its counterpart teamed with pure explorative expert ($\hat{\beta}_s = 1000$).

### A.8.2 BO-MUSE WITH EXPECTED IMPROVEMENT ACQUISITION FUNCTION

We have conducted an additional experiment to study the behaviour of our BO-Muse framework with different acquisition function strategies for the human experts. Expected Improvement (EI)

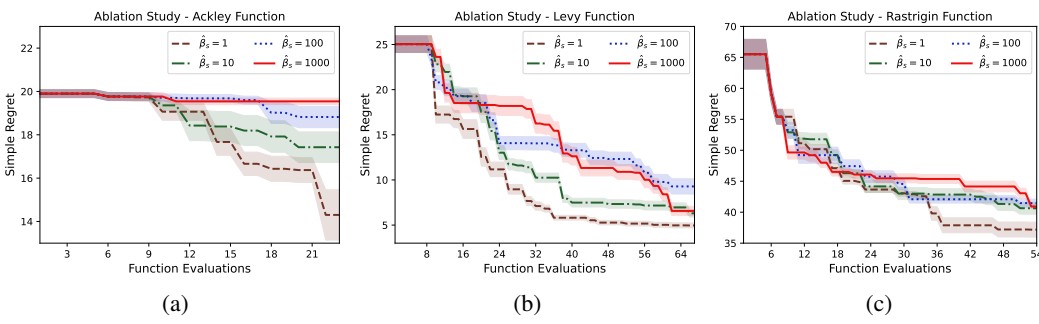

Figure 5: Ablation study with varying degrees of exploration-exploitation parameter ($\hat{\beta}_s$) obtained for (a) Ackley-4D (b) Levy-6D functions (c) Rastrigin-5D functions.

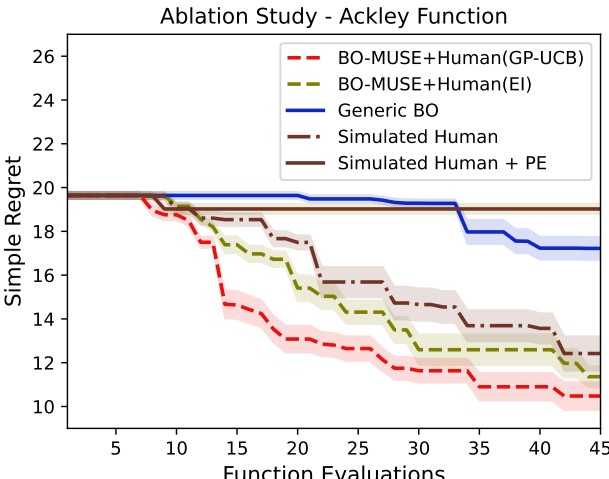

Figure 6: Ablation study with EI acquisition function obtained for Ackley function.

acquisition function (Wilson et al., 2018) guides the search for optima by taking into account the expected improvement over the current best solution. If $f^\star(\mathbf{x}^+)$ is the best value observed, then the next best query point is obtained by maximising the EI acquisition function $\hat{a}_s^{\text{EI}}(\mathbf{x})$, given by:

$$\hat{a}_s^{\text{EI}}(\mathbf{x}) = \begin{cases} (\hat{\mu}_s(\mathbf{x}) - f^\star(\mathbf{x}^+))\ \Phi(\mathcal{Z}) + \hat{\sigma}_s(\mathbf{x})\ \phi(\mathcal{Z}) & \text{if } \hat{\sigma}_s(\mathbf{x}) > 0 \\ 0 & \text{if } \hat{\sigma}_s(\mathbf{x}) = 0 \end{cases}$$

$$\mathcal{Z} = \frac{\hat{\mu}_s(\mathbf{x}) - f^\star(\mathbf{x}^+)}{\hat{\sigma}_s(\mathbf{x})}$$

where $\Phi(\mathcal{Z})$ and $\phi(\mathcal{Z})$ represents the Cumulative Distribution Function (CDF) and the Probability Density Function (PDF) of the standard normal distribution, respectively.

In this experiment, the GP-UCB acquisition function used by the human expert in BO-Muse framework is now replaced with the Expected Improvement acquisition function. We compare this new baseline (BO-Muse + Human (EI)) with BO-Muse + Human (GP-UCB) and all the other competing baselines. The empirical results obtained for the experiment with EI acquisition function is depicted in Figure 6. As expected BO-Muse with the EI acquisition function still outperforms the standard baselines considered. However, BO-Muse with GP-UCB acquisition function has superior performance when compared to its counterpart with the EI acquisition function.

### A.9 Additional Details of Classification Experiments

We have considered two real-world classification tasks using Support Vector Machines (SVMs) and Random Forests (RFs). This experiment involves hyperparameter tuning of SVMs and RFs operating on the *Biodeg* dataset to classify biodegradable and non-biodegradable materials. We used publicly available Biodeg dataset from the UCI data repository (Dua & Graff, 2017). Biodeg dataset consists of 1056 instances with 41 features. We randomly split the dataset into 80/20 train/test splits. Each time a hyperparameter set (design) is chosen, the model needs to retrained and evaluated on a held out set. The goal is to reach to the hyperparameter set that leads to a classification model with the minimum test classification error.

We have created two groups (arms) with 4 members (2 students and 2 postdoctoral researchers) randomly allocated in each group. Each expert in the first group teams up with AI as per BO-Muse, while each expert in the second group (baseline) tunes the classifier completely on their own. For each of the classification tasks, the two groups use the same tuning budget (3 random initial designs + 30 further iterations. The aforementioned real-world task is suitable for our case: (1) It is easier to find multiple human experts for this task as AI graduate students and post-doctoral

researchers have a good understanding of classification (SVM and RF) models and understand how its hyperparameters generally influence the model fitting, (2) This task is familiar to the ICLR and machine learning community.

The simple graphical interface used by each participant is shown in Figure 7a and 7b. The interface shows accumulated observations of classifier performance as a function of hyper-parameters, with the best results shown in darker shades of blue. Interfaces for both teams are the same, excepting that participants working with BO-Muse see the AI-generated observation in the previous (#) iteration (indicated by ▼) in addition to the existing set of observed samples. Experts click on the plot to provide their suggestion on the next hyper-parameter set to evaluate. The co-ordinates of the clicked point is used as the expert suggestion.

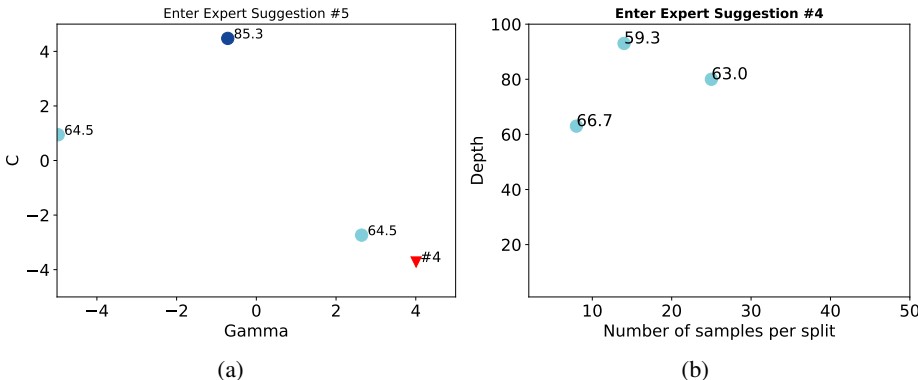

Figure 7: Simple graphical interface used by (a) Team 1 and (b) Team 2 for both the classification experiments using SVM and RF classifiers.

## A.10 SPACECRAFT SHIELDING DESIGN EXPERIMENT

Our third experiment is to team with an expert to design spacecraft shields to protect from impact by orbital debris particles.

### A.10.1 EXPERIMENTAL PROBLEM

Here we consider the design of a two- or three-wall shield for protection against a cubic steel projectile impacting face on, normal to the surface of the target plates, at an impact velocity of 7.0 km/s. There exists no state-of-the-art solution for such an impact threat, however for protecting against a spherical aluminium projectiles in this velocity domain the state-of-the-art solution would be a "stuffed Whipple shield" after Christiansen et al. (1995), consisting of an outer aluminium plate, inner layers of aramid and ceramic fabrics, and a rear wall (pressure hull) of aluminium. US, Japanese, and European modules on the ISS all utilise stuffed Whipple shield designs (Christiansen et al., 2009). The design space is schematically shown in Figure 8. Design variables include: (1) plate material - AA6061-T651 ("AL"), 4340 steel ("ST"), Kevlar/epoxy ("KE"), and ultra-high molecular weight polyethylene ("PE"); (2) plate thickness - 0.1 cm to 1.0 cm in 0.1 cm increments; (3) plate spacing, S - 0.0 cm to 10 cm in 1.0 cm increments, an; (4) number of plates, 2 or 3 (*i.e.,* the 'outer bumper' plate may or may not be used). Only metal plates (i.e., "AL" or "ST") may be used for the 3rd plate. The full factorial design space includes 577,365 options.

### A.10.2 BACKGROUND

Spacecraft are subject to impact by natural micrometeoroid and man-made orbital debris particles, collectively referred to as space debris, during their orbital lifetime. The impact of such particles (typically at velocities above 10 km/s) is a significant risk to the safe operation of spacecraft and the fulfillment of mission objectives. Indeed, for manned spacecraft such as the International Space

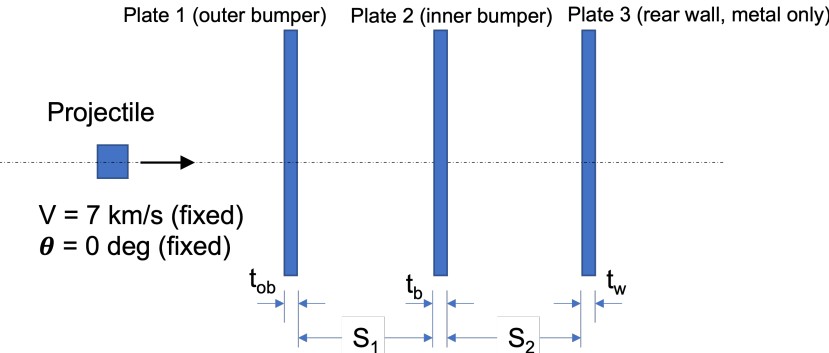

Figure 8: Schematic of the spec debris shield design problem. Our objective is a design solution that will prevent perforation of the spacecraft hull (Plate 3, rear wall) by modifying the plate materials, plate thicknesses, and plate spacing (S).

Station (ISS) and Space Shuttle Orbiter, space debris impact is the top mission risk (see e.g., Hamlin et al. (2013)). As such, all manned spacecraft and some robotic spacecraft carry dedicated protective shields. The most common shield configuration is a simple design known as a Whipple shield with two thin plates separated by a gap. Meteoroid and debris particles, upon impact with the outer plate, fragment into a cloud of solid, molten, and vaporised particles. This debris cloud expands as it propagates through the gap, resulting in a dispersed and substantially less lethal load upon the spacecraft hull. Multiple variants of the Whipple shield exist, including stuffed Whipple shields which utilise intermediate layers of high-strength and high-impedence fabric (see e.g., Christiansen et al. (1995)) and multi-shock shields which induce multiple impact shocks into the projectile to promote maximum melting and vaporisation (Cour-Palais & Crews, 1990).

Spacecraft debris shields are typically designed using a combination of semi-analytical equations, numerical simulations, and experimental testing. Simulations are performed in either explicit finite element solvers, e.g., ANSYS LS-DYNA, or shock physics solvers, e.g., CTH from Sandia National Laboratory. Modelling hypervelocity impact in those simulation codes requires substantial expertise to accurately projectile and target kinetmatics together with material response. Furthermore, such simulations can be computationally expensive, requiring hundreds of CPU hours depending on the geometric discretisation of the model. Experimentation is typically performed on laboratory accelerators known as two-stage light gas guns. The number of such facilities that can perform experiments with millimetre and centimetre sized proejctiles up to impact velocities of 7+ km/s is very limited (estimated to be < 20 globally). Such experiments are also expensive, costing on the order of thousands of dollars, with a low through-put of approximately 1 experiment per day. In the design of space debris shielding, in order to minimise the number of experiments and simulations required, space debris is typically simplified to spherical aluminium particles. In reality, of course, the debris environment consists of a range of materials, both metallic and non-metallic, the properties of which influence their impact lethality. Similarly, for robotic and manned spacecraft the majority of impact risk is represented by millimeter-sized objects, the majority of which are fragmentation debris that have been generated by catestrophic breakup of a satellite or rocket body and are thus highly irregular in shape, see Rivero et al. (2016). Until recently the engineering environment models used to predict mission risk to space debris impact have also simplified the debris population as spherical aluminium objects, thus there was little incentive to introduce the added complexity of projectile shape and material effects in shielding design or characterisation studies. However, recent improvements in orbital debris environment engineering models, e.g., ORDEM 3.0 (Krisko, 2014), and planned improvements to debris population source models, e.g., via DebriSat (Rivero et al., 2016), aim to address some of these deficiencies. Shield design and characterisation, therefore, must also begin to account for projectile shape and material effects.

### A.10.3  DETAILS OF EXPERIMENT

The spacecraft shielding design experiment utilises synthetic data generated via numerical simulation. This section provides additional information on the simulation setup and evaluation. Simula-

| Parameter | Value | Units | Source |
|---|---|---|---|
| *EoS: Gruneisen* | | | |
| Density, $\rho$ | 2700 | $kg/m^3$ | (Anon., 2003) |
| Shear modulus, $G$ | 26.2 | $GPa$ | (Anon., 2003) |
| Elastic modulus, $E$ | 68.3 | $GPa$ | (Anon., 2003) |
| Poisson's ratio, $\nu$ | 0.33 | - | (Anon., 2003) |
| Melting temperature, $T_m$ | 930 | K | (Corbett, 2006) |
| Bulk soundspeed, $c_0$ | 5240 | $m/s$ | (Corbett, 2006) |
| Parameter $S_1$ | 1.4 | - | (Corbett, 2006) |
| Parameter $S_2$ | 0.0 | - | (Corbett, 2006) |
| Parameter $S_3$ | 0.0 | - | (Corbett, 2006) |
| Gruneisen gamma, $\Gamma$ | 1.97 | - | (Corbett, 2006) |
| Specific heat, $c_p$ | 885 | $J/kgK$ | (Corbett, 2006) |
| | | | |
| *Strength: Johnson-Cook* | | | |
| Yield stress, $A$ | 324 | $MPa$ | (Anderson Jr et al., 2006) |
| Hardening constant, $B$ | 114 | $MPa$ | (Anderson Jr et al., 2006) |
| Hardening exponent, $n$ | 0.42 | - | (Anderson Jr et al., 2006) |
| Strain rate constant, $C$ | 0.002 | - | (Anderson Jr et al., 2006) |
| Thermal softening exponent, $m$ | 1.34 | - | (Anderson Jr et al., 2006) |
| Reference strain rate, $\dot{\varepsilon}_0$ | 1.0 | $/s$ | (Anderson Jr et al., 2006) |
| | | | |
| *Failure: Johnson-Cook* | | | |
| Failure constant, $d_1$ | 0.0 | - | (Anderson Jr et al., 2006) |
| Triaxiality constant, $d_2$ | 1.11 | - | (Anderson Jr et al., 2006) |
| Triaxiality exponent, $d_3$ | -1.5 | - | (Anderson Jr et al., 2006) |
| Strain rate constant, $d_4$ | 0.0 | - | (Anderson Jr et al., 2006) |
| Thermal softening constant, $d_5$ | 0.0 | - | (Anderson Jr et al., 2006) |

Table 3: Constitutive model parameters for AA6061-T651 used in the numerical simulations.

tions are performed in the explicit structural mechanics solver LS-DYNA from ANSYS (Hallquist, 2006). Simulations are performed in 3D using a smooth particle hydrodynamics (SPH) discretisation scheme, which enables projectile fragmentation to be modelled without arbitrary numerical erosion that would otherwise be required for a mesh-based Lagrangian scheme. SPH elements of 0.05 mm diameter are used to disretise all simulated parts. The metallic materials, AA6061-T651 ("AL") and 4340 steel ("ST"), utilise a Gruneisen equation of state (EoS) (Gruneisen, 1959), a Johnson-Cook viscoplasticity model (Johnson & Cook, 1983) and a Johnson-Cook fracture model (Johnson & Cook, 1985), the constants for which are given in Tables 3 and 4. The aramid composite ("KE") is modelled as a continuum using the elastic-plastic orthotropic strength with failure model from LS-DYNA (MAT_059) and a linear EoS, the constants for which are given in Table 5. The ultra-high molecular weight polyethylene ("PE"), specifically Dyneema HB26, is modelled using the orthotropic non-linear model and material constants from (Nguyen et al., 2016).

In Figure 9 a series of frames from a representative LS-DYNA simulation are provided, depicting the impact of the cubic steel projectile against a three-wall shield design.

Simulations are performed on AMD EPYC servers with 64 CPU cores (2.25 GHz) and 1 TB of RAM. All simulations are performed in parallel on 4 CPU cores and require 20-40 CPU hours to run, depending on the complexity (in this case thickness) of the target plates. The simulation models included a 3.0 cm thick aluminium alloy witness plate located 10.0 cm from the rear surface of the target rear wall (Plate 3 in Figure 8) to measure residual penetration in the event that the shield was perforated. Results were recorded as a binary pass/fail related to non-perforation or perforation of the target rear wall, respectively, together with a continuous Depth of Penetration (DoP) measurement into the witness plate. Our objective is to design a protective shield that can defeat the projectile threat for minimal weight.

| Parameter | Value | Units | Source |
|---|---|---|---|
| *EoS: Gruneisen* | | | |
| Density, $\rho$ | 7850 | $kg/m^3$ | (Anon., 2022) |
| Shear modulus, $G$ | 80.0 | $GPa$ | (Anon., 2022) |
| Elastic modulus, $E$ | 205.0 | $GPa$ | (Anon., 2022) |
| Poisson's ratio, $\nu$ | 0.29 | - | (Anon., 2022) |
| Melting temperature, $T_m$ | 1700 | K | (Anon., 2022) |
| Bulk soundspeed, $c_0$ | 3935 | $m/s$ | (Banerjee, 2007) |
| Parameter $S_1$ | 1.578 | - | (Banerjee, 2007) |
| Parameter $S_2$ | 0.0 | - | (Banerjee, 2007) |
| Parameter $S_3$ | 0.0 | - | (Banerjee, 2007) |
| Gruneisen gamma, $\Gamma$ | 1.69 | - | (Banerjee, 2007) |
| Specific heat, $c_p$ | 475 | $J/kgK$ | (Anon., 2022) |
| | | | |
| *Strength: Johnson-Cook* | | | |
| Yield stress, $A$ | 910 | $MPa$ | (Holmquist et al., 2001) |
| Hardening constant, $B$ | 586 | $MPa$ | (Holmquist et al., 2001) |
| Hardening exponent, $n$ | 0.26 | - | (Holmquist et al., 2001) |
| Strain rate constant, $C$ | 0.014 | - | (Holmquist et al., 2001) |
| Thermal softening exponent, $m$ | 1.03 | - | (Holmquist et al., 2001) |
| Reference strain rate, $\dot{\varepsilon_0}$ | 1.0 | $/s$ | (Holmquist et al., 2001) |
| | | | |
| *Failure: Johnson-Cook* | | | |
| Failure constant, $d_1$ | -0.80 | - | (Holmquist et al., 2001) |
| Triaxiality constant, $d_2$ | 2.10 | - | (Holmquist et al., 2001) |
| Triaxiality exponent, $d_3$ | -0.50 | - | (Holmquist et al., 2001) |
| Strain rate constant, $d_4$ | 0.002 | - | (Holmquist et al., 2001) |
| Thermal softening constant, $d_5$ | 0.61 | - | (Holmquist et al., 2001) |

Table 4: Constitutive model parameters for 4340 steel used in the numerical simulations.

| Parameter | Value | Units | Source |
|---|---|---|---|
| *EoS: Linear* | | | |
| Density, $\rho$ | 1230 | $kg/m^3$ | (van Hoof et al., 1999) |
| Shear modulus, $G_{11}$ | 0.77 | $GPa$ | (van Hoof et al., 1999) |
| Shear modulus, $G_{22}$ | 1.36 | $GPa$ | (van Hoof et al., 1999) |
| Shear modulus, $G_{33}$ | 1.36 | $GPa$ | (van Hoof et al., 1999) |
| Elastic modulus, $E_{11}$ | 18.5 | $GPa$ | (van Hoof et al., 1999) |
| Elastic modulus, $E_{22}$ | 18.5 | $GPa$ | (van Hoof et al., 1999) |
| Elastic modulus, $E_{33}$ | 6.0 | $GPa$ | (van Hoof et al., 1999) |
| Poisson's ratio, $\nu_{21}$ | 0.25 | - | (van Hoof et al., 1999) |
| Poisson's ratio, $\nu_{31}$ | 0.33 | - | (van Hoof et al., 1999) |
| Poisson's ratio, $\nu_{32}$ | 0.33 | - | (van Hoof et al., 1999) |
| | | | |
| *Strength: Composite Failure* | | | |
| In-plane shear strength, $S_{21}$ | 9.0 | $MPa$ | (van Hoof et al., 1999) |
| Transverse shear strength, $S_{31}$ | 271.5 | $MPa$ | (van Hoof et al., 1999) |
| Transverse shear strength, $S_{32}$ | 271.5 | $MPa$ | (van Hoof et al., 1999) |
| Longitudinal compressive strength, $C_{11}$ | 221.0 | $MPa$ | (Silcock et al., 2006) |
| Transverse compressive strength, $C_{22}$ | 221.0 | $MPa$ | (Silcock et al., 2006) |
| Normal compressive strength, $C_{33}$ | 1200.0 | $MPa$ | (van Hoof et al., 1999) |
| Longitudinal tensile strength, $T_{11}$ | 740.0 | $MPa$ | (van Hoof et al., 1999) |
| Transverse tensile strength, $T_{22}$ | 740.0 | $MPa$ | (van Hoof et al., 1999) |
| Normal tensile strength, $T_{33}$ | 34.5 | $MPa$ | (van Hoof et al., 1999) |

Table 5: Constitutive model parameters for woven aramid composite used in the numerical simulations.

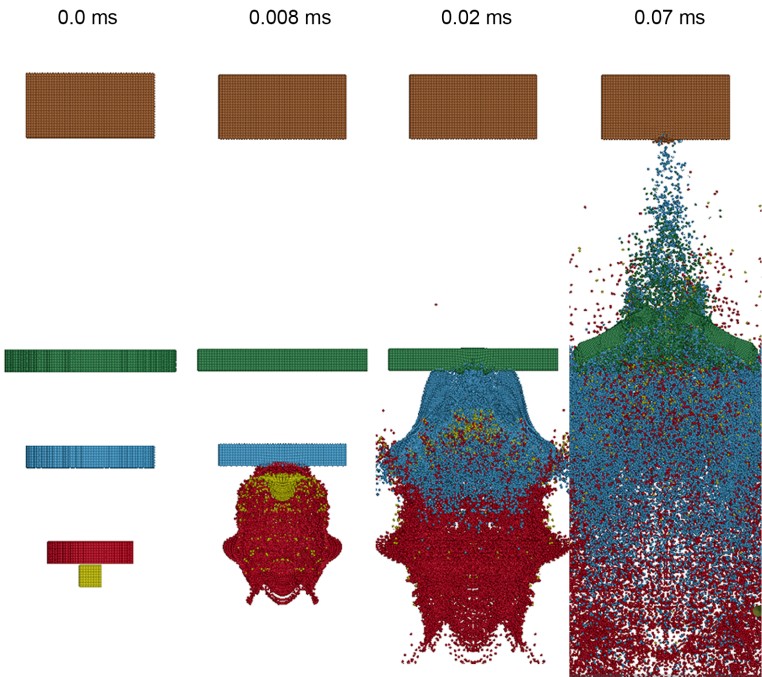

0.0 ms      0.008 ms      0.02 ms      0.07 ms

Figure 9: Series of frames from an LS-DYNA simulation showing the impact of a steel cubic particle (yellow) at 7 km/s against a shield consisting of three 1.0 cm thick plates of AA6061-T651 (red - outer bumper, blue - inner bumper, green - rear wall) with a 3.0 mm thick AA6061-T651 witness plate (brown) positioned 10 cm behind the rearmost surface of the shield. The projectile is shown to fragment into a dispersed cloud of projectile and shield fragments which radially disperse before impacting upon the rear wall. The rear wall is perforated and fragments are observed to crater the witness plate, providing a non-zero depth of penetration measurement.

Three initial designs are evaluated by the human expert, the details of which are provided in Table 6 together with the simulation results. Based on these results we assume that target designs with areal weights significantly less than 5 $kg/m^2$ are likely infeasible, while designs significantly heavier ($>$ $15kg/m^2$) are not of interest. Within this weight range the design space includes 577,365 potential options. We perform the optimisation in iterative batches of size 2, with one suggestion from the BO and one from the human expert.

### A.10.4   DIFFERENCE TO A CLASSICAL CS SETTING

We work with one expert and this differs from usual CS settings where multiple experts perform the same task because:

1. *Level of expertise required is high, and access is difficult:* The design of shields for protecting against space debris impact at hypervelocity is a highly specialised discipline typically limited to national space agencies or their primary contractors. For instance, shielding onboard the ISS was predominantly developed by NASA and Boeing for US modules, ROSCOSMOS and RKK Energia for Russian modules, and JAXA for the Japanese module. Shielding on the European Coloumbus module was designed primarily by Alenia Aerospazio under contract to the European Space Agency, but borrowed heavily from the NASA designs (see e.g.,Destefanis et al. (1999)). Therefore, recruiting multiple experts is a formidable task.

2. *The cost of experiments is high:* Due to the cost and access limitations on experimental facilities, we utilise numerical simulations for this design study. Such simulations are difficult to design and validate, our expert (with 20 years experience on such codes) required about 120 hours to build the simulation models. In addition, the simulations can be com-

putationally expensive, requiring on the order of 100-200 CPU hours per simulation for a moderate CPU.

3. *The design problem is hard:* There exists no state-of-the-art solution for the defined problem. The nearest analogue is a shield designed for a spherical aluminium projectile, for which the state-of-the-art is a stuffed Whipple shield. This shielding configuration has been used to define our optimisation variables. Existing semi-analytical penetration laws, such as those in Christiansen et al. (1995), are not valid for application with non-spherical or non-aluminium projectiles.

4. *The expert cannot repeat the same design task* as they learn during the first experiment.

All these factors mean that for a real experiment we can only show how BO-Muse helps a single expert for a new problem for which there is no state-of-art solution.

### A.10.5 RESULTS

Results of the experiment are provided in Table 6. We can observe that the human expert initially explored designs similar to the stuffed Whipple concept (i.e., metallic outer bumper, KE or PE inner bumper, and metallic rear wall, with the inner bumper being roughly located at the mid-point between the two metallic plates). By design ID 9 the expert had identified a stuffed Whipple design that was successful at defeating the projectile, with an areal weight of 14.3 $kg/m^2$. Between design IDs 9 and 19 we can observe the expert exploiting this successful design to identify a lower weight solution, without success. Up to this point the expert does not seem to have been influenced by the BO suggestions. By design ID 21, however, we can observe the expert beginning to exploit BO suggestions, with design ID 21 a modification of ID 5 and 14, design ID 23 a modification of 18, and so on. Design ID 27, an expert suggestion that is an exploitation of the BO suggested ID 22, was successful at defeating the projectile, albeit at a higher weight than the previously identified solutions at ID 9 and 11 (16.9 $kg/m^2$). The final expert design, ID 29, is a further exploitations of ID 27 intended to reduce weight and is found to be the best solution identified during the experiment, with an areal weight of 13.8 $kg/m^2$.

The BO-Muse design ID 22 and subsequent human exploitations (design IDs 27 and 29) are, according to our human expert, highly unusual configurations for spacecraft debris shielding that they would not have otherwise considered if not for the BO suggestion. The conventional design methodology based on typical shields used in flight harware suggest that the outer bumper have a density and a shock impedance comparable to that of the projectile and be sized such that the shock rarefaction and tensile release wave superimpose towards the back of the projectile, maximising fragmentation and radial dispersion. An internal fabric layer, such as that used in stuffed Whipple shields (see e.g., Christiansen et al. (1995)) is then intended to catch and decelerate projectile fragments prior to impact upon the shield rear wall. This general design principle has been established through ongoing investigation since the Apollo program and matured for the International Space Station with significant, proven success. Design ID 22 and the subsequent designs (IDs 27 and 29) are a substantial deviation from these established principles and at current it is unclear why they have been successful - further investigation is needed.

Evaluating this experiment - the role of BO-Muse was to inject novelty in the design process, from which the human expert could take inspiration and perform exploitation. We consider this to have been successfully demonstrated in a real applied engineering design experiment

### A.11 DISCUSSION OF LIMITATIONS

In this section we present a brief discussion of the limitations of our work. With regard to the human expert, we have assumed that "Cognitive entrenchment" behaviour occurs, which is backed by recent studies (Dane, 2010; Daw et al., 2006). This may not hold strictly in all cases, which may cause the algorithm's sample efficiency to be lower than expected as BO may over-compensate for expected expert over-exploitation that does not eventuate. Similarly, we assume that the expert is able to improve their model as the number of observations available to them increases. However a less skilled expert may fail to do this, and subsequently the algorithm's sample efficiency may suffer as, after a point, expert suggestions may cease to be useful. Finally, the human expert may

| ID | Source | Plate 1 | | Gap1 | Plate 2 | | Gap2 | Plate 3 | | SC | Weight | **Result** | DoP |
|---|---|---|---|---|---|---|---|---|---|---|---|---|---|
| | | Mat | $t$(cm) | (cm) | Mat | $t$(cm) | (cm) | Mat | t(cm) | (cm) | $(kg/m^2)$ | [1/0] | (cm) |
| 1 | expert | ST | 0.2 | 2.0 | ST | 0.2 | 5.0 | ST | 0.5 | 7.9 | 7.1 | 1 | 2.89 |
| 2 | expert | AL | 1.0 | 3.5 | AL | 1.0 | 3.0 | AL | 1.0 | 9.5 | 8.1 | 1 | 0.31 |
| 3 | expert | ST | 1.0 | 6.0 | AL | 1.0 | 1.0 | AL | 1.0 | 10.0 | 13.3 | 1 | 0.01 |
| 4 | BO | ST | 0.9 | 0.0 | AL | 0.9 | 7.0 | AL | 0.6 | 9.4 | 11.1 | 1 | 0.43 |
| 5 | expert | AL | 0.5 | 5.0 | KE | 1.0 | 2.0 | AL | 1.0 | 9.5 | 5.3 | 1 | 0.22 |
| 6 | BO | ST | 0.3 | 8.0 | PE | 0.6 | 0.0 | AL | 0.9 | 9.8 | 5.4 | 1 | 0.31 |
| 7 | expert | ST | 0.3 | 5.0 | KE | 1.0 | 2.0 | ST | 1.0 | 9.3 | 11.4 | 1 | 0.07 |
| 8 | BO | ST | 0.5 | 7.0 | ST | 0.9 | 0.0 | AL | 0.9 | 9.3 | 13.4 | 1 | 0.68 |
| **9** | **expert** | **ST** | **0.7** | **4.0** | **PE** | **1.0** | **3.0** | **ST** | **1.0** | **9.7** | **14.3** | **0** | **0.0** |
| 10 | BO | AL | 0.9 | 0.0 | KE | 0.2 | 8.0 | ST | 0.5 | 9.6 | 6.6 | 1 | 1.90 |
| **11** | **expert** | **ST** | **0.7** | **4.0** | **KE** | **1.0** | **3.0** | **ST** | **1.0** | **9.7** | **14.6** | **0** | **0.0** |
| 12 | BO | PE | 0.2 | 8.0 | AL | 0.9 | 0.0 | ST | 0.9 | 10.0 | 9.7 | 1 | 2.84 |
| 13 | expert | ST | 0.6 | 4.0 | KE | 1.0 | 3.0 | ST | 0.9 | 9.5 | 13.0 | 1 | 0.01 |
| 14 | BO | - | - | 0.0 | KE | 0.9 | 0.0 | ST | 0.9 | 1.8 | 8.2 | 1 | 0.84 |
| 15 | expert | ST | 0.5 | 4.0 | KE | 1.0 | 3.0 | ST | 1.0 | 9.5 | 13.0 | 1 | 0.01 |
| 16 | BO | - | - | 0.0 | ST | 0.9 | 9.0 | AL | 0.1 | 10.0 | 7.3 | 1 | 0.32 |
| 17 | expert | ST | 0.7 | 4.0 | KE | 1.0 | 3.0 | ST | 0.8 | 9.5 | 13.0 | 1 | 0.01 |
| 18 | BO | AL | 0.9 | 8.0 | ST | 0.1 | 0.0 | AL | 0.9 | 9.9 | 5.6 | 1 | 0.39 |
| 19 | expert | ST | 0.5 | 6.0 | PE | 1.0 | 1.0 | ST | 1.0 | 9.5 | 12.8 | 1 | 0.06 |
| 20 | BO | PE | 0.0 | 0.0 | ST | 0.4 | 8.0 | AL | 0.9 | 9.3 | 5.6 | 1 | 0.31 |
| 21 | expert | ST | 0.7 | 0.0 | KE | 1.0 | 7.0 | ST | 1.0 | 9.7 | 14.6 | 1 | 0.33 |
| 22 | BO | KE | 0.9 | 0.0 | ST | 0.9 | 8.0 | ST | 0.1 | 9.9 | 9.0 | 1 | 0.26 |
| 23 | expert | ST | 1.0 | 5.0 | KE | 1.0 | 2.0 | AL | 1.0 | 10.0 | 11.8 | 1 | 0.11 |
| 24 | BO | - | - | 0.0 | ST | 0.9 | 0.0 | ST | 0.1 | 1.0 | 7.9 | 1 | 0.82 |
| 25 | expert | ST | 0.4 | 6.0 | AL | 1.0 | 1.0 | ST | 1.0 | 9.4 | 13.7 | 1 | 0.03 |
| 26 | BO | - | - | 0.0 | PE | 0.1 | 9.0 | ST | 0.9 | 10.0 | 7.2 | 1 | 1.53 |
| **27** | **expert** | **KE** | **1.0** | **0.0** | **ST** | **1.0** | **7.0** | **ST** | **1.0** | **10.0** | **16.9** | **0** | **0.0** |
| 28 | BO | AL | 0.9 | 0.0 | ST | 0.1 | 8.0 | AL | 0.9 | 9.9 | 5.6 | 1 | 1.75 |
| **29** | **expert** | **KE** | **1.0** | **0.0** | **ST** | **0.8** | **7.0** | **ST** | **0.8** | **9.6** | **13.8** | **0** | **0.0** |

Table 6: Results of the spacecraft protective design optimisation experiment. The best results are indicated in bold (SC - space claim).

not behave precisely like a GP-UCB model would suggest, again resulting in lower efficiency. We note, however, that even when the human is unable to perform as expected, BO-Muse will still have sublinear convergence. In such a worst-case scenario every second experiment will, in effect, be wasted. However the data from these "wasted" experiments will still provide additional observations of $f^\star$ for the machine's GP model, which can only improve the model's accuracy. The behaviour of the algorithm in this case can therefore be analysed in the "machine running GP-UCB plus improved prior due to additional data" regime - that is, a GP-UCB algorithm with AI-generated suggestions, an exploration parameter $\beta$ that is increased by a constant multiplicative factor, and a stream of additional (harmless or even potentially informative) human-guided experimental observations - which suffices to ensure sublinear convergence (Srinivas et al., 2012). We also note that the aforesaid worst-case scenario is highly unlikely on the assumption that the human expert is knowledgeable in the relevant field.

