# OpenReview forum: "BO-Muse: A Human expert and AI teaming framework for accelerated experimental design "
_ICLR.cc/2023/Conference — Submitted to ICLR 2023_

### Official Review · Reviewer_LH2r · 2022-10-25

**Confidence:** 4
**Correctness:** 3
**Technical Novelty And Significance:** 3
**Empirical Novelty And Significance:** 3
**Recommendation:** 6

**Clarity, Quality, Novelty And Reproducibility:**

I have a few comments/questions regarding clarity, quality and novelty:

- For the experiment in Figure 2, do the Generic-BO, and Simulated Human approaches use the same number of samples as Human+BOMuse? If not, wouldn't it be an unfair comparison? For instance, if we do 50 iterations of BO-Muse this leads to 100 function evaluations. Does the AI-alone baseline use 100 function evaluations too?
- In the experiments with SVMs and RFs, on average, how often was the best design obtained from AI suggestion?
- Does the paper implicitly assume that the cost of getting a design from humans is negligible compared to evaluating the objective function? It seems that this is not discussed in the paper.
- Unlike the synthetic experiments, the real-world ones do not report performance for the AI-alone model. Thus, it is hard to assess if humans are helpful.
- I understand that the paper focuses on AI-assisted experimental design (humans drive the process). However, it would be interesting to compare BO-Muse to $\pi$BO (https://arxiv.org/abs/2204.11051) which allows experts to specify informative priors over the design space and then run regular BO (AI model). This would allow us to verify if BO-Muse is the best way to leverage expert knowledge for these experimental design tasks.
- Some of the choices are justified by being adopted in previous works. For instance, "Motivated by Borji and Itti, we assume the human expert in effect generates their recommendations using EC-GP-UCB...". How would BO-Muse perform using another human acquisition function on the synthetic experiments?

Reproducibility: the authors have submitted their code for the experiments on synthetic datasets.


**Strength And Weaknesses:**

Strengths
- The paper proposes a principled way to incorporate expert knowledge while providing theoretical guarantees in the form of improved sublinear regret bounds wrt AI-alone BO.
- The paper reports experiments with humans on relevant experimental design tasks.
- I like the simplicity of the idea, which basically equips humans with AI (GP) that leverages their designs and helps them to explore outside their strong beliefs.

Weaknesses
- Some modeling choices seem arbitrary and mainly motivated by following existing works.
- It needs to be clarified if the experiments are fair, e.g., if the baselines use the same number of data samples.




**Summary Of The Paper:**

The paper proposes a framework that combines expert knowledge and AI for improved experimental design. The principle is to let humans drive the discovery/optimization process while the AI provides additional data to overcome the strong exploitation bias of humans. The AI model consists of a GP that is fit using data from human + AI-recommended designs and employs a GP-UCB acquisition function. For simulation and theoretical developments, the human model consists of a misspecified model and employs an acquisition (EC-GP-UCB) function with parameters chosen to simulate human biases. The paper reports results on both synthetic and real-world experiments.


**Summary Of The Review:**

Overall, the paper reads well and proposes a relatively novel framework. Although the technical novelty is limited, the paper analyzes the proposal, providing theoretical regret bounds. I have some issues, mainly regarding the fairness of experiments. Thus, I am inclined toward accepting the paper.

-------
Given the limited time, I acknowledge that I have not checked the proof in the Appendix.

---

> ### Author Response · Authors · 2022-11-17
> **Response to reviewer comments**
>
> We are grateful to the reviewer for providing useful feedback on our paper and have revised and updated the paper in accordance with their suggestions.  We will address specific concerns below:
>
> W1: Some modeling choices seem arbitrary ...motivated by ..existing works.
>
> A: Our key assumptions regarding human expert behaviour in an optimisation setting have been motivated by (Borji and Itti, 2013; Dane, 2010), as well as observations of how experts approach practical problems that have arisen in our previous work on BO.  We have added additional discussion regarding our assumptions here:
>
> - page 4, paragraph after equation (3) discussing the exploration/exploitation trade-off of the human expert.
> - section 3.1, after possible definitions of $\hat{K}_s$ discussing the human model, learning, and maximum information gain.
>
> We do not believe that these assumptions are arbitrary.  eg expecting the human expert to use some unknown trade-off between exploitation/exploration and designing the AI to compensate for the worst-case exploitation of the expert is not too restrictive.  Moreover assuming the human has some form of unknown feature/similarity-based model of the system from which a kernel could, in principle, be derived, models closely the behaviour we would expect and indeed have observed in human-experts.  Nevertheless, as noted in the extensively re-written section 3.3, while the convergence of BO-Muse will be accelerated if these assumptions are met, even in the fall-back case (arbitrary human) BO-Muse will still converge at the same big-O rate as GP-UCB.
>
> W2: It needs to be clarified if the experiment.. baselines use the same number of data samples.
> Q1: ...Figure 2, do the Generic-BO [etc] use the same number of samples as Human+BOMuse?...50 iterations of BO-Muse... leads to 100 evaluations. Does the AI-alone baseline use 100 ...evaluations...
>
> A: the iteration counts in the experimental results represent the number of evaluations of $f^\star$ (data samples) in total, not the number of batches.  Thus comparisons between for example BO-Muse, AI-alone and standard-BO are relative to the number of data samples - ie in the example you have given, 50 iterations refers to 25 batches.  We have updated the paper to reflect this fact.
>
> Q2: In the experiments with SVMs and RFs, on average, how often was the best design obtained from AI?
>
> A: In our experiments with Random Forests (RFs), we have observed that a human expert, on average, was able to suggest the overall optimal design 72% of the time, whereas AI could guess only 28% of the time. On the other hand, in SVM classification tasks, although AI was suggesting near-optimal solutions, it could select the best design only 17% of time, whereas the expert was able to guess the best design 83% of the time.
>
> Apart from the SVM and RF examples, in the space shield experiment, we observed that the expert found several "best" designs, of which designs 9 and 11 were traditional and designs 27 and 29 were novel solutions inspired by AI design 22.  See the discussion in section 4.2.2 (and A.10.5) for more details.
>
> Q3: Does the paper implicitly assume that the cost of getting a design from humans is negligible compared to evaluating the objective function?
>
> A: We have implicitly assumed that the cost of both the human expert and AI generating their recommendation is negligible compared to the cost of evaluating the objective function.  In our experiments - including the space-shield design experiment - this was certainly true.  As future work it would be interesting to consider the case where this assumption may be abandoned - for example if we replace the expert with costly design software - in which case the operation would resemble something like multi-fidelity BO.  However this is beyond the scope of the present work.
>
> As an aside it is interesting to consider if the associated cost exceeds the benefits of the collaborative exercise, which would very much depend on the precise details of a given application.  For example in our space-craft shielding experiments the expert found the experience itself informative and reported gaining novel insights from the process for future work, which is a potential unforeseen benefit of human-AI teaming.
>
>
> Q4: Unlike the synthetic experiments, the real-world ones do not report performance for the AI-alone model. Thus, it is hard to assess if humans are helpful.
>
> A: As suggested, we have added an AI-alone baseline to our real-world classification experiments using Random Forests (RFs) and Support Vector Machines (SVMs). We have updated our simple regret plots mentioned in Figure 3a and Figure 3b. The new experimental results also demonstrate the superiority of our proposed BO-Muse framework. We acknowledge that we have not included AI-alone baselines in our space shield design experiment due to its expensive experimental evaluations.
>
> CHANGES: Updated regret plots in Figure 3a and Figure 3b.

---

> > ### Author Response · Authors · 2022-11-17
> > **Response to reviewer comments continued**
> >
> > Q5: I understand that the paper focuses on AI-assisted experimental design (humans drive the process). However, it would be interesting to compare BO-Muse to pi-BO, which allows experts to specify informative priors over the design space and then run regular BO (AI model). This would allow us to verify if BO-Muse is the best way to leverage expert knowledge for these experimental design tasks.
> >
> > A: In our work we are interested in the case where the human has insight into the objective function such as an understanding of the underlying physics, similarity between experiments etc; and our goal is to leverage this to collaboratively accelerate optimisation.  By contrast, it is our understanding that pi-BO focuses on the case where an expert is able to give insight into the probability distribution of the optima over design space (ie what neighbourhood they think the optimal solution might lie in).  These are two distinct scenarios - insight into the objective such as an understanding of the physics behind it does not imply knowledge of what the optimal design might be, and vice-versa.  That said, combining these strategies could be a interesting area to explore in future work.
> >
> >
> > Q6: Some of the choices are justified by being adopted in previous works. For instance, "Motivated by Borji and Itti, we assume the human expert in effect generates their recommendations using EC-GP-UCB...". How would BO-Muse perform using another human acquisition function on the synthetic experiments?
> >
> > A: The GP-UCB tradeoff discussed is actually very broad.  Pure exploration corresponds to the case $\zeta \to \infty$, and EI is also covered, though in this case beta is itself a function of posterior mean and variance.  Alternatives that fall outside our framework would be Thompson sampling or information-based criteria, though it is difficult to see how these relate to human experiment selection.
> >
> > RELEVANT CHANGES: We have conducted additional optimisation experiments to understand the behaviour of our proposed BO-Muse framework with the Expected Improvement (EI) acquisition function strategy considered for human experts. As expected BO-Muse with the EI acquisition function still outperforms the other competing baselines. However, BO-Muse with GP-UCB acquisition function has superior performance when compared to its counterpart with the EI acquisition function. We have updated the simple regret plot for the Ackley function to include the results for BO-Muse with the EI acquisition function. The updated regret plot is provided in Figure 6 of Appendix A.8.2.

---

> > > ### Comment · Reviewer_LH2r · 2022-12-11
> > > **Thanks for your response**
> > >
> > > I have read the rebuttal, the concerns raised by other reviewers, and the authors' responses to them. I want to thank the authors for their efforts to address my concerns, especially by clarifying the fairness issue and including further experiments using the AI-alone strategy and another acquisition function.
> > >
> > > I choose to keep my initial rating. The main reason is that I still have concerns regarding the novelty and some modeling assumptions (also raised by other reviewers).

---

### Official Review · Reviewer_XANP · 2022-10-26

**Confidence:** 4
**Correctness:** 4
**Technical Novelty And Significance:** 3
**Empirical Novelty And Significance:** 3
**Recommendation:** 5

**Clarity, Quality, Novelty And Reproducibility:**

As discussed earlier, my primary concern is with the paper's clarity -- I don't believe it sufficiently discusses theoretical assumptions or experimental details thoroughly. This clarity does hurt reproducibility a bit - which parameters of BO-Muse were chosen for the different experiments should be clearly stated.



**Strength And Weaknesses:**

Strengths: I overall really like the idea of bridging bayesian optimization / experiment design with human-AI collaboration, and think this is a promising direction.



W1: The notation throughout the analysis could be significantly improved in terms of clarity and description. Even as minor as stating "dataset" before introducing \mathcal{D}, "kernel function" before K (before Eq 1) would be good to have, as I think the strength of this paper could be its appeal to those who work in human-AI co-creative applications, and thus a more clear  and concise description of Bayesian optimization and GPs for such readers would significantly improve the paper quality (and since the paper's novelty is not on the BO side, and more an application-oriented, I'd argue such clarity is necessary). Sometimes variables are mentioned throughout the paper, and could benefit from reminders on what they are (e.g. max info gain \gamma). It would also help to use more distinctive markings for AI / human suggestions (maybe super scripts AI and H?).
Finally, the regret bounds in 3.3. feel quite isolated from the rest of the paper, and it's not clear what the central takeaway of that section is apart from just providing regret bounds -- without high level implications for parameter selection, it's not clear this is a more valuable part of the paper than, for example, experiment details that were pushed to the Appendix.

W2: Assumptions of human model - The novelty of this paper rests in the application of BO to human-AI co-creativity / experiment design, so I wish there was more thoughtful handling of the assumptions being made about the human in BO-Muse. For example, does the assumption that humans are conservation reasonable for all tasks and settings? Why can we assume the number of features (m_s) in the human model is non-decreasing (a human may discover a major highly predictive feature only later on)? Why can the convergence of the human's max info gain be stronger than the AI's? Why are linear/polynomial models sufficient for the human? It would have been really interesting to even see real-world experiments that try to validate some of these assumptions.

W3: Thoroughness of experiments - It would be helpful if the experiments in the main part of the paper had more thorough details provided. Were ablations over \Beta done for Ackley and Levy functions, so we could get a sense of sensitivity to parameter vs. method? In the Appendix for the Matyas function, why does Simulated Human to better than the proposed BO-Muse method? How does sensitivity to \beta change between tasks?

With the real-world human experiments, how do we know that the benefit of BO-Muse is not just due to humans receiving additional information, regardless of how well-optimized that information was for the task (e.g. a random acquisition heuristic)?

Minor:
- the template and font appears different from other ICLR papers
- could the authors provide more discussion on the limitations of assuming f^star is drawn from the AI's GP?
- there were quite a few grammatical issues , the authors should do another pass to edit for clarity

**Summary Of The Paper:**

The paper proposed a bayesian optimization framework for human-AI teamwork in collaborative tasks such as experiment design and optimizing blackbox functions. The authors assume that humans are likely to over-exploit, and thus propose the AI's role as a source of exploration, and they validate their method on synthetic and real-world optimization experiments.

**Summary Of The Review:**

I am really excited by the problem this paper addresses. Unfortunately, I don't believe it's ready for acceptance in its current state (largely due to clarity and presentation issues), but would be very excited to see this work in a more polished form. While the overarching idea -- providing a formalism for incorporating human domain expertise in optimal experiment design -- is very interesting and timely given the exciting progress we have seen in recent years of neural networks aiding humans in creative tasks / search problems,  I believe the paper focuses too much on the synthetic settings and a rather dense analysis of the Bo-Muse algorithm, leaving room for only minimal real-world experimental details (e.g. the space shield design set-up is almost entirely in the Appendix, whereas I feel like such experiments should be thoroughly analyzed and discussed, even if that means being presented on their own in a separate venue).

---

> ### Author Response · Authors · 2022-11-17
> **Response to reviewer comments**
>
> We are grateful to the reviewer for providing useful feedback on our paper and have revised and updated the paper in accordance with their suggestions.  We will address specific concerns below:
>
> W1: The notation throughout the analysis could be significantly improved in terms of clarity and description...
>
> A: We have thoroughly revised the paper to improve clarity and ensure that important terms and symbols are described where they are used to remind the reader of their significance.  We would like to include a more expansive introduction to Bayesian Optimisation and Gaussian Processes than that presented in section 2.2, but space constraints make this somewhat difficult to achieve.  We have done our best to improve the presentation of this background material and trust that the reader will refer to the citations given for a more detailed background if desired.
>
>
> W1: ...the regret bounds in 3.3. feel quite isolated from the rest of the paper, and it's not clear what the central takeaway of that section is apart from just providing regret bounds -- without high level implications for parameter selection, it's not clear this is a more valuable part of the paper than, for example, experiment details that were pushed to the Appendix.
>
> A: We have completely rewritten section 3.3 to include a additional fore-shadowing of the relevance of this theorem, the context in which it should be read, and how it is intended to be interpreted.  In particular, we have added discussion of the worst-case convergence to demonstrate that BO-Muse will converge like GP-UCB (in the big-O rate sense), and added discussion on what Theorem 1 tells us about the performance of BO-Muse in the more realistic setting.  In particular, the goal of theorem 1 is to show how the presence of a human expert in the loop may accelerate convergence in BO.  As discussed in section 3.3, this occurs due to the lower maximum information gain of the human expert compared to the AI.  See new material in section 3.1 for a full discussion of this point.
>
> As with all things in a page-limited paper there is a balance to be struck between the theoretical aspects and description of our proposed algorithm and the presentation of experimental results.  We have endevoured to include the most relevant experimental results in the paper and place details in the appendix, due in part to previous feedback that the space-shield design experiment may be difficult for people outside of that specialised field to interpret so, while the results are significant, we have tried to incorporate more information on more conventional applications and simulated experiments.  We hope that the interested reader will refer to the appendix if they desire a more thorough discussion of experiments.
>
> RELEVANT CHANGES: we have added additional explanatory material the paper to highlight the key distinguishing features of BO-Muse.  In particular see:
>
> - section 3.3 has been largely rewritten to (a) give more context to convergence discussion, (b) discuss convergence in the worst-case scenario where the human gives arbitrary recommendations and (c) give a more extensive discussion of theorem 1 and how it relates to the problem of accelerated convergence in BO-Muse when the human expert meets our assumptions.

---

> > ### Author Response · Authors · 2022-11-17
> > **Response to reviewer comments continued**
> >
> > W2: Assumptions of human model - ...more thoughtful handling of the assumptions being made about the human...does the assumption that humans are conservation reasonable for all tasks and settings?
> >
> > A: As a result of this feedback we have extensively modified relevant sections in the paper to give a more thorough discussion of the assumptions we are making here.  The regret analysis assumes that:
> >
> > 1. The human expert will maintain an unobserved model of the process or experiment in question. This model need not be perfect or explicit, but as we are dealing with an expert we assume that the human can and will learn as they go, refining their model based on experimental observations.
> >
> > 2. The expert selects points to explore based on an unknown trade-off between exploration and exploitation.  An example of an exploitative choice would be "my improved understanding of the model leads me to believe that this experiment will yield excellent results, so do that", while an example of an explorative choice would be "I don't really know what will happen in these circumstances, but it could be important, so let's run the experiment and see".
> >
> > We emphasise that the details of the human expert's model, and the precise trade-off they make between exploration and exploitation, is unknown and not needed to prove our regret bound.  The expert's model is presumed to evolve as they learn more about the objective, and the expert's trade-off between exploitation and exploration may change from one batch to the next.  When designing the BO strategy we have taken an intentionally "worst-case" view of expert behaviour to ensure convergence in all circumstances.  Our convergence result demonstrates that even under these very loose assumptions, the presence of an expert will lead to accelerated convergence. In fact we note that, even if the expert takes a worst-case approach, BO-Muse will still converge at the same big-O rate as GP-UCB.
> >
> > As noted in the paper - and in particular with reference to Borji and Itti (2013); Dane (2010) - we believe that the assumption that humans are conservative is reasonable.  However it is important to note that this is not restrictive.  The "human's are conservative" assumption constrains our parameter selection insofar as it places a lower bound on how explorative the AI must be, but an AI policy satisfying this bound will also converge as per theorem 1 will still converge if the human expert is non-conservative (see also Lemma 5 in the appendix, which is essentially the "minimal assumption" version of theorem 1).  Indeed, in the extreme case a human following an almost entirely explorative policy the resulting combination (human+AI), which corresponds roughly to GP-UCB-PE (Contal 2013), will still converge according to our analysis.
> >
> > [Contal 2013] Contal, Emile and Buffoni, David and Robicquet, Alexandre and Vayatis, Nicolas, "Parallel Gaussian process optimization with upper confidence bound and pure exploration", Joint European Conference on Machine Learning and Knowledge Discovery in Databases 2013
> >
> > RELEVANT CHANGES: we have added additional explanatory material the paper to highlight the key distinguishing features of BO-Muse.  In particular see:
> >
> > - page 4, paragraph after equation (3) discussing the exploration/exploitation trade-off of the human expert.
> > - section 3.1, after possible definitions of $\hat{K}_s$ discussing the human model, learning, and maximum information gain.
> > - section 3.3 has been largely rewritten to (a) give more context to convergence discussion, (b) discuss convergence in the worst-case scenario where the human gives arbitrary recommendations and (c) give a more extensive discussion of theorem 1 and how it relates to the problem of accelerated convergence in BO-Muse when the human expert meets our assumptions.
> >
> >
> > W2: ...Why can we assume the number of features (m_s) in the human model is non-decreasing (a human may discover a major highly predictive feature only later on)?
> >
> > A: This assumption is not necessary but simplifies the presentation.  In general we could abandon this assumption, so m_s is simply a sequence of positive integers, and discuss the max-information gain relative to the maximum m_s in the sequence (that is, the most pessimistic case).

---

> > > ### Author Response · Authors · 2022-11-17
> > > **Response to reviewer comments continued**
> > >
> > > W2: ...Why can the convergence of the human's max info gain be stronger than the AI's?
> > >
> > > A: As discussed in the updated version of section 3.1, we can reasonably assume that the human starts with a better understanding of the system than the AI.  Precisely, we may safely assume that an expert will have understanding of the underlying physics of the system, the behaviour one might expect in similar experiments etc, so for example the human may have an incomplete but informative set of features that relate to their knowledge of the system or systems similar to it, or an intuitive grasp that certain types of experiment are unlikely to give useful results (which we may, in principle, encode as go/no-go features).  This has two consequences:
> > >
> > > - the prior variance of the expert's GP will vary between a zero-knowledge base level in regions that are a mystery to the expert, and much lower in regions where the expert has a good understanding from past experience, understanding of underlying physics etc (the expert's prior mean is also non-zero in general, enabling low-variance regions in the prior).
> > >
> > > - the prior covariance of the expert's GP (the expert's kernel) will have a structure informed by the expert's understanding and knowledge - precisely a "region A will behave like region B because they have feature/attribute C in common" type understanding.  Thus an experiment in region A will reduce the expert's posterior variance both in region A and region B.
> > >
> > > By contrast, the AI will typically start with a generic kernel prior like an SE kernel, Matern kernel or similar, and a zero prior mean.  Such priors represent a ''flat'' or ''zero knowledge'' prior variance over the entire design space, without the areas of lower prior variance present in the expert's prior, and lack the covariance structure present in the expert's kernel.  Thus an experiment in region A will *only* reduce the variance in region A, but will have no (or negligable) impact elsewhere as the AI has no concept of "region A behaves like region B because they have some feature/attribute in common" (we do not consider the possibility of transfer learning of this structure, as this requires the expert to distill their knowledge in an amenable form which, as we discuss in the paper, is highly non-trivial).
> > >
> > > Thus we see that as the algorithm is run the human's variance will both start at a lower prior on design space and decrease more quickly than the AI.  We know that max information gain is bounded as the sum of the logs of the variance immediately before each experiment (see for example Srinivas, 2012, lemma 5.3), so we may reasonably assume that the expert's max information gain will be lower than the AIs max information gain.
> > >
> > > [Srinivas 2012] Srinivas, Niranjan and Krause, Andreas and Kakade, Sham~M. and Seeger, Matthias~W., "Information-Theoretic Regret Bounds for Gaussian Process Optimization in the Bandit Setting", IEEE Transactions on Information Theory, 2012.
> > >
> > > RELEVANT:
> > >
> > > - section 3.1, after possible definitions of $\hat{K}_s$ discussing the human model, learning, and maximum information gain.
> > > - section 3.3 has been largely rewritten to (a) give more context to convergence discussion, (b) discuss convergence in the worst-case scenario where the human gives arbitrary recommendations and (c) give a more extensive discussion of theorem 1 and how it relates to the problem of accelerated convergence in BO-Muse when the human expert meets our assumptions.
> > >
> > >
> > > W2: ...Why are linear/polynomial models sufficient for the human?
> > >
> > > A: Based on experience with industry partners, we are assuming here that the human is likely to model primary features (but not more complex schema and intuitions) explicitly as a set of function from design space to some set of features.  We can use this in principle to construct a linear kernel, or we can build-in the possibility of second-order, third-order etc corrections by using a second, third or higher order polynomial kernel that includes quadratic, cubic etc terms in its feature map (weight-space).  So for example a first-order approximation of motion might be linear (ignoring friction), which may then be refined to include friction (quadratic terms), and so on.  Of course this feature map will evolve over time, closing the mis-specification gas as the human expert gains a better understanding of the system.
> > >
> > > A more general model is certainly possible and would fit into our framework.  For example an alternative might be that a human thinking in terms of analogy - "experiment X is most like example Y" - in which case an RBF kernel is a better analogue.  More generally still, a better model of human understanding might involve a combination of polynomial, analogous and other (graph-like structures etc).  The key point however is that in principle all of these can be captured by a feature map, similarity measure or similar, thus ultimately falling under the broad umbrella of a kernel method.

---

> > > > ### Author Response · Authors · 2022-11-17
> > > > **Response to reviewer comments continued**
> > > >
> > > > W3: Thoroughness of experiments - It would be helpful if the experiments in the main part of the paper had more thorough details provided. Were ablations over \Beta done for Ackley and Levy functions, so we could get a sense of sensitivity to parameter vs. method? In the Appendix for the Matyas function, why does Simulated Human to better than the proposed BO-Muse method? How does sensitivity to \beta change between tasks?  With the real-world human experiments, how do we know that the benefit of BO-Muse is not just due to humans receiving additional information, regardless of how well-optimized that information was for the task (e.g. a random acquisition heuristic)?
> > > >
> > > > A: As suggested, we conducted the ablation study to understand the sensitivity of \hat{\beta}_{s} in our proposed framework for all the optimisation benchmark functions under consideration.  In the experimental results pertaining to Matyas-2D function, we believe that the superior performance of Simulated Human on this function compared to BO-Muse is due to the low complexity of the search space (so an expert can find the solution before the AI has time to find a non-trivial posterior). Furthermore, as observed in the results, even Generic BO is reaching the best solution within fewer function evaluations confirming the low complexity of the search space.
> > > >
> > > > In the ablation study with varying degrees of exploration-exploitation parameter (\hat{\beta}_{s}), we confirm our proposed idea by observing a common trend that lower the value selected for \hat{\beta}_{s} parameter, better the convergence.  In other words, our proposed BO-Muse finds the optimal design quicker if the human expert is conservative and exploits the known best solutions rather than exploring the search space - which is unsurprising as BO-Muse is tuned for conservative experts. However, if we know a-priori that expert is known or expected to be more explorative, then suitable changes can be made to the \breve{\beta}_{s} parameter in the proposed framework to account for their more explorative behaviour.
> > > >
> > > > As discussed in Section A.10.4, the experimental evaluations are expensive and recruiting human experts with a high level of domain expertise is a formidable task. Particularly, for spacecraft shielding design, we could not find many impact experts for the experiments. Therefore, with limited access to experts, we prioritised our work to evaluate only the designs suggested by the expert, rather than using a random acquisition heuristic to verify the benefits.  However we do note that in our simulated experiments BO-Muse consistently outperformed expert+PE, indicating that the additional information coming from an AI running explorative GP-UCB provides benefits beyond a purely explorative acquisition heuristic.
> > > >
> > > > RELEVANT CHANGES: We have provided additional experimental results in Figure 5a, 5b, and 5c.
> > > >
> > > > M1: the template and font appears different from other ICLR papers
> > > > A: This appears to have occurred due to the quirks of Lyx, so we have moved to pure latex, which seems to have fixed the problem.  We apologise for any confusion caused!
> > > >
> > > > M2: could the authors provide more discussion on the limitations of assuming f^star is drawn from the AI's GP?
> > > > A: We only require that f^\star lie in the RKHS of the AI's kernel, which is non-restrictive as we only use universal kernels here for the AI.  We have amended the paper to reflect this.
> > > >
> > > > M3: there were quite a few grammatical issues , the authors should do another pass to edit for clarity
> > > > A: We have endeavoured to fix any grammatical errors we could find, and have extensively rewritten parts of the paper to improve its clarity.

---

### Official Review · Reviewer_xwm5 · 2022-10-28

**Confidence:** 4
**Correctness:** 3
**Technical Novelty And Significance:** 2
**Empirical Novelty And Significance:** 2
**Recommendation:** 5

**Clarity, Quality, Novelty And Reproducibility:**

### Questions

* Since the problem formulation of this work is similar with the formulation of batch Bayesian optimization, the proposed method should be compared to batch Bayesian optimization strategies.  How do you think about this issue?

* Did you assume a time gap between AI and human recommendations?  I think if there exists the time gap, it encourages us to formulate a more interesting problem.  For example, it can be viewed as an asynchronous Bayesian optimization problem.

* I think the regret bound should be controlled by the amount of human knowledge.  It seems like such a controlling factor does not exist.

### Other issues

* I think that the authors modify a conference template style.  It might violate the policy of conference submissions.

**Strength And Weaknesses:**

### Strengths

* It solves an interesting topic that considers a realistic scenario with human experts.

* The introduction section is very well-written and plausible. In particular, the motivation is strong.

### Weaknesses

* The proposed method is not novel.  I think that it is a simple combination of existing methods.

* Regret analysis is limited, since the regret bound heavily relies on the bound by the AI system.  Since it utilizes the knowledge of human experts that controls how humans know the exact location of solutions, the bound is meaningless.

* Baseline methods are not enough.  I think it should be compared to diverse existing baselines.

**Summary Of The Paper:**

This work solves a problem of sample-efficient experimental design with teaming of a human expert and AI.  Since an AI starts to solve the experimental design from scratch, a human expert can help the AI to accelerate the design process.  To fully utilize such knowledge from human experts and AI system, the authors propose a batch Bayesian optimization technique.  Finally, they provide the experimental results that show the effectiveness of the proposed method.

**Summary Of The Review:**

Please see the text boxes above.

---

> ### Author Response · Authors · 2022-11-17
> **Response to reviewer comments**
>
> We are grateful to the reviewer for providing useful feedback on our paper and have revised and updated the paper in accordance with their suggestions.  We will address specific concerns below:
>
> W1: The proposed method is not novel... combination of existing methods.
>
> A: BO-Muse takes inspiration from batch BO methods but there are key differences that make it novel:
>
> - Motivation: batch-BO methods are design for the situation where multiple experiments can be done simultaneously for significantly less cost than the same number of experiments done sequentially, so the savings in experimental cost/time exceed to potential non-optimality of the experiments suggested.  BO-Muse is designed to accelerate human optimisation in the case where the cost of n experiments is the same whether they are done sequentially or batchwise.  Batching in BO-Muse refers to the grouping of one AI and one human-expert initiated experiment.  We note that experiments are reported in terms of number of evaluations of f^\star, not the number of batches.
>
> - Human expert freedom: unlike batch or other purely AI-led methods, where the algorithm defines exactly how all experiments are selected, in BO-Muse we include the human expert in the loop and make minimal assumptions on the process by which the human expert makes their recommendations.  In particular, we assume that the human expert will maintain some sort of unobserved model of f^\star which could, in principle, be used to derive a kernel; that this model will evolve as the expert gains a better understanding of f^\star; and that they select recommendations using this model based on a trade-off between exploitation ("based on my model, this experiment should yield good results") and exploration ("explore here, I’m curious").  Provided the human expert meets these minimal requirements, we demonstrate that their recommendations will accelerate convergence over vanilla BO (and if the human fails to meet there requirements, we demonstrate that BO-Muse will converge regardless, albeit at a the same asymptotic rate as GP-UCB).
>
> - Interaction: both AI and human expert share the same set of observations.  This helps break the human out of cognitive entrenchment, improving their model of f^\star and allowing them to explore regions they may otherwise have neglected based on eroneous assumptions or biases.
>
> CHANGES:
>
> - pg 4, paragraph after equation (3) discussing the exploration/exploitation trade-off of the human expert.
> - sec 3.1, after possible definitions of $\hat{K}_s$ discussing the human model, learning, and maximum information gain.
>
>
> W2: Regret analysis is limited... relies on... AI system. Since it utilizes the knowledge of human experts that controls how humans know the exact location of solutions, the bound is meaningless.
>
> A: The regret analysis assumes that:
>
> 1. The human expert will maintain an unobserved model of the process or experiment in question. This model need not be perfect or explicit, but as we are dealing with an expert we assume that the human can and will learn as they go, refining their model based on experimental observations.
>
> 2. The expert selects points to explore based on an unknown trade-off between exploration and exploitation.  An example of an exploitative choice would be "my improved understanding of the model leads me to believe that this experiment will yield excellent results, so do that", while an example of an explorative choice would be "I don't really know what will happen in these circumstances, but it could be important, so let's run the experiment and see".
>
> We emphasise that the details of the human expert's model, and the precise trade-off they make between exploration and exploitation, is unknown and not needed to prove our regret bound.  The expert's model is presumed to evolve as they learn more about the objective, and the expert's trade-off between exploitation and exploration may change from one batch to the next.  When designing the BO strategy we have taken an intentionally "worst-case" view of expert behaviour to ensure convergence in all circumstances.  Our convergence result demonstrates that even under these very loose assumptions, the presence of an expert will lead to accelerated convergence. In fact we note that, even if the expert takes a worst-case approach, BO-Muse will still converge at the same big-O rate as GP-UCB.
>
> CHANGES:
>
> - page 4, paragraph after equation (3) discussing the exploration/exploitation trade-off of the human expert.
> - section 3.1, after possible definitions of $\hat{K}_s$ discussing the human model, learning, and maximum information gain.
> - section 3.3 has been largely rewritten to (a) give more context to convergence discussion, (b) discuss convergence in the worst-case scenario where the human gives arbitrary recommendations and (c) give a more extensive discussion of theorem 1 and how it relates to the problem of accelerated convergence in BO-Muse when the human expert meets our assumptions.

---

> > ### Author Response · Authors · 2022-11-17
> > **Response to reviewer comments continued**
> >
> > W3: Baseline methods are not enough. I think it should be compared to diverse existing baselines.
> >
> > A: The aim of BO-Muse is to accelerate human exploration using an AI muse.  When constructing our experiments we have kept this front of mind.  Apart from BO-Muse, our most important baseline is human expert alone, which represents the case we aim to accelerate.  We have also experimented with expert teamed with purely exploratory AI agent, which is intended to test if BO-Muse is providing benefits on top of essentially random data-points to update the human's model and break cognitive entrenchment.  AI-alone (that is, generic GP-UCB-based BO) was also added for completeness, and is intended to provide an example of how performance may deteriorate if there is no human expert in the loop.  We believe that a more exhaustive comparison with the full armada of BO variants is beyond the scope of this work.
> >
> >
> > Q1: Since the problem formulation of this work is similar with the formulation of batch Bayesian optimization, the proposed method should be compared to batch Bayesian optimization strategies. How do you think about this issue?
> >
> > A: We do not believe that comparing with batch-BO would be a relevant comparison.  Batch-BO obtains superior performance in situations where evaluating a batch of n recommendations has the same-ish cost as evaluating a single recommendation (for example if evaluating f^\star requires spinning up a production line which can produce from 1 to n gizmos in roughly the same time).  We are concerned with the sequential case, where every f^\star evaluation has the same cost (no batch discount), and in this case sequential methods are typically superior as models etc can be updated after each experiment, so that each recommendation is better.
> >
> > So it would be interesting to consider the case where a batch discount applies in future work, but this lies outside of the scope of our current investigation.
> >
> >
> > Q2: Did you assume a time gap between AI and human recommendations? I think if there exists the time gap, it encourages us to formulate a more interesting problem. For example, it can be viewed as an asynchronous Bayesian optimization problem.
> >
> > A: We do not assume a time gap between AI and human expert recommendations in the current version of BO-Muse, although we did consider the possibility in earlier iterations of the algorithm (before publication).  There are two ways in which asynchrony could be interpreted here.  Following:
> >
> > Kandasamy et al, "Asynchronous Parallel Bayesian Optimisation via Thompson Sampling"
> >
> > we could have two separate channels of experiment, one AI-led and one human-expert-led, with asynchronous feedback of experimental results between the streams.  Alternatively, we could have asynchrony in-batch, where for example the AI makes a recommendation, and the human factors this into their recommendation.  The latter more-or-less fits our framework but requires modification to our proof to take into account the dependence of the human expert recommendations on the AI recommendation, which could make an interesting extension of the current work.  The former could presumably be analysed by borrowing ideas from Kandasamy et al, but as the human-based BO is not convergent on its own (in the provable sense using GP-UCB style analysis) then it is not clear how you could encode the recquisit, compensatory over-exploration in the AI to ensure accelerated overall convergence when compared to just the AI on its own.

---

> > > ### Author Response · Authors · 2022-11-17
> > > **Response to reviewer comments continued**
> > >
> > > Q3: I think the regret bound should be controlled by the amount of human knowledge. It seems like such a controlling factor does not exist.
> > >
> > > A: In the worst-case, as shown in the updated section 3.3, the algorithm will converge as the same big-O rate as GP-UCB.  However this is pessimistic.  The more realistic case is covered by theorem 1 and the expanded discussion in section 3.3.  The acceleration due to the human-expert in this case is critically dependent on the maximum information gain of the human expert model.
> > >
> > > As discussed in the updated version of section 3.1, we can reasonably assume that the human starts with a better understanding of the system than the AI.  Precisely, we may safely assume that an expert will have understanding of the underlying physics of the system, the behaviour one might expect in similar experiments etc, so for example the human may have an incomplete but informative set of features that relate to their knowledge of the system or systems similar to it, or an intuitive grasp that certain types of experiment are unlikely to give useful results (which we may, in principle, encode as go/no-go features).  This has two consequences:
> > >
> > > - the prior variance of the expert's GP will vary between a zero-knowledge base level in regions that are a mystery to the expert, and much lower in regions where the expert has a good understanding from past experience, understanding of underlying physics etc (the expert's prior mean is also non-zero in general, enabling low-variance regions in the prior).
> > >
> > > - the prior covariance of the expert's GP (the expert's kernel) will have a structure informed by the expert's understanding and knowledge - precisely a "region A will behave like region B because they have feature/attribute C in common" type understanding.  Thus an experiment in region A will reduce the expert's posterior variance both in region A and region B.
> > >
> > > By contrast, the AI will typically start with a generic kernel prior like an SE kernel, Matern kernel or similar, and a zero prior mean.  Such priors represent a ''flat'' or ''zero knowledge'' prior variance over the entire design space, without the areas of lower prior variance present in the expert's prior, and lack the covariance structure present in the expert's kernel.  Thus an experiment in region A will *only* reduce the variance in region A, but will have no (or negligable) impact elsewhere as the AI has no concept of "region A behaves like region B because they have some feature/attribute in common" (we do not consider the possibility of transfer learning of this structure, as this requires the expert to distill their knowledge in an amenable form which, as we discuss in the paper, is highly non-trivial).
> > >
> > > Thus we see that as the algorithm is run the human's variance will both start at a lower prior on design space and decrease more quickly than the AI.  We know that max information gain is bounded as the sum of the logs of the variance immediately before each experiment (see for example Srinivas, 2012, lemma 5.3), so we may reasonably assume that the expert's max information gain will be lower than the AIs max information gain.
> > >
> > > [Srinivas 2012] Srinivas, Niranjan and Krause, Andreas and Kakade, Sham~M. and Seeger, Matthias~W., "Information-Theoretic Regret Bounds for Gaussian Process Optimization in the Bandit Setting", IEEE Transactions on Information Theory, 2012.
> > >
> > > While we make an intentional design choice in BO-Muse to assume ignorance regarding the human expert's model and design choices, in future work it may be interesting to consider the possibility of inferring or modelling expert knowledge and behaviour.  This is a difficult problem that lies outside of the scope of the present paper, but in principle such an analysis would allow us to loosen the constraints on the AI (allowing it to be more exploitative) and even design an AI with a complementary model to that of the human expert.
> > >
> > > RELEVANT CHANGES: we have added additional explanatory material the paper to highlight the key distinguishing features of BO-Muse.  In particular see:
> > >
> > > - section 3.1, after possible definitions of $\hat{K}_s$ discussing the human model, learning, and maximum information gain.
> > > - section 3.3 has been largely rewritten to (a) give more context to convergence discussion, (b) discuss convergence in the worst-case scenario where the human gives arbitrary recommendations and (c) give a more extensive discussion of theorem 1 and how it relates to the problem of accelerated convergence in BO-Muse when the human expert meets our assumptions.
> > >
> > >
> > > I1: I think that the authors modify a conference template style. It might violate the policy of conference submissions.
> > >
> > > This appears to have occurred due to the quirks of Lyx, so we have moved to pure latex, which seems to have fixed the problem.  We apologise for any confusion caused!

---

> > > > ### Comment · Reviewer_xwm5 · 2022-12-01
> > > > **Thank you for your response**
> > > >
> > > > Thanks for your response.
> > > >
> > > > I have read your response and the other reviews.
> > > >
> > > > However, since my concerns on the novelty and the regret analysis still remain, I am going to maintain my score.
> > > >
> > > > In particular, I think that the assumptions in this paper are not minimal.  They are quite strong assumptions and should be expressed with formal factors.

---

### Decision · Program_Chairs · 2023-01-20

**Decision:**

Reject

**Justification For Why Not Higher Score:**

See the main concern (A) mentioned above.

**Justification For Why Not Lower Score:**

N/A.

**Metareview: Summary, Strengths And Weaknesses:**

The main contribution of this work lies in proposing a Bayesian optimization algorithm for human-AI teamwork in optimizing blackbox objective functions, which is interesting to all the reviewers.

After reviewing the authors' rebuttal and an active discussion, a general concern of the reviewers is that of limited novelty -- the authors have used existing GP-UCB-based acquisition functions for this problem. However, this issue can be "offset" by the introduction of an interesting problem in this work.

(A) The main concern raised by all the reviewers remains, that is, the assumptions on the human model are questionable and poorly explained. To verify that these assumptions are indeed practical, the authors can consider conducting a user study. Also, the authors can investigate how the user would respond in high-dimensional input spaces in practice.